# Mixture Weight Estimation and Model Prediction in Multi-source Multi-target Domain Adaptation

**Yuyang Deng**
Pennsylvania State University
yzd82@psu.edu

**Ilja Kuzborskij**
Google DeepMind
iljak@google.com

**Mehrdad Mahdavi**
Pennsylvania State University
mzm616@psu.edu

## Abstract

We consider the problem of learning a model from multiple heterogeneous sources with the goal of performing well on a new target distribution. The goal of learner is to mix these data sources in a target-distribution aware way and simultaneously minimize the empirical risk on the mixed source. The literature has made some tangible advancements in establishing theory of learning on mixture domain. However, there are still two unsolved problems. Firstly, how to estimate the optimal mixture of sources, given a target domain; Secondly, when there are numerous target domains, how to solve empirical risk minimization (ERM) for each target using possibly unique mixture of data sources in a computationally efficient manner. In this paper we address both problems efficiently and with guarantees. We cast the first problem, mixture weight estimation, as a convex-nonconcave compositional minimax problem, and propose an efficient stochastic algorithm with provable stationarity guarantees. Next, for the second problem, we identify that for certain regimes, solving ERM for each target domain individually can be avoided, and instead parameters for a target optimal model can be viewed as a non-linear function on a space of the mixture coefficients. Building upon this, we show that in the offline setting, a GD-trained overparameterized neural network can provably learn such function to *predict* the model of target domain instead of solving a designated ERM problem. Finally, we also consider an online setting and propose a label efficient online algorithm, which predicts parameters for new targets given an arbitrary sequence of mixing coefficients, while enjoying regret guarantees.

## 1 Introduction

With a rapidly increasing amount of decentralized data, multiple source domain adaptation has been an important learning scheme in modern machine learning, e.g., in learning with data collected from multiple sources (e.g. crowdsourcing) or learning in distributed systems where the data can be highly heterogeneous such as federated learning. In this learning scenario, given an input space $\mathcal{X}$ and output space $\mathcal{Y}$, we assume access to $N$ sources of data, each with its own underlying distributions $\mathcal{D}_j, j \in [N]$ over $\mathcal{X} \times \mathcal{Y}$. Then, given i.i.d. training samples $\widehat{\mathcal{D}}_1, \ldots, \widehat{\mathcal{D}}_N$, and a hypothesis space $\mathcal{H}$, our goal is to learn a model on the combination of these sources, for instance through the Empirical Risk Minimization (ERM) procedure $\widehat{h}_{\boldsymbol{\alpha}} = \arg\min_{h \in \mathcal{H}} \sum_{j=1}^{N} \alpha(j) \mathcal{L}_{\widehat{\mathcal{D}}_j}(h)$, where $\mathcal{L}_{\widehat{\mathcal{D}}_j}(h)$ is the empirical loss of a model $h \in \mathcal{H}$ over data samples in $\widehat{\mathcal{D}}_j$, and $\boldsymbol{\alpha} \in \Delta^N$ is some mixing parameter, such that predictor $\widehat{h}_{\boldsymbol{\alpha}}$ entails a good generalization performance on a target domain characterized by a distribution $\mathcal{T}$, i.e., yielding a small true risk $\mathcal{L}_{\mathcal{T}}(\widehat{h}_{\boldsymbol{\alpha}}) = \int \ell(\widehat{h}_{\boldsymbol{\alpha}}(\mathbf{x}), y) \, d\mathcal{T}(\mathbf{x}, y)$. It is natural to measure the quality of $\widehat{h}_{\boldsymbol{\alpha}}$ in terms of the excess risk — namely, the difference between the risk of optimal model for target domain $h_{\mathcal{T}}^* = \arg\min_{h \in \mathcal{H}} \mathcal{L}_{\mathcal{T}}(h)$, and that achieved by $\widehat{h}_{\boldsymbol{\alpha}}$. Clearly, the performance of $\widehat{h}_{\boldsymbol{\alpha}}$ will be influenced by several factors, such as the choice of mixing coefficients

37th Conference on Neural Information Processing Systems (NeurIPS 2023).

$\boldsymbol{\alpha}$ to aggregate the empirical losses, capacity of $\mathcal{H}$, and discrepancy between target and source data distributions. So, in order to design a good procedure for learning $\widehat{h}_{\boldsymbol{\alpha}}$ we need to understand aforementioned trade-offs. Over the years the literature on the multiple source learning has dedicated a considerable attention to this problem [25, 5, 35, 17, 12]. To this end, we consider the following bound on the excess risk of $\widehat{h}_{\boldsymbol{\alpha}}$:

**Theorem 1** (Multi-source learning bound [17]). *Given $N$ source data distributions $\mathcal{D}_1, \ldots, \mathcal{D}_N$ and a target data distribution $\mathcal{T}$, let $\widehat{h}_{\boldsymbol{\alpha}} = \arg\min_{h \in \mathcal{H}} \sum_{j=1}^{N} \alpha(j) \mathcal{L}_{\widehat{\mathcal{D}}_j}(h)$ be the ERM solution with fixed mixture weights $\boldsymbol{\alpha} \in \Delta^N$. Then for any $\nu \geq 0$, with probability at least $1 - 4e^{-\nu}$ it holds that*

$$\mathcal{L}_{\mathcal{T}}(\widehat{h}_{\boldsymbol{\alpha}}) \leq \mathcal{L}_{\mathcal{T}}(h_{\mathcal{T}}^*) + \mathcal{C}(\mathcal{H}, \boldsymbol{\alpha}) + \sup_{h \in \mathcal{H}} \sum_{j=1}^{N} \alpha(j) |\mathcal{L}_{\widehat{\mathcal{T}}}(h) - \mathcal{L}_{\widehat{\mathcal{D}}_j}(h)| + C \sqrt{\frac{\nu}{2} \sum_{j=1}^{N} \frac{\alpha^2(j)}{m_j}}$$

*where $C$ is some constant, the complexity term is $\mathcal{C}(\mathcal{H}, \boldsymbol{\alpha}) := \sum_{j=1}^{N} \alpha(j) \mathfrak{R}_j(\mathcal{H})$ with $\mathfrak{R}_j(\mathcal{H})$ being the Rademacher complexity of $\mathcal{H}$ w.r.t. data source $j$, and $m_j$ is the number of samples from source $j$.*

The above bound indicates that, the generalization ability of a model learnt by ERM on an $\boldsymbol{\alpha}$-combined sources, depends on the $\boldsymbol{\alpha}$-weighted sum of target-source discrepancies, and the number of samples drawn from each source. To entail a good generalization on target domain, it naturally motivates us to minimize right hand side of the bound over $\boldsymbol{\alpha} \in \Delta^N$ to get a *good* mixture parameter. In this paper we can cast this idea as solving the following minimax optimization problem:

$$\min_{\boldsymbol{\alpha} \in \Delta^N} \max_{h \in \mathcal{H}} \sum_{j=1}^{N} \alpha(j) |\mathcal{L}_{\widehat{\mathcal{T}}}(h) - \mathcal{L}_{\widehat{\mathcal{D}}_j}(h)| + C \sqrt{\sum_{j=1}^{N} \frac{\alpha^2(j)}{m_j}}, \tag{1}$$

where we drop the complexity term as it becomes identical for all sources by fixing the hypothesis space $\mathcal{H}$ and bounding it with a computable distribution-independent quantity such as VC dimension [31], or it can be controlled by choice of $\mathcal{H}$ or through data-dependent regularization. [17] gave a simple algorithm to minimize the bound of theorem 1 for binary classifiers and 0-1 loss, however their algorithm does not extend to a more general setting. [24] also looked at minimization of a similar bound with the goal to find weights for an optimal mixture, but they did not give a practical algorithm, nor a provable convergence guarantee. However, none of these works aimed to solve (1) because of its complex structure, and so an efficient algorithm for solving (1) so far has not been proposed. In particular, the first difficulty with (1) is that it is a convex-nonconcave objective, which means all minimax algorithms that require inner concavity [23, 22, 27, 28] or PL-condition [30] will fail to converge to a stationary point. However, recently the literature on optimization of this type of objectives has recently made a tangible progress: The first provable convex-nonconcave algorithm was proposed by [34], where they consider alternating gradient descent ascent algorithm. Their algorithm is deterministic, but in practice, we favor a stochastic gradient method. The second difficulty in solving (1) is its compositional structure, which means that simply replacing gradient with stochastic gradient in [34] will not retain convergence guarantees. To tackle these two difficulties, we propose a *stochastic corrected gradient descent ascent* algorithm, with provable convergence guarantee for solving (1). Our method can be viewed as a variant of the Stochastic Gradient Descent-Ascent (SGDA) algorithm, and moreover here we give a positive answer to the question posed by [34], on *whether an algorithm performing simultaneous updates can optimize convex-nonconcave problem?*, which could be interesting by its own right.

The discussion above concerns learning with one target domain, but in practice, a more common scenario is that we have multiple target domains to adapt to. For example, in federated learning [26], millions of users might wish to learn a good model from multiple sources, which can have good performance on their own data distribution. Hence, we propose to study *Multi-source Multi-target Domain Adaptation* scenario ($M^2$DA). Here we assume that we have $M$ target domains, each of them characterized by its own distribution $\mathcal{T}_i, i \in [M]$ over $\mathcal{X} \times \mathcal{Y}$. Adapting to $M$ different target domains requires different mixture weights $\boldsymbol{\alpha}_1, \ldots, \boldsymbol{\alpha}_M$, either obtained by solving (1), or supplied by the user. Equipped with mixing parameters, next we have to solve $M$ weighted ERM problems to tailor solutions for each target domain $\mathcal{T}_i$, that is

$$\widehat{h}_i = \arg\min_{h \in \mathcal{H}} \sum_{j=1}^{N} \alpha_i(j) \mathcal{L}_{\widehat{\mathcal{D}}_j}(h), \qquad i \in [M].$$

Notice that these $M$ ERM objectives share the same component functions, so we name this problem as a *co-component empirical risk minimization*. A straightforward and naïve approach is to solve all $M$ weighted ERMs individually which becomes computationally inefficient when dealing with a large number of data sources. Nevertheless, given the benign structure of these $M$ ERM problems, we may inquire whether there is a computationally efficient method for discovering all solutions without the necessity of solving each one individually.

We give an affirmative answer to this question by replacing the *learning* of the target model by *predicting* the target model and propose two efficient strategies to learn such predictors (for instance, a neural network). Our algorithm designs are based on the following observation: if we assume that each empirical risk $\mathcal{L}_{\widehat{\mathcal{D}}_j}(h)$ is strongly convex and smooth in parameters of a hypothesis $h$, then the optimal parameters are given by *a Lipschitz function* of mixture weights $\boldsymbol{\alpha}$. More formally, denoting $\mathbf{w}^*(\boldsymbol{\alpha})$ as optimal parameters of a hypothesis $h$ for $\boldsymbol{\alpha}$-weighted ERM, $\mathbf{w}^*(\boldsymbol{\alpha})$ is Lipschitz in $\boldsymbol{\alpha}$. This means that we can learn function $\mathbf{w}^*(\cdot)$ with, say, a neural network and gradient descent, with provably small generalization error. Moreover, analysis of generalization error allows us to understand when such target model prediction is more efficient than direct learning. In particular, we look at such a *phase transition of efficiency*, and conclude that when the number of targets $M$ is much larger than number of sources $N$, i.e, $M \geq \Omega((1/\epsilon)^{N/2})$, learning to predict solution is more efficient than optimizing to solve all $M$ ERMs. When $M$ is relatively smaller, optimizing to solve ERMs is more efficient than learning to predict.

Finally, as a second learning scenario we consider an online learning setting, where mixture weights $\boldsymbol{\alpha}_1, ..., \boldsymbol{\alpha}_M$ are arriving sequentially, and may not even originate from the same distribution. We cast this problem as an online non-parametric regression problem with inexact labels and propose an label-efficient online algorithm to predict models.

**Our contributions**   The main contributions of this paper is summerized as follows:

- We study the multi-source domain adaptation problem where there are multiple source domains and we wish to learn a new model given the mixture of source domains, that can perform well on a given target domain. We build upon existing learning-theoretic results on multi-source domain adaptation and design a new algorithm for weighing of source domains, that casts this problem as a a convex-nonconcave minimax optimization problem. In section 2 we give the first stochastic optimization algorithm for this problem which provably converges to a stationary point. The proposed algorithm is the first provably convergent algorithm for a stochastic *compositional* convex-nonconcave minimax problem.

- We further consider the above adaptation problem with multiple target domains, the Multi-source Multi-target Domain Adaptation ($M^2$DA). We observe that these multiple adaptation might share a common structure, which allows us to avoid solving adaptation problem for each target domain individually, and instead we can replace it by direct prediction of parameters for a new problem. We consider offline and online settings for prediction of target parameters, and propose computationally efficient algorithms for both. For the offline setting, in section 3.1 we propose to use a two-layer neural network to learn optimal parameters $\mathbf{w}^*(\boldsymbol{\alpha})$ using bilevel gradient descent. We show that our algorithm can achieve $O(n^{-\frac{2}{2+N}})$ excess risk. We also identify the regime where our learning based approach is more efficient compared to directly solving each target problem individually.

- Finally, in section 3.2 we focus on scenario where target problems arrive sequentially (and could be dependent) and extend our study of direct target parameter prediction to the online setting. We propose a label-efficient algorithm which enjoys $O(n^{-\frac{1}{1+N}})$ average regret.

**Notation**   We introduce some basic definitions and notation that will be used throughout the paper. Let $\mathbb{B}_q^d(r)$ be the ball in $q$-metric centered at the origin and of radius $r > 0$, and let $\mathbb{S}^{d-1}(r) = \{\mathbf{x} \in \mathbb{R}^d : \|\mathbf{x}\|_2 = r\} \subset \mathbb{R}^d$ be the $\ell_2$-norm unit sphere centered at the origin, and finally let $\mathbb{S}^{d-1} = \mathbb{S}^{d-1}(1)$. In addition, the probability simplex is defined as $\Delta^N = \{\boldsymbol{\alpha} \in [0,1]^N : \|\boldsymbol{\alpha}\|_1 = 1\}$. Concatenation of vectors is denoted by parentheses, that is $(\mathbf{w}_1, \ldots, \mathbf{w}_m) = [\mathbf{w}_1^\top, \ldots, \mathbf{w}_m^\top]^\top$. A vector norm $\|\cdot\|$ is understood as Euclidean norm, while $\|\mathbf{x}\|_\infty = \max_i |x_i|$. For a matrix $\mathbf{M}$, $\|\mathbf{M}\|_{\mathrm{op}}$ denotes its spectral norm while $\|\mathbf{M}\|_F$ is its Frobenius norm. For some $f : \mathbb{S}^{d-1} \to \mathbb{R}$ the *empirical semi-norm* is defined as $\|f\|_n^2 = \frac{1}{n}(f(\mathbf{x}_1)^2 + \cdots + f(\mathbf{x}_n)^2)$ and is always taken

w.r.t. the training sample $S$. In addition, for $g : \mathbb{S}^{d-1} \to \mathbb{R}$, we define an empirical inner product $\langle f, g \rangle_n = \frac{1}{n}(f(\mathbf{x}_1)g(\mathbf{x}_1) + \cdots + f(\mathbf{x}_n)g(\mathbf{x}_n))$. At the same time, $\|f\|_2 = \|f\|_{L^2(P_X)}^2$.

## 2 Mixture Weights Estimation via Convex-nonconcave Minimax Optimization

In this section we focus on a single target domain and present an Algorithm 1 designed to solve a minimax problem (1) to estimate the mixture weights. We assume that hypothesis $h$ is parameterized by a vector space $\{\mathbf{w} \in \mathcal{W} \subseteq \mathbb{R}^d\}$, and use $f_j(\mathbf{w}) = \mathcal{L}_{\widehat{\mathcal{D}}_j}(h)$ to denote the empirical risk over data source $j$. Similarly we define $f_{\widehat{\mathcal{T}}}(\mathbf{w}) = \mathcal{L}_{\widehat{\mathcal{T}}}(h)$. We do the following standard relaxations. First, for the sake of simplicity in computation, we relax the square root on the quadratic term w.r.t. $\boldsymbol{\alpha}$. Second, since the absolute value function is non-smooth, we shall use the smooth approximation function $g$ to replace it, e.g., $g(x) = \sqrt{x^2 + c}$ where $c$ is some small number (here $g(\cdot)$ is smooth approximation of $|\cdot|$). These relaxations lead to solving the following compositional convex-nonconcave minimax optimization problem:

$$\min_{\alpha \in \Delta^N} \max_{\mathbf{w} \in \mathcal{W}} F(\boldsymbol{\alpha}, \mathbf{w}) := \sum_{j=1}^{N} \alpha(j) g(f_{\widehat{\mathcal{T}}}(\mathbf{w}) - f_j(\mathbf{w})) + C\boldsymbol{\alpha}^\top \mathbf{M} \boldsymbol{\alpha} , \tag{2}$$

where $\mathbf{M} = \text{diag}\{\frac{1}{m_1}, \ldots, \frac{1}{m_N}\}$. We are interested in developing a stochastic optimization algorithm to solve (2). It is a strongly-convex-nonconcave minimax problem, and it is one of most difficult type of minimax problem due to the absence of inner concavity. To the best of our knowledge, only Xu *et*

---

**Algorithm 1:** `Mixture Weight Estimation`

---

**Input:** Target domain $\mathcal{T}$, Source domains $\mathcal{D}_1, ..., \mathcal{D}_N$, Initialization variable $\mathbf{w}^0 = \mathbf{w}^{-1}$, $z_1^0, ..., z_N^0$, Positive hyper-parameters $(B, \beta, \eta, \gamma)$ (see theorem 2).
**for** $t = 0, ..., T-1$ **do**

    Sample a minibatch $\xi_{\mathcal{T}}^t$ of size $B$ from target domain $\mathcal{T}$, and $\xi_1^t, \ldots, \xi_N^t$ from source domains $\mathcal{D}_1, \ldots, \mathcal{D}_N$

    $z_j^{t+1} = (1 - \beta^t)\left(z_j^t + f_{\widehat{\mathcal{T}}}(\mathbf{w}^t; \xi_{\mathcal{T}}^t) - f_j(\mathbf{w}^t; \xi_j^t) - (f_{\widehat{\mathcal{T}}}(\mathbf{w}^{t-1}; \xi_{\mathcal{T}}^t) - f_j(\mathbf{w}^{t-1}; \xi_j^t))\right)$
        $+ \beta^t (f_{\widehat{\mathcal{T}}}(\mathbf{w}^t; \xi_{\mathcal{T}}^t) - f_j(\mathbf{w}^t; \xi_j^t))$.

    Compute gradient for $\mathbf{w}$: $\mathbf{g}_{\mathbf{w}}^t = \sum_{j=1}^{N} \alpha_j^t \nabla g(z_j^{t+1})(\nabla f_{\widehat{\mathcal{T}}}(\mathbf{w}^t; \xi_{\mathcal{T}}^t) - \nabla f_j(\mathbf{w}^t; \xi_j^t))$

    $\mathbf{w}^{t+1} = \mathcal{P}_{\mathcal{W}}(\mathbf{w}^t + \gamma \mathbf{g}_{\mathbf{w}}^t)$

    Make vector $\mathbf{v} \in \mathbb{R}^N$ whose $j$th coordinate is $g(z_j^t)$.

    Compute gradient for $\boldsymbol{\alpha}$: $\mathbf{g}_{\boldsymbol{\alpha}}^t = \mathbf{v} + 2C\mathbf{M}\boldsymbol{\alpha}^t$.

    $\boldsymbol{\alpha}^{t+1} = \mathcal{P}_{\Delta^N}(\boldsymbol{\alpha}^t - \eta \mathbf{g}_{\boldsymbol{\alpha}}^t)$

---

*al.* [34] proposed a deterministic algorithm to solve it, but it is still unknown whether a stochastic algorithm can solve it with provable guarantee. We give an affirmative answer to this question, by proposing an algorithm built on celebrated stochastic gradient descent ascent [23]. In addition to nonconcavity nature of 2, another difficulty that arises from the compositional structure of objective is that we cannot simply compute stochastic gradients, namely (with $\mathbb{E}[\cdot] \equiv \mathbb{E}[\cdot \mid \text{data}]$):

$$\mathbb{E}[g'(f_{\widehat{\mathcal{T}}}(\mathbf{w}; \xi_{\mathcal{T}}) - f_j(\mathbf{w}; \xi_j))(\nabla f_{\widehat{\mathcal{T}}}(\mathbf{w}; \xi_{\mathcal{T}}) - \nabla f_j(\mathbf{w}; \xi_j))] \neq g'(f_{\widehat{\mathcal{T}}}(\mathbf{w}) - f_j(\mathbf{w}))(\nabla f_{\widehat{\mathcal{T}}}(\mathbf{w}) - \nabla f_j(\mathbf{w})),$$

where $\xi_1, \xi_2, \ldots$ are independent random elements in sample space $\Xi = \mathcal{X} \times \mathcal{Y}$ that capture stochasticity of the algorithm. To alleviate this issue, we borrow 'the stochastic corrected gradient' idea from [7], and maintain an auxiliary variable by introducing

$$z_j^{t+1} = (1 - \beta^t)\left(z_j^t + f_{\widehat{\mathcal{T}}}(\mathbf{w}^t; \xi_{\mathcal{T}}^t) - f_j(\mathbf{w}^t; \xi_j^t) - (f_{\widehat{\mathcal{T}}}(\mathbf{w}^{t-1}; \xi_{\mathcal{T}}^t) - f_j(\mathbf{w}^{t-1}; \xi_j^t))\right)$$
$$+ \beta^t (f_{\widehat{\mathcal{T}}}(\mathbf{w}^t; \xi_{\mathcal{T}}^t) - f_j(\mathbf{w}^t; \xi_j^t))$$

In words, for each source $j$, we maintain a variable $z_j^t$ serving as a correction term, to ensure that $z_j^t$ is getting closer to inner function component $f_{\widehat{\mathcal{T}}}(\mathbf{w}^t) - f_j(\mathbf{w}^t)$. Relying on the auxiliary variable, we then have gradient estimates

$$\mathbf{g}_{\mathbf{w}}^t = \sum_{j=1}^{N} \alpha^t(j)\nabla g(z_j^{t+1})(\nabla f_{\widehat{\mathcal{T}}}(\mathbf{w}^t; \xi_{\mathcal{T}}^t) - \nabla f_j(\mathbf{w}^t; \xi_j^t)), \quad \mathbf{g}_{\alpha}^t = [g(z_1^t), ..., g(z_N^t)] + 2C\mathbf{M}\boldsymbol{\alpha}^t.$$

Then, denoting the projection operator onto convex set $\mathcal{C}$ by $\mathcal{P}_{\mathcal{C}}(\cdot)$, the update rule becomes

$$\mathbf{w}^{t+1} = \mathcal{P}_{\mathcal{W}}\left(\mathbf{w}^t + \gamma \mathbf{g}_w^t\right), \qquad \boldsymbol{\alpha}^{t+1} = \mathcal{P}_{\Delta^N}\left(\boldsymbol{\alpha}^t - \eta \mathbf{g}_\alpha^t\right).$$

## 2.1 Convergence Analysis

In this section we are going to present the convergence guarantee for Algorithm 1. We make the following standard assumption on objective in (2).

**Assumption 1.** *We make the following assumptions on $g$ and $f$:*

1. *$g(z)$ is $G_g$ Lipschitz and $L_g$ smooth. $f_j(\mathbf{w}; \xi)$ is $G_f$ Lipschitz and $L_f$ smooth, $\forall \mathbf{w} \in \mathcal{W}, j \in [N], \xi \in \Xi$.*

2. *$\mathbb{E}\left\|\nabla f_j(\mathbf{w}; \xi) - \nabla f_j(\mathbf{w})\right\|^2 \leq \sigma^2, \forall \mathbf{w} \in \mathcal{W}$.*

3. *$\max_{\boldsymbol{\alpha} \in \Delta^N, \mathbf{w} \in \mathcal{W}} F(\boldsymbol{\alpha}, \mathbf{w}) \leq F_{max}$, $\max_{\mathbf{w} \in \mathcal{W}} g(f_{\widehat{\tau}}(\mathbf{w}) - f_j(\mathbf{w})) \leq B_g, \forall j \in [N]$.*

Points 1 and 2 of Assumption 1 are standard in the literature on compositional optimization [7]. Point 3 guarantees boundedness of objective value, which can be ensured since we are working in the bounded parameter domain. Assumption 1 also implies the following property of $F$.

**Proposition 1.** *Under Assumption 1, $F(\boldsymbol{\alpha}, \mathbf{w})$ is $L := \max\left\{4G_f^2 L_g + 2G_g L_f, \frac{2C}{m_{\min}}\right\}$ smooth, and $\mu = \frac{2C}{m_{\max}}$ strongly convex in $\boldsymbol{\alpha}$.*

Next, we consider the following convergence measure:

**Definition 1** (Convergence Measure [34]). *Given two parameters, $\boldsymbol{\alpha}$ and $\mathbf{w}$, we define the following quantity as a stationary gap*

$$\nabla G(\boldsymbol{\alpha}, \mathbf{w}) = \begin{pmatrix} \frac{1}{\eta}\left(\boldsymbol{\alpha} - \mathcal{P}_{\Delta^N}\left(\boldsymbol{\alpha} - \eta \nabla_{\boldsymbol{\alpha}} F(\boldsymbol{\alpha}, \mathbf{w})\right)\right) \\ \frac{1}{\gamma}\left(\mathbf{w} - \mathcal{P}_{\mathcal{W}}\left(\mathbf{w} + \gamma \nabla_{\mathbf{w}} F(\boldsymbol{\alpha}, \mathbf{w})\right)\right) \end{pmatrix}.$$

Given the nonconcave nature of (2), we are only able to show the convergence to a stationary point. Definition 1 measures the stationarity given parameter pair $(\boldsymbol{\alpha}, \mathbf{w})$ by examining how much the parameter will change if we run one step projected gradient descent-ascent on them. Alternatively, one could consider the widely employed *primal function* [23] as a convergence measure, $\|\nabla\Phi(\boldsymbol{\alpha})\|$ with $\Phi(\boldsymbol{\alpha}) = \max_{\mathbf{w} \in \mathcal{W}} F(\boldsymbol{\alpha}, \mathbf{w})$, but it is ill-suited to express stationarity since $F(\boldsymbol{\alpha}, \cdot)$ is non-concave.

One of our main results, proved in appendix A, establishes the convergence rate of Algorithm 1:

**Theorem 2.** *Consider Assumption 1 and let $L$ and $\mu$ be defined in Proposition 1. Then, letting $B = \Theta\left(\max\left\{\frac{G_g^2 N \sigma^2}{\epsilon^2}, \frac{\kappa L \sigma^2}{\epsilon^2}\right\}\right)$, $\beta = 0.1$, $\eta = \Theta\left(\frac{\mu}{L^2}\right)$, $\gamma = \Theta\left(\frac{\mu^3}{N G_g^2 G_f^2 L^2}\right)$, the Algorithm 1 guarantees that*

$$\frac{1}{T}\sum_{t=1}^{T} \mathbb{E}\left\|\nabla G(\boldsymbol{\alpha}^t, \mathbf{w}^t)\right\|^2 \leq \epsilon^2$$

*with the gradient complexity bounded by:*

$$O\left(\frac{\kappa L F_{\max}}{\epsilon^2} \cdot \max\left\{\frac{\kappa L \sigma^2}{\epsilon^2}, \frac{G_g^2 N \sigma^2}{\epsilon^2}, 1\right\}\right).$$

To the best of our knowledge, this is the first convergence proof for stochastic algorithm on solving strongly-convex-nonconcave problem. We achieve $O(\epsilon^{-4})$ gradient complexity required to reach an $\epsilon$ stationary point. In contrast to the most relevant result of [34], they show the rate $O(\epsilon^{-2})$ for a *deterministic* Alternating Gradient Projection (AGP) in a strongly-convex-nonconcave setting. Note that our result also positively answers the question posed by [34], on whether some algorithm performing simultaneous instead of alternative updates can optimize strongly-convex-nonconcave minimax problem. Finally, compared to $O(\epsilon^{-4})$ rate of SGDA given a nonconvex-strongly-concave problem [23], we need roughly same stochastic gradient evaluations.

# 3 Multiple Target Domains: Learning to Solve Co-component ERM

Up till now, the main focus was on the problem of learning *good* mixture parameters given a single target domain. Now we turn to a more general setting where we have $M$ target domains, each associated with a different data distribution which necessitates per target mixture weights $\boldsymbol{\alpha}_1, \ldots, \boldsymbol{\alpha}_M$, either obtained by our algorithm, or provided by the user to guarantee good generalization on individual domains. Next, to get personalized models for these $M$ domains, we have to solve $M$ different ERM problems based on these mixture weights:

$$\min_{\mathbf{w} \in \mathcal{W}} f_{\boldsymbol{\alpha}_1}(\mathbf{w}) := \sum_{j=1}^{N} \alpha_1(j) f_j(\mathbf{w}), \quad \cdots, \quad \min_{\mathbf{w} \in \mathcal{W}} f_{\boldsymbol{\alpha}_M}(\mathbf{w}) := \sum_{j=1}^{N} \alpha_n(j) f_j(\mathbf{w});$$

A naïve way is to solve each of them, which will result in a computational complexity of $M$ multiplied by the cost required to minimize each individual $f_{\boldsymbol{\alpha}_i}$ to a desired precision. We note that such a solution does not exploit the benign structure of these ERM problems: they share the same component functions $(f_j)_j$, and the only difference is in the mixture weights. It naturally motivates us to ask, can we propose an efficient algorithm which avoids solving all these $M$ co-component ERM problems from scratch? Consider the solution of $\min_{\mathbf{w} \in \mathcal{W}} f_\alpha(\mathbf{w}) := \sum_{j=1}^{N} \alpha(j) f_j(\mathbf{w})$ as a function of $\boldsymbol{\alpha}$,

$$\mathbf{w}^*(\boldsymbol{\alpha}) := \arg \min_{\mathbf{w} \in \mathcal{W}} \sum_{j=1}^{N} \alpha(j) f_j(\mathbf{w}). \tag{3}$$

Fortunately, if we assume that each source empirical risk $f_j$ is strongly convex and $L_f$ smooth in model parameters, we have the following Lipschitz property (shown in appendix B.1):

**Lemma 1.** *If each $f_j$ is $\mu_f$ strongly convex and $L_f$ smooth, then $\mathbf{w}^*(\cdot)$ is $\kappa^* = \sqrt{N} G_f / \mu_f$ Lipschitz.*

Some basic algebra shows that in the above example $\mathbf{w}^*(\boldsymbol{\alpha})$ is indeed Lipschitz in $\boldsymbol{\alpha}$ with respect to $\ell^2$ metric. The Lipschitz property allows us to learn $\mathbf{w}^*$ efficiently. In particular, learning arbitrary Lipschitz (and bounded) vector-valued function $\mathbf{w}^* : \mathbb{R}^N \to \mathbb{R}^d$ is an instance of a well-studied *nonparametric regression* problem [11]. In the following we will consider algorithms for learning $\mathbf{w}^*$ in both offline and online setting and which are provably capable of estimating $\mathbf{w}^*$ at an almost optimal rate. In offline setting, we assume that we have access to a subset of $M$ mixture weights, say $\boldsymbol{\alpha}_1, ..., \boldsymbol{\alpha}_n$, and we shall use a two layer neural network $\mathbf{h}_{\boldsymbol{\theta}}(\cdot)$ to learn $\mathbf{w}^*(\cdot)$. Our algorithm is GD based empirical risk minimization with adaptive label refining. In a nutshell, given an $\boldsymbol{\alpha}$, since we do not have access to $\mathbf{w}^*(\boldsymbol{\alpha})$, we will use gradient descent to jointly solve weighted ERM with $\boldsymbol{\alpha}$ to get an approximation of $\mathbf{w}^*(\boldsymbol{\alpha})$ as well as optimizing neural network parameters. With a mild distributional assumption on $\boldsymbol{\alpha}$, we show that our algorithm guarantees that the two layer network learns $\mathbf{w}^*(\boldsymbol{\alpha})$, that is, it achieves a small excess risk $\mathbb{E}_{\boldsymbol{\alpha}} \|\mathbf{h}_{\boldsymbol{\theta}}(\boldsymbol{\alpha}) - \mathbf{w}^*(\boldsymbol{\alpha})\|^2$.

In online setting, we assume that we observe an arbitrary sequence $\boldsymbol{\alpha}_1, \boldsymbol{\alpha}_2, ...$ on a simplex, and we wish to predict parameters close to $\mathbf{w}^*(\boldsymbol{\alpha}_1), \mathbf{w}^*(\boldsymbol{\alpha}_2), ....$ As baseline algorithm we will consider a well-known online nonparametric regression that greedily covers the simplex with local online learners and which enjoys almost-optimal regret [13]. However, in the considered online protocol, the algorithm will need access to labels, and revealing each label requires to solve (3) to some desired accuracy. Here we explore a possibility that in practice we might be satisfied with $\epsilon$-average regret, while saving the labelling cost. To this end we propose a modification of the algorithm that randomly skips some labels, while incurring a slightly larger regret.

## 3.1 Offline Setting: Learning Lipschitz function with ReLU Neural Network

In this section we consider offline learning of $\mathbf{w}^\star$. The Lipschitzness guarantees that the $\mathbf{w}^*(\boldsymbol{\alpha})$ function can be efficiently learnt on finite $\boldsymbol{\alpha}$s, and generalizable to unseen $\boldsymbol{\alpha}$. Hence, we propose to use a vector-valued two layer ReLU neural network $\mathbf{h}_{\boldsymbol{\theta}}$ to learn $\mathbf{w}^*(\boldsymbol{\alpha})$.

We consider a two layer vector-valued neural network $\mathbf{h}_{\boldsymbol{\theta}} : \mathbb{R}^N \to \mathbb{R}^d$, $\mathbf{h}_{\boldsymbol{\theta}}(\mathbf{x}) = [\mathbf{a}_1^\top (\mathbf{U}^1 \mathbf{x})_+, ..., \mathbf{a}_d^\top (\mathbf{U}^d \mathbf{x})_+]$, where parameters of the *hidden layer* are matrices $\mathbf{U}^i \in \mathbb{R}^{m \times N}$, collectively captured by the parameter vector $\boldsymbol{\theta} = (\text{vec}(\mathbf{U}^1), \ldots, \text{vec}(\mathbf{U}^d)) \in \mathbb{R}^{dmN}$. Here $\mathbf{a}_i \in \{\pm 1/\sqrt{m}\}^m$ are parameters of the *output layer*. In the following the hidden layer is tuned by Algorithm 2, while parameters of the output layer are fixed throughout training. We assume that at initialization, for each $\mathbf{U}^i$, the first half of its rows are drawn i.i.d. from isotropic standard Gaussian and the remaining half is identical to the first half. Similarly, for each $\mathbf{a}_i$, half of the entries are set

---

**Algorithm 2:** Learning $\mathbf{w}^*$ function by a neural network

---

**Input:** Number of global iteration $T$ , Number of Iteration for inner problem $R$.

**for** $t = 1, ..., T$ **do**

    $\boldsymbol{\theta}^{t+1} = \boldsymbol{\theta}^t - \eta \nabla_{\boldsymbol{\theta}^t} \mathbf{h}_{\boldsymbol{\theta}^t}(\boldsymbol{\alpha}_i)(\mathbf{h}_{\boldsymbol{\theta}^t}(\boldsymbol{\alpha}_i) - \mathbf{w}_i^t)$              ▷ Neural network parameter update

    **for** $i = 1, ..., n$ **do**

        $\mathbf{w}_i^{t+1} = \texttt{GD}(\mathbf{w}_i^t, \boldsymbol{\alpha}_i, K)$                  ▷ Label refining by $K$-step gradient descent

---

---

**Algorithm 3:** $\texttt{GD}(\mathbf{v}, \boldsymbol{\alpha}, K)$

---

Initialize $\mathbf{v}^0 = \mathbf{v}$

**for** $t = 0, ..., K - 1$ **do**

    $\mathbf{v}^{t+1} = \mathcal{P}_{\mathcal{W}} \left( \mathbf{v}^t - \gamma \sum_{j=1}^N \alpha_i(j) \nabla f_j(\mathbf{v}^t) \right),$

**Output:** $\mathbf{v}^K$ .

---

to $-1/\sqrt{m}$ and the rest to $1/\sqrt{m}$ (we assume that $m$ is even). This initialization ensures that each output coordinate is $0$ and so the empirical risk is bounded by a constant at initialization.

We assume that we observe mixture weights $\boldsymbol{\alpha}_1, \ldots, \boldsymbol{\alpha}_n \in \Delta^N$ i.i.d. according to some underlying distribution $\mathcal{U}$. Such mixture weights can be obtained by Algorithm 1 and their independence means that samples originating from target domains are independent from each other.

We learn the neural network by solving the following **Bi-level** ERM:

$$\min_{\boldsymbol{\theta}} \widehat{\mathcal{R}}(\boldsymbol{\theta}) := \frac{1}{n} \sum_{i=1}^n \|\mathbf{h}_{\boldsymbol{\theta}}(\boldsymbol{\alpha}_i) - \mathbf{w}^*(\boldsymbol{\alpha}_i)\|^2, \quad \text{s.t.} \quad \mathbf{w}^*(\boldsymbol{\alpha}_i) = \arg\min_{\mathbf{w} \in \mathcal{W}} \sum_{j=1}^N \alpha_i(j) f_j(\mathbf{w}). \quad (4)$$

The parameters of a neural network $\boldsymbol{\theta}$ should have the well-controlled excess risk on unseen $\boldsymbol{\alpha}$:

$$\mathcal{R}(\boldsymbol{\theta}) = \int_{\Delta^N} \|\mathbf{h}_{\boldsymbol{\theta}}(\boldsymbol{\alpha}) - \mathbf{w}^\star(\boldsymbol{\alpha})\|^2 \, d\mathcal{U}(\boldsymbol{\alpha}) \, .$$

To solve (4), we use a nested loop procedure which performs a GD step on a neural network objective $\widehat{\mathcal{R}}$, while in the inner loop we approximately find 'labels' $\mathbf{w}^*(\boldsymbol{\alpha}_1), ..., \mathbf{w}^*(\boldsymbol{\alpha}_n)$ using a $K$-step GD. The entire procedure for solving (4) is described in Algorithm 2. Then, the following theorem shows that the two layer neural net optimized by Algorithm 2 learns $\mathbf{w}^*(\boldsymbol{\alpha})$:

**Theorem 3.** *Let* $\lambda_0 = N \cdot \text{polylog}(N, n)$. *Consider a neural network of with* $m \geq \Omega\left(n^{8 + \frac{2}{2+N}}\right)$. *Then, for the Algorithm* 2 *with* $\eta \leq \frac{1}{2}$, $\gamma = \frac{1}{L_f}$, $T \geq \Omega\left(\frac{n}{N\lambda_0^2 \eta} \log(n)\right)$, *and* $K \geq \Omega\left(\kappa \log\left(\frac{\eta T n D}{\lambda_0}\right)\right)$, *the following excess risk bound holds with probability at least* $0.99$:

$$\mathcal{R}(\boldsymbol{\theta}_{T+1}) \leq O\left((\kappa^*)^2 d n^{-\frac{2}{2+N}}\right) \, .$$

The proof given in appendix B.5 is based on a more-or-less standard Neural Tangent Kernel (NTK) approximation argument [15, 4], namely we use the key fact that predictions made by a GD-trained overparameterized neural network are close to those made by a Kernelized Least-Squares (KLS) predictor (given that the width of the network is sufficiently large). Now, such a KLS GD-trained predictor can learn Lipschitz target functions: It is well known that by learning on a sufficiently large Reproducing kernel Hilbert space (RKHS) (with polynomial spectral decay), one can approximate Lipschitz functions well [8, 2]. Here our goal is to approximate a vector-valued function, however, by treating each output independently we follow existing proofs [14, 21] for scalar-valued Lipschitz regression by GD-trained neural networks and arrive at the same excess risk times $d$.

**Optimality of our rate.** Here we show that a two-layer neural network trained by a bi-level Gradient Descent (GD) can learn a vector-valued function with $O(dn^{-\frac{2}{2+N}})$ excess risk. If we ignore the dependency on $d$, our result matches the minimax rate of learning a scalar valued Lipschitz function [11].

**Algorithm 4:** Label efficient nonparametric online regression

---

**Data:** Radii schedule $\varepsilon_1, \varepsilon_2 \ldots$ with $\varepsilon_t \in \mathbb{R}_+$, Label efficiency parameter $p \in [0,1]$

$S \leftarrow \varnothing$ ;                                             $\triangleright$ Set of centers

**for** $t = 1, 2, \ldots$ **do**

    Observe $\boldsymbol{\alpha}_t$;

    **if** $S = \varnothing$ **then**

        |   $S \leftarrow \{t\}, \quad T_t \leftarrow \varnothing$ ;                     $\triangleright$ Create initial ball

    $s \leftarrow \arg\min_{s \in S} \|\boldsymbol{\alpha}_t - \boldsymbol{\alpha}_s\|$ ;                   $\triangleright$ Find active center

    **if** $T_s = \varnothing$ **then**

        |   $\hat{\mathbf{w}}_t = \frac{1}{d'}\mathbf{1}$

    **else**

        |   $\hat{\mathbf{w}}_t \leftarrow \frac{1}{|T_s|}\sum_{t' \in T_s} \mathbf{w}_{t'}$ ;            $\triangleright$ Predict using active center

    **if** $\|\boldsymbol{\alpha}_t - \boldsymbol{\alpha}_s\| \leq \varepsilon_t$ **then**

        |   $T_s \leftarrow T_s \cup \{t\}$ ;                        $\triangleright$ Update list for active center

    **else**

        |   $S \leftarrow S \cup \{s\}, \quad T_s \leftarrow \varnothing$ ;          $\triangleright$ Create new center

    Draw a Bernoulli random variable $Z_t$ such that $\mathbb{P}(Z_t = 1) = p$;

    Observe $\mathbf{w}_t = \mathbb{I}\{Z_t = 1\}\,\texttt{GD}(\mathbf{0}, \boldsymbol{\alpha}_t, K)$ ;       $\triangleright$ Update pseudo label for $\boldsymbol{\alpha}_t$

---

**Efficiency of our learning-based approach.** Given $M$ mixture weights $(\boldsymbol{\alpha}_i)_{i=1}^M$, the baseline naïve approach is to solve all $M$ weighted ERM with gradient descent, to accuracy level $\epsilon$, which requires $\Theta(M\kappa \log(1/\epsilon))$ time complexity. Using our approach, we first need to learn a neural network with $\epsilon$ excess risk, and it needs $n = \Theta\left((\kappa^{*2}d/\epsilon)^{1+N/2}\right)$ samples, which implies that we need to solve this many weighted ERM problems, resulting a complexity of $\Theta\left((\kappa^{*2}d/\epsilon)^{1+N/2} \cdot \kappa \log(1/\epsilon)\right)$. Once we learn a neural network, we just need to pay for the inference cost to predict $\mathbf{w}^*$ for each $\boldsymbol{\alpha}_i$. Putting things together, the total time complexity is $\Theta\left((\kappa^{*2}d/\epsilon)^{1+N/2} \cdot \kappa \log(1/\epsilon) + M\right)$. We observe the following regimes:

- When $M \geq \Omega\left(\frac{(\kappa^{*2}d/\epsilon)^{1+N/2} \cdot \kappa \log(1/\epsilon)}{\kappa \log(1/\epsilon) - 1}\right)$, learning is more efficient than solving $M$ ERMs.
- Otherwise, directly solving $M$ ERMs is more efficient than learning a model predictor.

Intuitively, when the number of target domains is much larger than number of source domains, our learning based approach is strictly more efficient. It is also interesting to note that our learning based approach can avoid computational overhead of $M$, but suffers exponential cost from the number of sources $N$. While solving ERMs avoids the price for $N$, the computational cost increases linearly in terms of $M$.

### 3.2 Online Setting: Label Efficient Nonparametric Online Regression

In previous section we discussed nonparametric offline learning of $\mathbf{w}^*$ with distributional assumption on $\boldsymbol{\alpha}_1, ..., \boldsymbol{\alpha}_M$. In this section we consider the following online learning protocol with oblivious adversary. Given a known and fixed parameter $p \in [0,1]$, and an unknown sequence $(\boldsymbol{\alpha}_1, \mathbf{w}^*(\boldsymbol{\alpha}_1)), (\boldsymbol{\alpha}_2, \mathbf{w}^*(\boldsymbol{\alpha}_2)), \cdots \in \Delta^N \times \mathbb{B}_2^d(D)$ of inputs and labels, at every round $t = 1, 2, \ldots$

1. the environment reveals mixture weights $\boldsymbol{\alpha}_t \in \Delta^N$;

2. the learner selects a label $\hat{\mathbf{w}}_t \in \mathbb{B}_2^d(D)$ and incurs loss $\ell_t(\hat{\mathbf{w}}_t) = \|\hat{\mathbf{w}}_t - \mathbf{w}^\star(\boldsymbol{\alpha}_t)\|^2$;

3. the learner samples $Z_t \sim \text{Bern}(p)$ and observes $\mathbb{I}\{Z_t = 1\}\mathbf{w}_t$, when $\mathbf{w}_t$ is a GD-optimized approximation of $\mathbf{w}^\star(\boldsymbol{\alpha}_t)$.

In particular, we introduce Algorithm 4, a modified version of the online nonparametric regression algorithm proposed by [13]. Algorithm 4 iteratively constructs a packing of $\Delta^N$ using $\ell^2$ balls centered on a subset of previously observed inputs. At each step $t$, the label (parameters) associated with the current input $\boldsymbol{\alpha}_t$ is predicted by averaging the labels of past inputs within the ball whose center $\boldsymbol{\alpha}_s$ is closest to $\boldsymbol{\alpha}_t$ in the $\ell^2$ metric (note that labels are vector-valued). If $\boldsymbol{\alpha}_t$ lies outside the nearest ball, a new ball with center $\boldsymbol{\alpha}_t$ is created. The radii $\varepsilon_t$ of all balls shrink at a rate $t^{-1/(1+N)}$.

Note that efficient (log in the number of centers) algorithms for approximate nearest-neighbor search are well-known [19], as well as highly-optimized open-source packages are readily available [16].

In contrast to the original algorithm by [13], where all sequentially observed labels are used to generate predictions (update local learners), our algorithm variant uses only a $p$-fraction of labels on average. This reduces label complexity at the cost of increased regret. This approach is referred to as *label-efficient prediction* in online learning [6, Sec. 6.2] and is beneficial when accessing labels is costly. The following theorem, shown in appendix C, establishes the regret bound of Algorithm 4.

**Theorem 4** (Regret bound). *Let $f : \Delta^N \to \mathbb{R}^d$ be arbitrary $\kappa^*$-Lipschitz function with respect to $\|\cdot\|$ metric and let $C_N > 0$ be a metric-dependent constant. Then, Algorithm 4 with $\varepsilon_t = t^{-\frac{1}{1+N}}$ satisfies*

$$\mathbb{E}\left[\sum_{t=1}^{T} (\ell_t(\hat{\mathbf{w}}_t) - \ell_t(f(\boldsymbol{\alpha}_t)))\right] \leq (pC_N \ln(eT) + 4D\kappa^*)T^{\frac{N}{1+N}} + (1-p)TD + 4pDT(1 - \kappa^{-1})^K D$$

*Moreover, by definition $\ell_t(\mathbf{w}^*(\boldsymbol{\alpha}_t)) = 0$, so choosing $f(\cdot) = \mathbf{w}^*$, the above bound implies that:*

$$\frac{1}{T}\sum_{t=1}^{T}\mathbb{E}[\ell_t(\hat{\mathbf{w}}_t)] \leq 8(pC_N \ln(eT) + 4D\kappa^*)T^{-\frac{1}{1+N}} + (1-p)D + 4pD\left(1 - \kappa^{-1}\right)^K D.$$

The core idea behind the proof (deferred to appendix C) involves maintaining a balance between the regret contribution of each ball and an added regret term arising from approximating the target function using Voronoi partitioning. The regret contribution of each ball is logarithmic in the number of predictions made due to regret of online quadratic optimization [6, p. 42]. Ignoring log factors, the overall regret contribution equals the number of balls, which is essentially governed by the packing number with respect to the $\ell^2$ metric. The additional term in the regret comes from the algorithm's prediction being constant within the Voronoi cells of $\Delta^N$ induced by the current centers (considering that we predict using the nearest neighbor). Thus, an extra term equal to the product of the balls' radius and the Lipschitz constant is incurred. Finally, the label efficient algorithm we present here incurs yet another, $p$-dependent terms, which accounts for the missed labels.

**Corollary 1.** *If our desired average regret is $\epsilon > 0$, then Algorithm 4 has label complexity:*

$$\tilde{O}\left(\max\left\{\left(\frac{D\kappa^*}{\epsilon}\right)^{1+N}, \left(\frac{1-\epsilon/D}{\epsilon}\right)^{1+N}\right\}\right)\left(1 - \frac{\epsilon}{D}\right).$$

Note that we recover the standard version of the algorithm (non-label efficient) by trivially setting $p = 1$, which in contrast has label complexity of order $\tilde{O}(\max\{(D\kappa^*/\epsilon)^{1+N}, (1/\epsilon)^{1+N}\})$, which is strictly larger than the label efficient version as long as $\epsilon$ is not zero. When our desired regret goes to zero, the label complexity of two algorithms will tend to be the same asymptotically.

## 4 Experiments

To demonstrate the effectiveness of our proposed mixture weights estimation algorithm, we conducted an experiment using MNIST dataset [1] according to the following specifications. We consider a scenario with 15 source distributions, and dividing them into 3 groups. For group 1, it contains 5 source distributions and each distribution contains 100 data (80 for training and 20 for testing) samples which are drawn uniformly randomly from class '0', '1' and '2'. Group 2 and 3 share similar settings but their distributions' data are drawn from class '3', '4', '5', and class '6', '7', '9', '10' respectively. The data generation process is summarized in Table 1.

| Group | Classes | Domains per Group | Samples per Domain |
|-------|---------|-------------------|--------------------|
| 1 | 0, 1, 2 | 5 | 100 |
| 2 | 3, 4, 5 | 5 | 100 |
| 3 | 6, 7, 8, 9 | 5 | 100 |

Table 1: Classes and samples per domain for each group

To demonstrate the effectiveness of our Algorithm 1, we implemented and run experiments with two-layer MLP neural network. We choose four different target setting: (1) target distribution

|  | Target (Group 1) | Target (Group 2) | Target (Group 3) | Target (mix of Group 1 and 2) |
|---|---|---|---|---|
| Average ERM | 69.9 % | 40.0 % | 34.9 % | 59.9 % |
| Pure target training | 69.9 % | 55.0% | 40.0 % | 55.0% |
| Our method | 80.0 % | 69.9 % | 55.0 % | 65.0 % |

Table 2: Accuracy comparison with two baseline algorithms. Each row represents the accuracy of model learnt by Average ERM, Pure target training or Our method, on different target domain. We can see that the models learnt using mixture weights from our algorithm (Algorithm 1) always yield the best accuracy.

from Group 1, (2) target distribution from Group 2, (3) target distribution from Group 3, (4) target distribution from the mix of Group 1 and Group 2. We compare three algorithms: 1. Weighted ERM using our learnt weights, 2. ERM on averaged weights and 3. ERM solely on target domain, and presented our findings in Table 2. The results indicate that the accuracy achieved using the learnt alphas outperforms the other two approaches.

## 5 Discussion and Conclusions

In this paper we studied the multi-source multi-target domain adaptation problem. In the first part of the paper we gave an algorithm for adressing a minimax problem, that provably finds good mixture weights of source domains, given a single target domain. In the second part we studied the problem of domain adaptation with multiple target domains, and introduced the co-component ERM problem. We gave two concrete algorithms to solve co-component ERM problem, in offline and online settings. There are several potential future venues for future work, which we briefly discuss below.

**Online mixture weight prediction**   Throughout Section 3, we assumed that the target domains' $\alpha$s are given. However, it would be interesting if given a new target domain, one could predict *good* mixture weight in an online fashion, and our Algorithm 1 could serve as an oracle to give the inexact label. Meanwhile, since the Algorithm 1 takes considerable time to converge, a desired online algorithm should also be label-efficient.

**The complexity of solving co-component ERM**   The co-component ERM problem we introduced in section 3 is interesting from pure optimization perspective. Even though we proposed the learning-based approach to avoid heavy computation, one alternative direction is to develop efficient algorithms to directly solve $M$ co-component ERM problems, and give upper and lower complexity bounds.

**More structure in $\mathbf{w}^\star$ and better phase transition laws**   In this work we considered a basic structure in co-component ERM problems (strong-convexity and smoothness), which gave rise to Lipschitzness of $\mathbf{w}^\star$. Lipschitz class of functions is very large and in general can only be learned at a rate $\Theta(n^{-\frac{2}{2+N}})$. As discussed in section 3.1 this allowed us to argue that the learning approach is more efficient than solving co-component ERM whenever $M \geq \Omega((1/\epsilon)^{N/2})$. However, we could potentially obtain better rates (and better laws) of learning $\mathbf{w}^\star$ having more structure in $\mathbf{w}^\star$. For example, assuming that $\mathbf{w}^\star$ is $H$-times differentiable, the excess risk would behave as $\Theta(n^{-\frac{2H}{2H+N}})$ [11]. Thus, the learning approach would be more efficient when $M \geq \Omega((1/\epsilon)^{N/(2H)})$, that is with potentially much fewer sources than targets. It is an intriguing question to figure our which co-component ERM problems would allows for a nicer structure in $\mathbf{w}^\star$.

**Alternative learning based approach for co-component ERM**   There may be several other learning based method to solve co-component ERM. One potential approach is Meta learning [10]. The idea is to train a meta model $\mathbf{w}_{\mathrm{meta}}$ by optimizing a pre-defined meta objective based on some sampled mixture weights, and the goal would be to find a model that can quickly adapt to tasks with different mixture weights $\boldsymbol{\alpha}$. We leave this as a promising open problem.

## Acknowledgement

The work of YD and MM was partially supported by NSF CAREER Award # 2239374 and CNS Award #1956276.

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

# A Proof of theorem 2: Stationarity of Algorithm 1

## A.1 Proof Sketch

To prove the convergence of Algorithm 1 to a stationary point, we first need to relate our convergence measure to actual iterates during the algorithm dynamics. Defining

$$G_w(\boldsymbol{\alpha}, \mathbf{w}) = \frac{1}{\gamma} \left( \mathbf{w} - \mathcal{P}_{\mathcal{W}} \left( \mathbf{w} + \gamma \nabla_{\mathbf{w}} F(\boldsymbol{\alpha}, \mathbf{w}) \right) \right), \, G_{\alpha}(\boldsymbol{\alpha}, \mathbf{w}) = \frac{1}{\eta} \left( \boldsymbol{\alpha} - \mathcal{P}_{\Lambda} \left( \boldsymbol{\alpha} - \eta \nabla_{\boldsymbol{\alpha}} F(\boldsymbol{\alpha}, \mathbf{w}) \right) \right),$$

the following statements hold:

$$\mathbb{E} \left\| G_{\alpha}(\boldsymbol{\alpha}^t, \mathbf{w}^t) \right\|^2 \leq \frac{2}{\eta^2} \mathbb{E} \left\| \boldsymbol{\alpha}^t - \boldsymbol{\alpha}^{t+1} \right\|^2 + 2G_g^2 \sum_{j=1}^{N} \mathbb{E} \left\| z_j^{t+1} - (f_{\widehat{\mathcal{T}}}(\mathbf{w}^t) - f_j(\mathbf{w}^t)) \right\|^2,$$

$$\mathbb{E} \left\| G_w(\boldsymbol{\alpha}^t, \mathbf{w}^t) \right\|^2 \leq \frac{2}{\gamma^2} \mathbb{E} \left\| \mathbf{w}^t - \mathbf{w}^{t+1} \right\|^2 + 4G_g^2 \frac{\sigma^2}{B} + 8G_f^2 L_g^2 \sum_{j=1}^{N} \alpha^t(j) \mathbb{E} \left\| z_j^{t+1} - (f_{\widehat{\mathcal{T}}}(\mathbf{w}^t) - f_j(\mathbf{w}^t)) \right\|^2.$$

Bounds above show that the gradient norm can be bounded by the difference between two iterates generated by Algorithm 1, and the approximation error of our stochastic correction steps. Hence, we need to quantify how well the proposed stochastic correction step can approximate inner finite functions, i.e., the difference between $z_j^{t+1}$ and $f_{\widehat{\mathcal{T}}}(\mathbf{w}^t) - f_j(\mathbf{w}^t)$. This gap can be bounded following a standard result from [7]:

$$\mathbb{E} \left\| z_j^{t+1} - (f_{\widehat{\mathcal{T}}}(\mathbf{w}^t) - f_j(\mathbf{w}^t)) \right\|^2 \leq (1 - \beta)^2 \mathbb{E} \left\| z_j^t - (f_{\widehat{\mathcal{T}}}(\mathbf{w}^{t-1}) - f_j(\mathbf{w}^{t-1})) \right\|^2$$

$$+ 4(1 - \beta)^2 G_f^2 \left\| \mathbf{w}^t - \mathbf{w}^{t-1} \right\|^2 + 2\beta^2 \frac{\sigma^2}{B}.$$

Besides above tracking error, we will also need to control the iterates gap of auxiliary variable $z_j^t$:

$$\mathbb{E} \left\| z_j^{t+1} - z_j^t \right\|^2 \leq (1 - \frac{\beta}{2})^2 \mathbb{E} \left\| z_j^t - z_j^{t-1} \right\|^2 + 8 \left( 1 + \frac{2}{\beta} \right) G_f^2 \left( \left\| \mathbf{w}^t - \mathbf{w}^{t-1} \right\|^2 + \left\| \mathbf{w}^{t-1} - \mathbf{w}^{t-2} \right\|^2 \right)$$

$$+ 8 \left( 1 + \frac{2}{\beta} \right) \beta^2 G_f^2 \left\| \mathbf{w}^t - \mathbf{w}^{t-1} \right\|^2.$$

It shows that as the primal iterates approach the stationary point (i.e., iterates will not change too much), our auxiliary variable $z_j^t$ also will not change too much.

Next, we are going to characterize the upper bound of $\mathbb{E} \left\| \boldsymbol{\alpha}^t - \boldsymbol{\alpha}^{t+1} \right\|^2, \mathbb{E} \left\| \mathbf{w}^t - \mathbf{w}^{t+1} \right\|^2$. To this end, our proof relies on constructing two-level potential functions. Our first level potential function is defined as

$$\hat{F}^{t+1} := F(\boldsymbol{\alpha}^{t+1}, \mathbf{w}^{t+1}) - \frac{2}{\eta^2 \mu} \|\boldsymbol{\alpha}^{t+1} - \boldsymbol{\alpha}^t\|^2 - O \left( \frac{1}{4\gamma} + L_g^2 G_f^4 \gamma + \frac{\eta L^2}{2} + \frac{NG_g^2}{\mu \eta \beta} \right) \left\| \mathbf{w}^{t+1} - \mathbf{w}^t \right\|^2$$

$$- O \left( \frac{1}{8\gamma} + \frac{NG_g^2}{\mu \eta \beta} \right) \left\| \mathbf{w}^t - \mathbf{w}^{t-1} \right\|^2 + O \left( \frac{7}{2\eta} + \mu - \frac{\eta L^2}{2} \right) \left\| \boldsymbol{\alpha}^{t+1} - \boldsymbol{\alpha}^t \right\|^2$$

Then we prove the following one-iteration relationship:

$$\mathbb{E}[\hat{F}^{t+1} - \hat{F}^t] \geq C_1 \mathbb{E} \left\| \mathbf{w}^{t+1} - \mathbf{w}^t \right\|^2 + \frac{1}{8\gamma} \mathbb{E} \left\| \mathbf{w}^t - \mathbf{w}^{t-1} \right\|^2 + \frac{1}{8\gamma} \mathbb{E} \left\| \mathbf{w}^{t-1} - \mathbf{w}^{t-2} \right\|^2$$

$$+ C_2 \mathbb{E} \left\| \boldsymbol{\alpha}^{t+1} - \boldsymbol{\alpha}^t \right\|^2 - (1 - \beta)^2 O (\gamma + \eta) \sum_{j=1}^{N} \mathbb{E} \left\| z_j^t - (f_{\widehat{\mathcal{T}}}(\mathbf{w}^{t-1}) - f_j(\mathbf{w}^{t-1})) \right\|^2$$

$$- O (\gamma + \beta^2 \gamma) \frac{\sigma^2}{B} - \left( 1 - \frac{\beta}{2} \right)^2 O \left( \frac{1}{\eta} \right) \sum_{j=1}^{N} \mathbb{E} \left\| z_j^t - z_j^{t-1} \right\|^2,$$

where $C_1$ and $C_2$ are some constant depending on $\eta, \gamma, L, \mu$. To eliminate the approximation error $\mathbb{E}\left\|z_j^t - (f_{\widehat{\mathcal{T}}}(\mathbf{w}^{t-1}) - f_j(\mathbf{w}^{t-1}))\right\|^2$ and difference of auxiliary iterates $\mathbb{E}\left\|z_j^t - z_j^{t-1}\right\|^2$, we need construct another potential function $\tilde{F}^{t+1}$:

$$\tilde{F}^{t+1} := \hat{F}^{t+1} - \sum_{j=1}^{N} \mathbb{E}\left\|z_j^{t+1} - (f_{\widehat{\mathcal{T}}}(\mathbf{w}^t) - f_j(\mathbf{w}^t))\right\|^2 - O\left(\frac{(1-\frac{\beta}{2})^2}{\beta}\frac{G_g^2}{\mu^2\eta}\right)\sum_{j=1}^{N}\mathbb{E}\left\|z_j^{t+1} - z_j^t\right\|^2$$

Finally, we arrive at the following bounds.

$$\mathbb{E}[\tilde{F}^{t+1} - \tilde{F}^t] \geq \frac{C_1}{2}\gamma^2\,\mathbb{E}\left\|\nabla_{\mathbf{w}}G(\boldsymbol{\alpha}^t, \mathbf{w}^t)\right\|^2 + \frac{C_2}{2}\eta^2\,\mathbb{E}\left\|\nabla_{\boldsymbol{\alpha}}G(\boldsymbol{\alpha}^t, \mathbf{w}^t)\right\|^2$$

$$- \left(2C_1\gamma^2 L_g^2 N\beta^2 + 2C_2\eta^2 G_g^2 N^2\beta^2 + 4\gamma G_g^2 + 2\beta^2 L_g^2 G_f^2\gamma + 2\beta^2\right)\frac{\sigma^2}{B}\,.$$

Conducting telescoping summation will yield the desired result.

## A.2 Technical Lemmas

**Proposition 2.** *If $g(\cdot)$ is $G_g$ Lipschitz and $L_g$ smooth, and $f_{\widehat{\mathcal{T}}}(\cdot), f_1(\cdot), ..., f_N(\cdot)$ are all $G_f$ Lipschitz and $L_f$ smooth, then $F(\boldsymbol{\alpha}, \mathbf{w})$ is $L := \max\left\{4G_f^2 L_g + 2G_g L_f, \frac{2C}{m_{\min}}\right\}$ smooth, and $\mu = \frac{2C}{m_{\max}}$ strongly convex in $\boldsymbol{\alpha}$.*

*Proof.* We first examine the Lipschitzness of gradient of $F$ w.r.t. $\boldsymbol{\alpha}$.

$$\nabla_{\boldsymbol{\alpha}}F(\boldsymbol{\alpha}, \mathbf{w}) = [g(f_{\widehat{\mathcal{T}}}(\mathbf{w}) - f_1(\mathbf{w})), ..., g(f_{\widehat{\mathcal{T}}}(\mathbf{w}) - f_N(\mathbf{w})))] + 2C\mathbf{M}\boldsymbol{\alpha}$$

Hence

$$\|\nabla_{\boldsymbol{\alpha}}F(\boldsymbol{\alpha}, \mathbf{w}) - \nabla_{\boldsymbol{\alpha}}F(\boldsymbol{\alpha}', \mathbf{w})\| \leq 2C\|\boldsymbol{\alpha} - \boldsymbol{\alpha}'\|$$

Similarly for gradient of $F$ w.r.t. $\mathbf{w}$:

$$\|\nabla_{\mathbf{w}}F(\boldsymbol{\alpha}, \mathbf{w}) - \nabla_{\mathbf{w}}F(\boldsymbol{\alpha}, \mathbf{w}')\|$$

$$\leq \sum_{i=1}^{N}\alpha(i)\left\|\nabla g(f_{\widehat{\mathcal{T}}}(\mathbf{w}) - f_i(\mathbf{w}))(\nabla f_{\widehat{\mathcal{T}}}(\mathbf{w}) - \nabla f_i(\mathbf{w})) - \nabla g(f_{\widehat{\mathcal{T}}}(\mathbf{w}') - f_i(\mathbf{w}'))(\nabla f_{\widehat{\mathcal{T}}}(\mathbf{w}') - \nabla f_i(\mathbf{w}'))\right\|$$

$$\leq \sum_{i=1}^{N}\alpha(i)\left\|\nabla g(f_{\widehat{\mathcal{T}}}(\mathbf{w}) - f_i(\mathbf{w}))(\nabla f_{\widehat{\mathcal{T}}}(\mathbf{w}) - \nabla f_i(\mathbf{w})) - \nabla g(f_{\widehat{\mathcal{T}}}(\mathbf{w}') - f_i(\mathbf{w}'))(\nabla f_{\widehat{\mathcal{T}}}(\mathbf{w}) - \nabla f_i(\mathbf{w}))\right\|$$

$$+ \sum_{i=1}^{N}\alpha(i)\left\|\nabla g(f_{\widehat{\mathcal{T}}}(\mathbf{w}') - f_i(\mathbf{w}'))(\nabla f_{\widehat{\mathcal{T}}}(\mathbf{w}) - \nabla f_i(\mathbf{w})) - \nabla g(f_{\widehat{\mathcal{T}}}(\mathbf{w}') - f_i(\mathbf{w}'))(\nabla f_{\widehat{\mathcal{T}}}(\mathbf{w}') - \nabla f_i(\mathbf{w}'))\right\|$$

$$\leq \sum_{i=1}^{N}\alpha(i)L_g\left\|f_{\widehat{\mathcal{T}}}(\mathbf{w}) - f_i(\mathbf{w}) - (f_{\widehat{\mathcal{T}}}(\mathbf{w}') - f_i(\mathbf{w}'))\right\| \cdot 2G_f + \sum_{i=1}^{N}\alpha(i)2L_f\|\mathbf{w} - \mathbf{w}'\| \cdot G_g$$

$$\leq \sum_{i=1}^{N}\alpha(i)4G_f^2 L_g\|\mathbf{w} - \mathbf{w}'\| + \sum_{i=1}^{N}\alpha(i)2G_g L_f\|\mathbf{w} - \mathbf{w}'\|\,.$$

For strong convexity, we compute Hessian w.r.t. $\boldsymbol{\alpha}$:

$$\nabla_{\boldsymbol{\alpha}}^2 F(\mathbf{w}, \boldsymbol{\alpha}) = 2C\mathbf{M} \succeq 2C\frac{1}{m_{\max}}\mathbf{I},$$

so $F(\boldsymbol{\alpha}, \mathbf{w})$ is $2C\mathbf{M}$ strongly convex in $\boldsymbol{\alpha}$. $\qquad\square$

The following lemma bounds the tracking error of the stochastic correction algorithm.

**Lemma 2** (Tracking Error [7]). *For Algorithm 1, under the assumptions of Theorem 2, the following statement holds true:*

$$\mathbb{E}\left\|z_j^{t+1} - (f_{\widehat{\mathcal{T}}}(\mathbf{w}^t) - f_j(\mathbf{w}^t))\right\|^2 \leq (1-\beta)^2\,\mathbb{E}\left\|z_j^t - (f_{\widehat{\mathcal{T}}}(\mathbf{w}^{t-1}) - f_j(\mathbf{w}^{t-1}))\right\|^2$$

$$+ 4(1-\beta)^2 G_f^2\left\|\mathbf{w}^t - \mathbf{w}^{t-1}\right\|^2 + 2\beta^2\frac{\sigma^2}{B}.$$

An immediate implication of Lemma 2 is the following corollary:

**Corollary 2.** *For Algorithm 1, under the assumptions of Theorem 2, the following statement holds true:*

$$\mathbb{E}\left\|z_j^{T+1} - (f_{\widehat{\mathcal{T}}}(\mathbf{w}^T) - f_j(\mathbf{w}^T))\right\|^2 \le (1-\beta)^{2T}\,\mathbb{E}\left\|z_j^0 - (f_{\widehat{\mathcal{T}}}(\mathbf{w}^{-1}) - f_j(\mathbf{w}^{-1}))\right\|^2$$
$$+ 4\frac{(1-\beta)^4}{1-(1-\beta)^2}\gamma^2 G_f^4 G_g^2 + 2\beta^2\frac{(1-\beta)^2}{1-(1-\beta)^2}\frac{\sigma^2}{B}.$$

*Proof.* Due to Lemma 2 and updating rule for $\mathbf{w}$, we have

$$\mathbb{E}\left\|z_j^{t+1} - (f_{\widehat{\mathcal{T}}}(\mathbf{w}^t) - f_j(\mathbf{w}^t))\right\|^2 \le (1-\beta)^2\,\mathbb{E}\left\|z_j^t - (f_{\widehat{\mathcal{T}}}(\mathbf{w}^{t-1}) - f_j(\mathbf{w}^{t-1}))\right\|^2$$
$$+ 4(1-\beta)^2 G_f^2\,\mathbb{E}\left\|\gamma\mathbf{g}_{\mathbf{w}}^{t-1}\right\|^2 + 2\beta^2\frac{\sigma^2}{B}$$
$$\le (1-\beta)^2\,\mathbb{E}\left\|z_j^t - (f_{\widehat{\mathcal{T}}}(\mathbf{w}^{t-1}) - f_j(\mathbf{w}^{t-1}))\right\|^2$$
$$+ 4(1-\beta)^2 G_f^2 \gamma^2 G_g^2 G_f^2 + 2\beta^2\frac{\sigma^2}{B}.$$

Then we unroll the recursion in Lemma 2:

$$\mathbb{E}\left\|z_j^{T+1} - (f_{\widehat{\mathcal{T}}}(\mathbf{w}^T) - f_j(\mathbf{w}^T))\right\|^2 \le (1-\beta)^{2T}\,\mathbb{E}\left\|z_j^0 - (f_{\widehat{\mathcal{T}}}(\mathbf{w}^{-1}) - f_j(\mathbf{w}^{-1}))\right\|^2$$
$$+ \sum_{t=0}^{T}(1-\beta)^{2t}\left(4(1-\beta)^2\gamma^2 G_g^2 G_f^4 + 2\beta^2\frac{\sigma^2}{B}\right)$$
$$\le (1-\beta)^{2T}\,\mathbb{E}\left\|z_j^0 - (f_{\widehat{\mathcal{T}}}(\mathbf{w}^{-1}) - f_j(\mathbf{w}^{-1}))\right\|^2$$
$$+ \frac{(1-\beta)^2}{1-(1-\beta)^2}\left(4(1-\beta)^2\gamma^2 G_g^2 G_f^4 + 2\beta^2\frac{\sigma^2}{B}\right).$$

$\square$

Besides the above lemma, we also need the following bound on tracking error between two consecutive iterates.

**Lemma 3** (Second Order Tracking Error)**.** *For Algorithm 1, under the assumptions of Theorem 2, the following statement holds true:*

$$\mathbb{E}\left\|z_j^{t+1} - z_j^t\right\|^2 \le (1-\frac{\beta}{2})^2\,\mathbb{E}\left\|z_j^t - z_j^{t-1}\right\|^2 + 8\left(1+\frac{2}{\beta}\right)G_f^2\left(\left\|\mathbf{w}^t - \mathbf{w}^{t-1}\right\|^2 + \left\|\mathbf{w}^{t-1} - \mathbf{w}^{t-2}\right\|^2\right)$$
$$+ 8\left(1+\frac{2}{\beta}\right)\beta^2 G_f^2\left\|\mathbf{w}^t - \mathbf{w}^{t-1}\right\|^2.$$

*Proof.* For the ease of presentation, we define the following two auxiliary variables:

$$f_j^t = f_i(\mathbf{w}^t;\xi_i^t) - f_j(\mathbf{w}^t;\xi_j^t),$$
$$f_j^{t\mapsto t-1} = f_i(\mathbf{w}^t;\xi_i^t) - f_j(\mathbf{w}^t;\xi_j^t) - (f_i(\mathbf{w}^{t-1};\xi_i^t) - f_j(\mathbf{w}^{t-1};\xi_j^t)).$$

According to updating rule of $z$, we have:

$$z_j^{t+1} - z_j^t = (1-\beta)(z_j^t - z_j^{t-1}) + (1-\beta)(f_j^{t\mapsto t-1} - f_j^{t-1\mapsto t-2}) + \beta(f_j^t - f_j^{t-1})$$

Taking expectation w.r.t. $\xi_j^t, \xi_i^t, \xi_j^{t-1}$ and $\xi_i^{t-1}$ yields:

$$
\mathbb{E}\left\|z_j^{t+1} - z_j^t\right\|^2
$$

$$
= \mathbb{E}\left\|(1-\beta)(z_j^t - z_j^{t-1}) + (1-\beta)(f_j^{t\mapsto t-1} - f_j^{t-1\mapsto t-2}) + \beta(f_j^t - f_j^{t-1})\right\|^2
$$

$$
\stackrel{(\Delta)}{\leq} \left(1 + \frac{\beta}{2-2\beta}\right)(1-\beta)^2 \mathbb{E}\left\|z_j^t - z_j^{t-1} + (f_j^{t\mapsto t-1} - f_j^{t-1\mapsto t-2})\right\|^2
$$

$$
+ \left(1 + \frac{2-2\beta}{\beta}\right)\left\|\beta(f_j^t - f_j^{t-1})\right\|^2
$$

$$
\stackrel{(\Theta)}{\leq} \left(1 - \frac{\beta}{2}\right)(1-\beta)\mathbb{E}\left\|z_j^t - z_j^{t-1} + (f_j^{t\mapsto t-1} - f_j^{t-1\mapsto t-2})\right\|^2 + \left(1 + \frac{2}{\beta}\right)\beta^2\left\|f_j^t - f_j^{t-1}\right\|^2
$$

$$
\stackrel{(\Lambda)}{\leq} \left(1 - \frac{\beta}{2}\right)(1-\beta)\left(1 + \frac{\beta}{2-2\beta}\right)\mathbb{E}\left\|z_j^t - z_j^{t-1}\right\|^2
$$

$$
+ \left(1 - \frac{\beta}{2}\right)(1-\beta)\left(1 + \frac{2-2\beta}{\beta}\right)\mathbb{E}\left\|f_j^{t\mapsto t-1} - f_j^{t-1\mapsto t-2}\right\|^2 + \left(1 + \frac{2}{\beta}\right)\beta^2\left\|f_j^t - f_j^{t-1}\right\|^2
$$

$$
\stackrel{(\Xi)}{\leq} \left(1 - \frac{\beta}{2}\right)^2 \mathbb{E}\left\|z_j^t - z_j^{t-1})\right\|^2 + \underbrace{\left(1 + \frac{2}{\beta}\right)\mathbb{E}\left\|f_j^{t\mapsto t-1} - f_j^{t-1\mapsto t-2}\right\|^2}_{T_1} + \underbrace{\left(1 + \frac{2}{\beta}\right)\beta^2\left\|f_j^t - f_j^{t-1}\right\|^2}_{T_2}
$$

where in (1) and (3) we use Young's inequality that $\|\mathbf{a} + \mathbf{b}\|^2 \leq (1+a)\|\mathbf{a}\|^2 + (1 + \frac{1}{a})\|\mathbf{b}\|^2$. Now we bound $T_1$ as follows:

$$
T_1 \leq 4\left(1 + \frac{2}{\beta}\right)\left(\left\|f_i(\mathbf{w}^t; \xi_i^t) - f_i(\mathbf{w}^{t-1}; \xi_i^t)\right\|^2 + \left\|f_j(\mathbf{w}^t; \xi_j^t) - f_j(\mathbf{w}^{t-1}; \xi_j^t)\right\|^2\right)
$$

$$
+ 4\left(1 + \frac{2}{\beta}\right)\left(\left\|f_i(\mathbf{w}^{t-1}; \xi_i^{t-1}) - f_i(\mathbf{w}^{t-2}; \xi_i^{t-1})\right\|^2 + \left\|f_j(\mathbf{w}^{t-1}; \xi_j^{t-1}) - f_j(\mathbf{w}^{t-2}; \xi_j^{t-1})\right\|^2\right)
$$

$$
\leq 8\left(1 + \frac{2}{\beta}\right)G_f^2\left(\left\|\mathbf{w}^t - \mathbf{w}^{t-1}\right\|^2 + \left\|\mathbf{w}^{t-1} - \mathbf{w}^{t-2}\right\|^2\right),
$$

and for $T_2$:

$$
T_2 \leq 8\left(1 + \frac{2}{\beta}\right)\beta^2 G_f^2\left\|\mathbf{w}^t - \mathbf{w}^{t-1}\right\|^2.
$$

Putting pieces together will conclude the proof. $\qquad\square$

The following lemma establishes the difference between the exact gradients computed on $F$ and the gradients we actually used in Algorithm 1.

**Lemma 4** (Gradient difference). *For Algorithm 1, under the assumptions of Theorem 2, the following statement holds true:*

$$
\mathbb{E}\left\|\mathbf{g}_{\mathbf{w}}^t - \nabla_{\mathbf{w}} F(\boldsymbol{\alpha}^t, \mathbf{w}^t)\right\|^2 \leq 4G_g^2\frac{\sigma^2}{B} + 8G_f^2 L_g^2 \sum_{j=1}^N \alpha^t(j)\,\mathbb{E}\left\|z_j^{t+1} - (f_{\widehat{\mathcal{T}}}(\mathbf{w}^t) - \nabla f_j(\mathbf{w}^t))\right\|^2
$$

$$
\mathbb{E}\left\|\mathbf{g}_{\boldsymbol{\alpha}}^t - \nabla_{\boldsymbol{\alpha}} F(\boldsymbol{\alpha}^t, \mathbf{w}^t)\right\|^2 \leq \sum_{j=1}^N G_g^2\,\mathbb{E}\left\|z_j^{t+1} - (f_{\widehat{\mathcal{T}}}(\mathbf{w}^t) - f_j(\mathbf{w}^t))\right\|^2
$$

*Proof.* By the definition of $\mathbf{g}_\mathbf{w}^t$, we have

$$\mathbb{E}\left\|\mathbf{g}_\mathbf{w}^t - \nabla_\mathbf{w} F(\boldsymbol{\alpha}^t, \mathbf{w}^t)\right\|^2$$

$$= \left\|\sum_{j=1}^N \alpha^t(j)\nabla g(z_j^{t+1})(\nabla f_{\widehat{\mathcal{T}}}(\mathbf{w}^t; \xi_i^t) - \nabla f_j(\mathbf{w}^t; \xi_j^t)) - \nabla_\mathbf{w} F(\boldsymbol{\alpha}^t, \mathbf{w}^t)\right\|^2$$

$$\leq 2\,\mathbb{E}\left\|\sum_{j=1}^N \alpha^t(j)\nabla g(z_j^{t+1})(\nabla f_{\widehat{\mathcal{T}}}(\mathbf{w}^t; \xi_i^t) - \nabla f_j(\mathbf{w}^t; \xi_j^t) - (\nabla f_{\widehat{\mathcal{T}}}(\mathbf{w}^t) - \nabla f_j(\mathbf{w}^t)))\right\|^2$$

$$+ 2\,\mathbb{E}\left\|\sum_{j=1}^N \alpha^t(j)(\nabla g(z_j^{t+1}) - \nabla g(f_{\widehat{\mathcal{T}}}(\mathbf{w}^t) - \nabla f_j(\mathbf{w}^t))(\nabla f_{\widehat{\mathcal{T}}}(\mathbf{w}^t) - \nabla f_j(\mathbf{w}^t))\right\|^2$$

$$\leq 4G_g^2 \frac{\sigma^2}{B} + 8G_f^2 L_g^2 \sum_{j=1}^N \alpha^t(j)\,\mathbb{E}\left\|z_j^{t+1} - (f_{\widehat{\mathcal{T}}}(\mathbf{w}^t) - \nabla f_j(\mathbf{w}^t))\right\|^2.$$

where at the last step we use the bounded variance assumption. Similarly by the definition of $\mathbf{g}_{\boldsymbol{\alpha}}^t$ we have:

$$\mathbb{E}\left\|\mathbf{g}_{\boldsymbol{\alpha}}^t - \nabla_{\boldsymbol{\alpha}} F(\boldsymbol{\alpha}^t, \mathbf{w}^t)\right\|^2 = \mathbb{E}\left\|[g(z_1^{t+1}), ..., g(z_N^{t+1})] - \nabla_\mathbf{w} F(\boldsymbol{\alpha}^t, \mathbf{w}^t)\right\|^2$$

$$\leq \sum_{j=1}^N G_g^2\,\mathbb{E}\left\|z_j^{t+1} - (f_{\widehat{\mathcal{T}}}(\mathbf{w}^t) - f_j(\mathbf{w}^t))\right\|^2.$$

$\square$

**Lemma 5** (Connection between stationary measure and iterates). *For Algorithm 1, if we define*

$$G_w(\boldsymbol{\alpha}, \mathbf{w}) = \frac{1}{\gamma}\left(\mathbf{w} - \mathcal{P}_\mathcal{W}\left(\mathbf{w} + \gamma\nabla_\mathbf{w} F(\boldsymbol{\alpha}, \mathbf{w})\right)\right), G_\alpha(\boldsymbol{\alpha}, \mathbf{w}) = \frac{1}{\eta}\left(\boldsymbol{\alpha} - \mathcal{P}_\Lambda\left(\boldsymbol{\alpha} - \eta\nabla_{\boldsymbol{\alpha}} F(\boldsymbol{\alpha}, \mathbf{w})\right)\right),$$

*then the following statements hold:*

$$\mathbb{E}\left\|G_\alpha(\boldsymbol{\alpha}^t, \mathbf{w}^t)\right\|^2 \leq \frac{2}{\eta^2}\,\mathbb{E}\left\|\boldsymbol{\alpha}^t - \boldsymbol{\alpha}^{t+1}\right\|^2 + 2G_g^2 \sum_{j=1}^N \mathbb{E}\left\|z_j^{t+1} - (f_{\widehat{\mathcal{T}}}(\mathbf{w}^t) - f_j(\mathbf{w}^t))\right\|^2,$$

$$\mathbb{E}\left\|G_w(\boldsymbol{\alpha}^t, \mathbf{w}^t)\right\|^2 \leq \frac{2}{\gamma^2}\,\mathbb{E}\left\|\mathbf{w}^t - \mathbf{w}^{t+1}\right\|^2 + 4G_g^2 \frac{\sigma^2}{B} + 8G_f^2 L_g^2 \sum_{j=1}^N \alpha^t(j)\,\mathbb{E}\left\|z_j^{t+1} - (f_{\widehat{\mathcal{T}}}(\mathbf{w}^t) - f_j(\mathbf{w}^t))\right\|^2.$$

*Proof.* We begin with proving the first statement. According to updating rule we have

$$\mathbb{E}\left\|G_{\boldsymbol{\alpha}}(\boldsymbol{\alpha}^t, \mathbf{w}^t)\right\|^2$$

$$= \mathbb{E}\left\|\frac{1}{\eta}\left(\boldsymbol{\alpha}^t - \mathcal{P}_\Lambda\left(\boldsymbol{\alpha}^t - \eta\nabla_{\boldsymbol{\alpha}} F(\boldsymbol{\alpha}^t, \mathbf{w}^t)\right)\right)\right\|^2$$

$$\leq \frac{2}{\eta^2}\,\mathbb{E}\left\|\boldsymbol{\alpha}^t - \mathcal{P}_\Lambda\left(\boldsymbol{\alpha}^t - \eta\mathbf{g}_{\boldsymbol{\alpha}}^t\right)\right\|^2 + \frac{2}{\eta^2}\,\mathbb{E}\left\|\mathcal{P}_\Lambda\left(\boldsymbol{\alpha}^t - \eta\mathbf{g}_{\boldsymbol{\alpha}}^t\right) - \mathcal{P}_\Lambda\left(\boldsymbol{\alpha}^t - \eta\nabla_{\boldsymbol{\alpha}} F(\boldsymbol{\alpha}^t, \mathbf{w}^t)\right)\right\|^2$$

$$\leq \frac{2}{\eta^2}\,\mathbb{E}\left\|\boldsymbol{\alpha}^t - \boldsymbol{\alpha}^{t+1}\right\|^2 + 2\,\mathbb{E}\left\|\mathbf{g}_{\boldsymbol{\alpha}}^t - \nabla_{\boldsymbol{\alpha}} F(\boldsymbol{\alpha}^t, \mathbf{w}^t)\right\|^2$$

$$\leq \frac{2}{\eta^2}\,\mathbb{E}\left\|\boldsymbol{\alpha}^t - \boldsymbol{\alpha}^{t+1}\right\|^2 + 2G_g^2 \sum_{j=1}^N \mathbb{E}\left\|z_j^{t+1} - (f_{\widehat{\mathcal{T}}}(\mathbf{w}^t) - f_j(\mathbf{w}^t))\right\|^2$$

where at last step we apply Lemma 4. Similarly, for the second statement we have:

$$\mathbb{E}\left\|G_{\mathbf{w}}(\boldsymbol{\alpha}^t, \mathbf{w}^t)\right\|^2$$

$$= \mathbb{E}\left\|\frac{1}{\gamma}\left(\mathbf{w}^t - \mathcal{P}_{\mathcal{W}}\left(\mathbf{w}^t + \eta\nabla_{\mathbf{w}}F(\boldsymbol{\alpha}^t, \mathbf{w}^t)\right)\right)\right\|^2$$

$$\leq \frac{2}{\gamma^2}\mathbb{E}\left\|\mathbf{w}^t - \mathcal{P}_{\mathcal{W}}\left(\mathbf{w}^t + \gamma\mathbf{g}_{\mathbf{w}}^t\right)\right\|^2 + \frac{2}{\gamma^2}\mathbb{E}\left\|\mathcal{P}_{\mathcal{W}}\left(\mathbf{w}^t + \gamma\mathbf{g}_{\mathbf{w}}^t\right) - \mathcal{P}_{\mathcal{W}}\left(\mathbf{w}^t + \gamma\nabla_{\mathbf{w}}F(\boldsymbol{\alpha}^t, \mathbf{w}^t)\right)\right\|^2$$

$$\leq \frac{2}{\gamma^2}\mathbb{E}\left\|\mathbf{w}^t - \mathbf{w}^{t+1}\right\| + 2\mathbb{E}\left\|\mathbf{g}_{\mathbf{w}}^t - \nabla_{\mathbf{w}}F(\boldsymbol{\alpha}^t, \mathbf{w}^t)\right\|^2$$

$$\leq \frac{2}{\gamma^2}\mathbb{E}\left\|\mathbf{w}^t - \mathbf{w}^{t+1}\right\|^2 + 4G_g^2\frac{\sigma^2}{B} + 8G_f^2 L_g^2 \sum_{j=1}^{N}\alpha^t(j)\mathbb{E}\left\|z_j^{t+1} - (f_{\widehat{\mathcal{T}}}(\mathbf{w}^t) - \nabla f_j(\mathbf{w}^t))\right\|^2.$$

$$\square$$

The following lemma connects $F(\boldsymbol{\alpha}^{t+1}, \mathbf{w}^{t+1})$ and $F(\boldsymbol{\alpha}^{t+1}, \mathbf{w}^t)$.

**Lemma 6** (Dual Variable One Iteration Analysis). *Let* $L := \max\left\{4G_f^2 L_g + 2G_g L_f, 2C\right\}$. *For Algorithm 1, under the assumptions of Theorem 2, the following statement holds true:*

$$\mathbb{E}[F(\boldsymbol{\alpha}^{t+1}, \mathbf{w}^{t+1}) - F(\boldsymbol{\alpha}^{t+1}, \mathbf{w}^t)]$$

$$\geq \left(\frac{1}{2\gamma} - \frac{L}{2} - \frac{L^2\eta}{2}\right)\mathbb{E}\left\|\mathbf{w}^{t+1} - \mathbf{w}^t\right\|^2 - \frac{1}{2\eta}\mathbb{E}\left\|\boldsymbol{\alpha}^{t+1} - \boldsymbol{\alpha}^t\right\|^2 - \left(2\gamma G_g^2 + 8\beta^2 L_g^2 G_f^2 \gamma\right)\frac{\sigma^2}{B}$$

$$- 4L_g^2 G_f^2 \gamma \sum_{j=1}^{N}\alpha^t(j)(1-\beta)^2\mathbb{E}\left\|z_j^t - (f_{\widehat{\mathcal{T}}}(\mathbf{w}^{t-1}) - f_j(\mathbf{w}^{t-1}))\right\|^2$$

$$- 16(1-\beta)^2 L_g^2 G_f^4 \gamma \mathbb{E}\left\|\mathbf{w}^t - \mathbf{w}^{t-1}\right\|^2.$$

*Proof.* By the optimality of projection and updating rule of $\mathbf{w}$, we have:

$$\left\langle \gamma\mathbf{g}_{\mathbf{w}}^t + (\mathbf{w}^t - \mathbf{w}^{t+1}), \mathbf{w}^t - \mathbf{w}^{t+1}\right\rangle \leq 0$$

Re-arranging terms yields:

$$\left\langle \mathbf{g}_{\mathbf{w}}^t, \mathbf{w}^t - \mathbf{w}^{t+1}\right\rangle \leq -\frac{1}{\gamma}\|\mathbf{w}^{t+1} - \mathbf{w}^t\|^2$$

Adding and subtracting $\nabla_{\mathbf{w}}F(\boldsymbol{\alpha}^t, \mathbf{w}^t)$ gives:

$$\left\langle \nabla_{\mathbf{w}}F(\boldsymbol{\alpha}^t, \mathbf{w}^t), \mathbf{w}^t - \mathbf{w}^{t+1}\right\rangle + \left\langle \mathbf{g}_{\mathbf{w}}^t - \nabla_{\mathbf{w}}F(\boldsymbol{\alpha}^t, \mathbf{w}^t), \mathbf{w}^t - \mathbf{w}^{t+1}\right\rangle \leq -\frac{1}{\gamma}\|\mathbf{w}^{t+1} - \mathbf{w}^t\|^2$$

Applying Cauchy-Schwartz inequality yields:

$$\frac{1}{\gamma}\|\mathbf{w}^{t+1} - \mathbf{w}^t\|^2 \leq \left\langle \nabla_{\mathbf{w}}F(\boldsymbol{\alpha}^t, \mathbf{w}^t), \mathbf{w}^{t+1} - \mathbf{w}^t\right\rangle + \frac{1}{2\gamma}\left\|\mathbf{w}^{t+1} - \mathbf{w}^t\right\|^2 + \frac{1}{2}\gamma\left\|\mathbf{g}_{\mathbf{w}}^t - \nabla_{\mathbf{w}}F(\boldsymbol{\alpha}^t, \mathbf{w}^t)\right\|^2$$

Now, we take expectation over the randomness of $\zeta_{\mathcal{T}}^t$, $\zeta_j^t$, and apply Lemma 4 to get:

$$\frac{1}{2\gamma}\mathbb{E}\|\mathbf{w}^{t+1} - \mathbf{w}^t\|^2 \leq \mathbb{E}\left\langle \nabla_{\mathbf{w}}F(\boldsymbol{\alpha}^t, \mathbf{w}^t), \mathbf{w}^{t+1} - \mathbf{w}^t\right\rangle + 2\gamma G_g^2\frac{\sigma^2}{B}$$

$$+ 4\gamma G_f^2 L_g^2 \sum_{j=1}^{N}\alpha^t(j)\mathbb{E}\left\|z_j^{t+1} - (f_{\widehat{\mathcal{T}}}(\mathbf{w}^t) - \nabla f_j(\mathbf{w}^t))\right\|^2.$$

Plugging in Lemma 2 yields:

$$\frac{1}{2\gamma}\,\mathbb{E}\,\|\mathbf{w}^{t+1} - \mathbf{w}^t\|^2 \leq \mathbb{E}\,\langle \nabla_{\mathbf{w}} F(\boldsymbol{\alpha}^t, \mathbf{w}^t), \mathbf{w}^{t+1} - \mathbf{w}^t \rangle + 2\gamma G_g^2 \frac{\sigma^2}{B}$$

$$+ 4\gamma G_f^2 L_g^2 \sum_{j=1}^{N} \alpha^t(j)\left( (1-\beta)^2\,\mathbb{E}\,\left\| z_j^t - (f_{\hat{\mathcal{T}}}(\mathbf{w}^{t-1}) - f_j(\mathbf{w}^{t-1})) \right\|^2 \right.$$

$$\left. + 4(1-\beta)^2 G_f^2 \left\| \mathbf{w}^t - \mathbf{w}^{t-1} \right\|^2 + 2\beta^2 \frac{\sigma^2}{B} \right).$$

On the other hand, from Proposition 2 on the smoothness of $F$ with parameter $L$ we have:

$$F(\boldsymbol{\alpha}^{t+1}, \mathbf{w}^{t+1}) - F(\boldsymbol{\alpha}^{t+1}, \mathbf{w}^t)$$

$$\geq \langle \nabla_{\mathbf{w}} F(\boldsymbol{\alpha}^{t+1}, \mathbf{w}^t),\, \mathbf{w}^{t+1} - \mathbf{w}^t \rangle - \frac{L}{2}\,\|\mathbf{w}^{t+1} - \mathbf{w}^t\|^2$$

$$= \langle \nabla F(\boldsymbol{\alpha}^t, \mathbf{w}^t),\, \mathbf{w}^{t+1} - \mathbf{w}^t \rangle - \frac{L}{2}\,\|\mathbf{w}^{t+1} - \mathbf{w}^t\|^2 + \langle \nabla F(\boldsymbol{\alpha}^{t+1}, \mathbf{w}^t) - \nabla F(\boldsymbol{\alpha}^t, \mathbf{w}^t),\, \mathbf{w}^{t+1} - \mathbf{w}^t \rangle$$

$$\geq \langle \nabla F(\boldsymbol{\alpha}^t, \mathbf{w}^t),\, \mathbf{w}^{t+1} - \mathbf{w}^t \rangle - \frac{L}{2}\,\|\mathbf{w}^{t+1} - \mathbf{w}^t\|^2$$

$$- \frac{1}{L\sqrt{\eta}}\,\left\| \nabla F(\boldsymbol{\alpha}^{t+1}, \mathbf{w}^t) - \nabla F(\boldsymbol{\alpha}^t, \mathbf{w}^t) \right\| L\sqrt{\eta}\,\|\mathbf{w}^{t+1} - \mathbf{w}^t\|$$

$$\geq \langle \nabla F(\boldsymbol{\alpha}^t, \mathbf{w}^t),\, \mathbf{w}^{t+1} - \mathbf{w}^t \rangle - \frac{L}{2}\,\|\mathbf{w}^{t+1} - \mathbf{w}^t\|^2 - \frac{1}{2\eta}\,\|\boldsymbol{\alpha}^{t+1} - \boldsymbol{\alpha}^t\|^2 - \frac{L^2\eta}{2}\,\|\mathbf{w}^{t+1} - \mathbf{w}^t\|^2$$

$$\geq \langle \nabla F(\boldsymbol{\alpha}^t, \mathbf{w}^t),\, \mathbf{w}^{t+1} - \mathbf{w}^t \rangle - \left( \frac{L}{2} + \frac{L^2\eta}{2} \right)\|\mathbf{w}^{t+1} - \mathbf{w}^t\|^2 - \frac{1}{2\eta}\,\|\boldsymbol{\alpha}^{t+1} - \boldsymbol{\alpha}^t\|^2.$$

Putting pieces together will conclude the proof:

$$\mathbb{E}[F(\boldsymbol{\alpha}^{t+1}, \mathbf{w}^{t+1}) - F(\boldsymbol{\alpha}^{t+1}, \mathbf{w}^t)]$$

$$\geq \left( \frac{1}{2\gamma} - \frac{L}{2} - \frac{L^2\eta}{2} \right)\mathbb{E}\,\|\mathbf{w}^{t+1} - \mathbf{w}^t\|^2 - \frac{1}{2\eta}\,\mathbb{E}\,\|\boldsymbol{\alpha}^{t+1} - \boldsymbol{\alpha}^t\|^2 - \left( 2\gamma G_g^2 + 8\beta^2 L_g^2 G_f^2 \gamma \right)\frac{\sigma^2}{B}$$

$$- 4L_g^2 G_f^2 \gamma \sum_{j=1}^{N} \alpha^t(j)(1-\beta)^2\,\mathbb{E}\,\left\| z_j^t - (f_{\hat{\mathcal{T}}}(\mathbf{w}^{t-1}) - f_j(\mathbf{w}^{t-1})) \right\|^2$$

$$- 16(1-\beta)^2 L_g^2 G_f^4 \gamma\,\mathbb{E}\,\|\mathbf{w}^t - \mathbf{w}^{t-1}\|^2.$$

$\square$

The following lemma connects $F(\boldsymbol{\alpha}^{t+1}, \mathbf{w}^{t+1})$ and $F(\boldsymbol{\alpha}^t, \mathbf{w}^t)$.

**Lemma 7** (One Iteration Descent Lemma). *For Algorithm 1, under the assumptions of Theorem 2, the following statement holds true:*

$$\mathbb{E}[F(\boldsymbol{\alpha}^{t+1}, \mathbf{w}^{t+1}) - F(\boldsymbol{\alpha}^t, \mathbf{w}^t)]$$

$$\geq \left( \frac{1}{2\gamma} - \frac{L}{2} - \frac{L^2\eta}{2} \right)\mathbb{E}\,\|\mathbf{w}^{t+1} - \mathbf{w}^t\|^2 - \left( 16(1-\beta)^2 L_g^2 G_f^4 \gamma + \frac{L^2\eta}{2} \right)\mathbb{E}\,\|\mathbf{w}^t - \mathbf{w}^{t-1}\|^2$$

$$+ \left( \frac{\mu}{2} - \frac{3}{2\eta} - \frac{1}{4\eta(1-\beta)^2} \right)\mathbb{E}\,\|\boldsymbol{\alpha}^{t+1} - \boldsymbol{\alpha}^t\|^2 + \left( \mu - \frac{1}{2\eta} - \frac{\eta L^2}{2} \right)\mathbb{E}\,\|\boldsymbol{\alpha}^t - \boldsymbol{\alpha}^{t-1}\|^2$$

$$- \sum_{j=1}^{N} (1-\beta)^2 \left( 4\alpha^t(j) L_g^2 G_f^2 \gamma + \eta G_g^2 \right)\mathbb{E}\,\left\| z_j^t - (f_{\hat{\mathcal{T}}}(\mathbf{w}^{t-1}) - f_j(\mathbf{w}^{t-1})) \right\|^2$$

$$- \left( 2\gamma G_g^2 + 8\beta^2 L_g^2 G_f^2 \gamma \right)\frac{\sigma^2}{B}.$$

*Proof.* According to property of projection we have:

$$\left\langle \mathbf{g}_\alpha^t + \frac{1}{\eta}(\boldsymbol{\alpha}^{t+1} - \boldsymbol{\alpha}^t), \boldsymbol{\alpha} - \boldsymbol{\alpha}^{t+1} \right\rangle \geq 0 \tag{5}$$

$$\left\langle \mathbf{g}_\alpha^t + \frac{1}{\eta}(\boldsymbol{\alpha}^{t+1} - \boldsymbol{\alpha}^t), \boldsymbol{\alpha}^t - \boldsymbol{\alpha}^{t+1} \right\rangle \geq 0 \tag{6}$$

$$\left\langle \mathbf{g}_\alpha^{t-1} + \frac{1}{\eta}(\boldsymbol{\alpha}^t - \boldsymbol{\alpha}^{t-1}), \boldsymbol{\alpha}^{t+1} - \boldsymbol{\alpha}^t \right\rangle \geq 0 \tag{7}$$

Since $F(\cdot, \mathbf{w})$ is strongly convex for any $\mathbf{w} \in \mathcal{W}$, we have:

$$\begin{aligned} F(\boldsymbol{\alpha}^{t+1}, \mathbf{w}^t) - F(\boldsymbol{\alpha}^t, \mathbf{w}^t) &\geq \left\langle \nabla_\alpha F(\boldsymbol{\alpha}^t, \mathbf{w}^t), \boldsymbol{\alpha}^{t+1} - \boldsymbol{\alpha}^t \right\rangle + \frac{\mu}{2} \|\boldsymbol{\alpha}^{t+1} - \boldsymbol{\alpha}^t\|^2 \\ &\geq \left\langle \nabla_\alpha F(\boldsymbol{\alpha}^t, \mathbf{w}^t) - \mathbf{g}_\alpha^{t-1}, \boldsymbol{\alpha}^{t+1} - \boldsymbol{\alpha}^t \right\rangle + \frac{\mu}{2} \|\boldsymbol{\alpha}^{t+1} - \boldsymbol{\alpha}^t\|^2 \\ &\quad - \frac{1}{\eta} \left\langle \boldsymbol{\alpha}^t - \boldsymbol{\alpha}^{t-1}, \boldsymbol{\alpha}^{t+1} - \boldsymbol{\alpha}^t \right\rangle \\ &\geq \left\langle \nabla_\alpha F(\boldsymbol{\alpha}^t, \mathbf{w}^t) - \nabla_\alpha F(\boldsymbol{\alpha}^{t-1}, \mathbf{w}^{t-1}), \boldsymbol{\alpha}^{t+1} - \boldsymbol{\alpha}^t \right\rangle + \frac{\mu}{2} \|\boldsymbol{\alpha}^{t+1} - \boldsymbol{\alpha}^t\|^2 \\ &\quad + \left\langle \mathbf{g}_\alpha^{t-1} - \nabla_\alpha F(\boldsymbol{\alpha}^{t-1}, \mathbf{w}^{t-1}), \boldsymbol{\alpha}^{t+1} - \boldsymbol{\alpha}^t \right\rangle \\ &\quad - \frac{1}{\eta} \left\langle \boldsymbol{\alpha}^t - \boldsymbol{\alpha}^{t-1}, \boldsymbol{\alpha}^{t+1} - \boldsymbol{\alpha}^t \right\rangle \end{aligned}$$

where at second inequality we use the fact of (7) . For the first dot product term, we observe that:

$$\left\langle \nabla_\alpha F(\boldsymbol{\alpha}^t, \mathbf{w}^t) - \nabla_\alpha F(\boldsymbol{\alpha}^{t-1}, \mathbf{w}^{t-1}), \boldsymbol{\alpha}^{t+1} - \boldsymbol{\alpha}^t \right\rangle \tag{8}$$
$$= \left\langle \nabla_\alpha F(\boldsymbol{\alpha}^t, \mathbf{w}^t) - \nabla_\alpha F(\boldsymbol{\alpha}^t, \mathbf{w}^{t-1}), \boldsymbol{\alpha}^{t+1} - \boldsymbol{\alpha}^t \right\rangle$$
$$+ \left\langle \nabla_\alpha F(\boldsymbol{\alpha}^t, \mathbf{w}^{t-1}) - \nabla_\alpha F(\boldsymbol{\alpha}^{t-1}, \mathbf{w}^{t-1}), \Delta \right\rangle$$
$$+ \left\langle \nabla_\alpha F(\boldsymbol{\alpha}^t, \mathbf{w}^{t-1}) - \nabla_\alpha F(\boldsymbol{\alpha}^{t-1}, \mathbf{w}^{t-1}), \boldsymbol{\alpha}^t - \boldsymbol{\alpha}^{t-1} \right\rangle$$
$$\geq -\frac{L^2\eta}{2} \|\mathbf{w}^t - \mathbf{w}^{t-1}\|^2 - \frac{1}{2\eta} \|\boldsymbol{\alpha}^{t+1} - \boldsymbol{\alpha}^t\|^2 - \frac{\eta L^2}{2} \|\boldsymbol{\alpha}^t - \boldsymbol{\alpha}^{t-1}\|^2 - \frac{1}{2\eta} \|\Delta\|^2 + \mu \|\boldsymbol{\alpha}^t - \boldsymbol{\alpha}^{t-1}\|^2 \tag{9}$$

where $\Delta = (\boldsymbol{\alpha}^{t+1} - \boldsymbol{\alpha}^t) - (\boldsymbol{\alpha}^t - \boldsymbol{\alpha}^{t-1})$.

For the second dot product, we have:

$$\begin{aligned} \mathbb{E} &\left\langle \mathbf{g}_\alpha^{t-1} - \nabla_\alpha F(\boldsymbol{\alpha}^{t-1}, \mathbf{w}^{t-1}), \boldsymbol{\alpha}^{t+1} - \boldsymbol{\alpha}^t \right\rangle \\ &\geq -\eta(1-\beta)^2 \mathbb{E} \left\| \mathbf{g}_\alpha^{t-1} - \nabla_\alpha F(\boldsymbol{\alpha}^{t-1}, \mathbf{w}^{t-1}) \right\|^2 - \frac{1}{4\eta(1-\beta)^2} \mathbb{E} \left\| \boldsymbol{\alpha}^{t+1} - \boldsymbol{\alpha}^t \right\|^2 \\ &\geq -\eta(1-\beta)^2 \sum_{j=1}^N G_g^2 \mathbb{E} \left\| z_j^t - (f_{\widehat{\mathcal{T}}}(\mathbf{w}^{t-1}) - f_j(\mathbf{w}^{t-1})) \right\|^2 - \frac{1}{4\eta(1-\beta)^2} \mathbb{E} \left\| \boldsymbol{\alpha}^{t+1} - \boldsymbol{\alpha}^t \right\|^2 . \end{aligned}$$

where we plug in Lemma 4 at last step.

For the third dot product, we use the following identity:

$$\frac{1}{\eta} \left\langle \boldsymbol{\alpha}^t - \boldsymbol{\alpha}^{t-1}, \boldsymbol{\alpha}^{t+1} - \boldsymbol{\alpha}^t \right\rangle = \frac{1}{\eta} \left( \frac{1}{2} \|\boldsymbol{\alpha}^t - \boldsymbol{\alpha}^{t-1}\|^2 + \frac{1}{2} \|\boldsymbol{\alpha}^{t+1} - \boldsymbol{\alpha}^t\|^2 - \frac{1}{2} \|\Delta\|^2 \right) \tag{10}$$

So we have:

$$\begin{aligned} \mathbb{E}[F(\boldsymbol{\alpha}^{t+1}, \mathbf{w}^t) - F(\boldsymbol{\alpha}^t, \mathbf{w}^t)] &\geq -\frac{L^2\eta}{2} \mathbb{E} \|\mathbf{w}^t - \mathbf{w}^{t-1}\|^2 + \left( \frac{\mu}{2} - \frac{1}{\eta} - \frac{1}{4\eta(1-\beta)^2} \right) \mathbb{E} \|\boldsymbol{\alpha}^{t+1} - \boldsymbol{\alpha}^t\|^2 \\ &\quad + \left( \mu - \frac{1}{2\eta} - \frac{\eta L^2}{2} \right) \mathbb{E} \|\boldsymbol{\alpha}^t - \boldsymbol{\alpha}^{t-1}\|^2 \\ &\quad - \eta(1-\beta)^2 \sum_{j=1}^N G_g^2 \mathbb{E} \left\| z_j^t - (f_{\widehat{\mathcal{T}}}(\mathbf{w}^{t-1}) - f_j(\mathbf{w}^{t-1})) \right\|^2 \tag{11} \end{aligned}$$

Recall Lemma 6 gives the following lower bound of $\mathbb{E}[F(\boldsymbol{\alpha}^{t+1}, \mathbf{w}^{t+1}) - F(\boldsymbol{\alpha}^{t+1}, \mathbf{w}^t)]$:

$$\mathbb{E}[F(\boldsymbol{\alpha}^{t+1}, \mathbf{w}^{t+1}) - F(\boldsymbol{\alpha}^{t+1}, \mathbf{w}^t)] \tag{12}$$

$$\geq \left(\frac{1}{2\gamma} - \frac{L}{2} - \frac{L^2\eta}{2}\right) \mathbb{E}\left\|\mathbf{w}^{t+1} - \mathbf{w}^t\right\|^2 - \frac{1}{2\eta}\mathbb{E}\left\|\boldsymbol{\alpha}^{t+1} - \boldsymbol{\alpha}^t\right\|^2 - \left(2\gamma G_g^2 + 8\beta^2 L_g^2 G_f^2\gamma\right)\frac{\sigma^2}{B}$$

$$- 4L_g^2 G_f^2\gamma \sum_{j=1}^N \alpha^t(j)(1-\beta)^2 \mathbb{E}\left\|z_j^t - (f_{\widehat{T}}(\mathbf{w}^{t-1}) - f_j(\mathbf{w}^{t-1}))\right\|^2$$

$$- 16(1-\beta)^2 L_g^2 G_f^4\gamma \mathbb{E}\left\|\mathbf{w}^t - \mathbf{w}^{t-1}\right\|^2. \tag{13}$$

Hence, Adding (11) and (13) yields:

$$\mathbb{E}[F(\boldsymbol{\alpha}^{t+1}, \mathbf{w}^{t+1}) - F(\boldsymbol{\alpha}^t, \mathbf{w}^t)]$$

$$\geq \left(\frac{1}{2\gamma} - \frac{L}{2} - \frac{L^2\eta}{2}\right) \mathbb{E}\left\|\mathbf{w}^{t+1} - \mathbf{w}^t\right\|^2 - \left(16(1-\beta)^2 L_g^2 G_f^4\gamma + \frac{L^2\eta}{2}\right)\mathbb{E}\left\|\mathbf{w}^t - \mathbf{w}^{t-1}\right\|^2$$

$$+ \left(\frac{\mu}{2} - \frac{3}{2\eta} - \frac{1}{4\eta(1-\beta)^2}\right)\mathbb{E}\left\|\boldsymbol{\alpha}^{t+1} - \boldsymbol{\alpha}^t\right\|^2 + \left(\mu - \frac{1}{2\eta} - \frac{\eta L^2}{2}\right)\mathbb{E}\left\|\boldsymbol{\alpha}^t - \boldsymbol{\alpha}^{t-1}\right\|^2$$

$$- \sum_{j=1}^N (1-\beta)^2\left(4\alpha^t(j)L_g^2 G_f^2\gamma + \eta G_g^2\right)\mathbb{E}\left\|z_j^t - (f_{\widehat{T}}(\mathbf{w}^{t-1}) - f_j(\mathbf{w}^{t-1}))\right\|^2$$

$$- \left(2\gamma G_g^2 + 8\beta^2 L_g^2 G_f^2\gamma\right)\frac{\sigma^2}{B},$$

which concludes the proof.

$\square$

**Lemma 8.** *For Algorithm 1, under the assumptions of Theorem 2, define the following auxiliary quantity $\hat{F}^{t+1}$:*

$$\hat{F}^{t+1} := F(\boldsymbol{\alpha}^{t+1}, \mathbf{w}^{t+1}) + s^{t+1}$$

$$- \left(\frac{1}{4\gamma} + 4L_g^2 G_f^4\gamma + \frac{\eta L^2}{2} + \frac{192NG_g^2}{\mu\eta\beta} + \frac{96NG_g^2 G_f^2\beta}{\mu^2\eta}\right)\left\|\mathbf{w}^{t+1} - \mathbf{w}^t\right\|^2$$

$$- \left(\frac{1}{8\gamma} + \frac{96NG_g^2}{\mu\eta\beta}\right)\left\|\mathbf{w}^t - \mathbf{w}^{t-1}\right\|^2 + \left(\frac{7}{2\eta} + \mu - \frac{\eta L^2}{2}\right)\left\|\boldsymbol{\alpha}^{t+1} - \boldsymbol{\alpha}^t\right\|^2$$

*where*

$$s^{t+1} := -\frac{2}{\eta^2\mu}\|\boldsymbol{\alpha}^{t+1} - \boldsymbol{\alpha}^t\|^2.$$

*Then the following statement holds:*

$$\mathbb{E}[\hat{F}^{t+1} - \hat{F}^t] \geq C_1\mathbb{E}\left\|\mathbf{w}^{t+1} - \mathbf{w}^t\right\|^2 + \frac{1}{8\gamma}\mathbb{E}\left\|\mathbf{w}^t - \mathbf{w}^{t-1}\right\|^2 + \frac{1}{8\gamma}\mathbb{E}\left\|\mathbf{w}^{t-1} - \mathbf{w}^{t-2}\right\|^2$$

$$+ C_2\mathbb{E}\left\|\boldsymbol{\alpha}^{t+1} - \boldsymbol{\alpha}^t\right\|^2$$

$$- (1-\beta)^2\left(4L_g^2 G_f^2\gamma + \eta G_g^2\right)\sum_{j=1}^N \mathbb{E}\left\|z_j^t - (f_{\widehat{T}}(\mathbf{w}^{t-1}) - f_j(\mathbf{w}^{t-1}))\right\|^2$$

$$- \left(2\gamma G_g^2 + 8\beta^2 L_g^2 G_f^2\gamma\right)\frac{\sigma^2}{B} - \left(1 - \frac{\beta}{2}\right)^2 \frac{4G_g^2}{\mu^2\eta}\sum_{j=1}^N \mathbb{E}\left\|z_j^t - z_j^{t-1}\right\|^2,$$

*where*

$$C_1 = \frac{1}{4\gamma} - \frac{L}{2} - \frac{3}{2}\eta L^2 - 16L_g^2 G_f^4\gamma - \frac{192NG_g^2 G_f^2}{\mu^2\eta\beta} - \frac{96NG_g^2 G_f^2\beta}{\mu^2\eta},$$

$$C_2 = \frac{1}{\eta} + \frac{3}{2}\mu - \frac{\eta L^2}{2} - \frac{1}{4\eta(1-\beta)^2} - \frac{2L^2}{\mu}.$$

*Proof.* According to (6) and (7):

$$\frac{1}{\eta}\left\langle \Delta^{t+1}, \boldsymbol{\alpha}^t - \boldsymbol{\alpha}^{t+1} \right\rangle \geq \left\langle \mathbf{g}_{\alpha}^t - \mathbf{g}_{\alpha}^{t-1}, \boldsymbol{\alpha}^{t+1} - \boldsymbol{\alpha}^t \right\rangle.$$

If we define $\mathbf{g}_{\alpha}(z, \boldsymbol{\alpha}) = [g(z_1), ..., g(z_N)] + 2C\mathbf{M}\boldsymbol{\alpha}$.

$$\begin{aligned}
\frac{1}{\eta}\left\langle \Delta^{t+1}, \boldsymbol{\alpha}^t - \boldsymbol{\alpha}^{t+1} \right\rangle &\geq \left\langle \mathbf{g}_{\alpha}^t - \mathbf{g}_{\alpha}(z^t, \boldsymbol{\alpha}^t), \boldsymbol{\alpha}^{t+1} - \boldsymbol{\alpha}^t \right\rangle \\
&\quad + \left\langle \mathbf{g}_{\alpha}(z^t, \boldsymbol{\alpha}^t) - \mathbf{g}_{\alpha}^{t-1}, \boldsymbol{\alpha}^{t+1} - \boldsymbol{\alpha}^t \right\rangle \\
&\geq \left\langle \mathbf{g}_{\alpha}^t - \mathbf{g}_{\alpha}(z^t, \boldsymbol{\alpha}^t), \boldsymbol{\alpha}^{t+1} - \boldsymbol{\alpha}^t \right\rangle \\
&\quad + \left\langle \mathbf{g}_{\alpha}(z^t, \boldsymbol{\alpha}^t) - \mathbf{g}_{\alpha}^{t-1}, \boldsymbol{\alpha}^{t+1} - \boldsymbol{\alpha}^t - (\boldsymbol{\alpha}^t - \boldsymbol{\alpha}^t) \right\rangle \\
&\quad + \left\langle \mathbf{g}_{\alpha}(z^t, \boldsymbol{\alpha}^t) - \mathbf{g}_{\alpha}^{t-1}, (\boldsymbol{\alpha}^t - \boldsymbol{\alpha}^{t-1}) \right\rangle \\
&\geq -\frac{G_g^2}{\mu} \sum_{j=1}^{N} \left\| z_j^{t+1} - z_j^t \right\|^2 - \frac{\mu}{4} \left\| \boldsymbol{\alpha}^{t+1} - \boldsymbol{\alpha}^t \right\|^2 \\
&\quad - \frac{\eta L^2}{2} \left\| \boldsymbol{\alpha}^t - \boldsymbol{\alpha}^{t-1} \right\|^2 - \frac{1}{2\eta} \left\| \Delta^{t+1} \right\|^2 + \mu \left\| \boldsymbol{\alpha}^t - \boldsymbol{\alpha}^{t-1} \right\|^2.
\end{aligned}$$

Since $\frac{1}{\eta}\left\langle \Delta^{t+1}, \boldsymbol{\alpha}^t - \boldsymbol{\alpha}^{t+1} \right\rangle = \frac{1}{2\eta}\left\| \boldsymbol{\alpha}^t - \boldsymbol{\alpha}^{t-1} \right\|^2 - \frac{1}{2\eta}\left\| \boldsymbol{\alpha}^{t+1} - \boldsymbol{\alpha}^t \right\|^2 - \frac{1}{2\eta}\left\| \Delta^{t+1} \right\|^2$ Hence we have:

$$\begin{aligned}
\frac{1}{2\eta}\left\| \boldsymbol{\alpha}^t - \boldsymbol{\alpha}^{t-1} \right\|^2 - \frac{1}{2\eta}\left\| \boldsymbol{\alpha}^{t+1} - \boldsymbol{\alpha}^t \right\|^2 &\geq -\frac{G_g^2}{\mu} \sum_{j=1}^{N} \left\| z_j^{t+1} - z_j^t \right\|^2 - \frac{\mu}{4}\left\| \boldsymbol{\alpha}^{t+1} - \boldsymbol{\alpha}^t \right\|^2 \\
&\quad - \frac{\eta L^2}{2}\left\| \boldsymbol{\alpha}^t - \boldsymbol{\alpha}^{t-1} \right\|^2 + \mu \left\| \boldsymbol{\alpha}^t - \boldsymbol{\alpha}^{t-1} \right\|^2
\end{aligned}$$

Multiplying both sides with $\frac{4}{\mu\eta}$ yields:

$$\begin{aligned}
\frac{2}{\mu\eta^2}\left\| \boldsymbol{\alpha}^t - \boldsymbol{\alpha}^{t-1} \right\|^2 - \frac{2}{\mu\eta^2}\left\| \boldsymbol{\alpha}^{t+1} - \boldsymbol{\alpha}^t \right\|^2 &\geq -\frac{4G_g^2}{\mu^2\eta} \sum_{j=1}^{N} \left\| z_j^{t+1} - z_j^t \right\|^2 - \frac{1}{\eta}\left\| \boldsymbol{\alpha}^{t+1} - \boldsymbol{\alpha}^t \right\|^2 \\
&\quad - \frac{2L^2}{\mu}\left\| \boldsymbol{\alpha}^t - \boldsymbol{\alpha}^{t-1} \right\|^2 + \frac{4}{\eta}\left\| \boldsymbol{\alpha}^t - \boldsymbol{\alpha}^{t-1} \right\|^2, \quad (14)
\end{aligned}$$

and so, putting all together,

$$\begin{aligned}
&\mathbb{E}[F(\boldsymbol{\alpha}^{t+1}, \mathbf{w}^{t+1}) - F(\boldsymbol{\alpha}^t, \mathbf{w}^t)] \\
&\geq \left( \frac{1}{2\gamma} - \frac{L}{2} - \frac{L^2\eta}{2} \right) \mathbb{E}\left\| \mathbf{w}^{t+1} - \mathbf{w}^t \right\|^2 - \left( 16(1-\beta)^2 L_g^2 G_f^4 \gamma + \frac{L^2\eta}{2} \right) \mathbb{E}\left\| \mathbf{w}^t - \mathbf{w}^{t-1} \right\|^2 \\
&\quad + \left( \frac{\mu}{2} - \frac{3}{2\eta} - \frac{1}{4\eta(1-\beta)^2} \right) \mathbb{E}\left\| \boldsymbol{\alpha}^{t+1} - \boldsymbol{\alpha}^t \right\|^2 + \left( \mu - \frac{1}{2\eta} - \frac{\eta L^2}{2} \right) \mathbb{E}\left\| \boldsymbol{\alpha}^t - \boldsymbol{\alpha}^{t-1} \right\|^2 \\
&\quad - \sum_{j=1}^{N} (1-\beta)^2 \left( 4\alpha^t(j) L_g^2 G_f^2 \gamma + \eta G_g^2 \right) \mathbb{E}\left\| z_j^t - (f_{\widehat{\mathcal{T}}}(\mathbf{w}^{t-1}) - f_j(\mathbf{w}^{t-1})) \right\|^2 \\
&\quad - \left( 2\gamma G_g^2 + 8\beta^2 L_g^2 G_f^2 \gamma \right) \frac{\sigma^2}{B}.
\end{aligned}$$

Now evoking Lemma 7 together with (14) yields:

$$\mathbb{E}[F(\boldsymbol{\alpha}^{t+1}, \mathbf{w}^{t+1}) + s^{t+1} - (F(\boldsymbol{\alpha}^t, \mathbf{w}^t) + s^t)]$$

$$\geq \left(\frac{1}{2\gamma} - \frac{L}{2} - \frac{L^2\eta}{2}\right) \mathbb{E}\left\|\mathbf{w}^{t+1} - \mathbf{w}^t\right\|^2 - \left(16(1-\beta)^2 L_g^2 G_f^4 \gamma + \frac{L^2\eta}{2}\right) \mathbb{E}\left\|\mathbf{w}^t - \mathbf{w}^{t-1}\right\|^2$$

$$+ \left(\frac{\mu}{2} - \frac{3}{2\eta} - \frac{1}{4\eta(1-\beta)^2} - \frac{1}{\eta}\right) \mathbb{E}\left\|\boldsymbol{\alpha}^{t+1} - \boldsymbol{\alpha}^t\right\|^2 + \left(\mu - \frac{1}{2\eta} - \frac{\eta L^2}{2} + \frac{4}{\eta} - \frac{2L^2}{\mu}\right) \mathbb{E}\left\|\boldsymbol{\alpha}^t - \boldsymbol{\alpha}^{t-1}\right\|^2$$

$$- \sum_{j=1}^{N}(1-\beta)^2 \left(4L_g^2 G_f^2 \gamma + \eta G_g^2\right) \mathbb{E}\left\|z_j^t - (f_{\widehat{\mathcal{T}}}(\mathbf{w}^{t-1}) - f_j(\mathbf{w}^{t-1}))\right\|^2$$

$$- \left(2\gamma G_g^2 + 8\beta^2 L_g^2 G_f^2 \gamma\right)\frac{\sigma^2}{B} - \frac{4G_g^2}{\mu^2\eta} \sum_{j=1}^{N} \mathbb{E}\left\|z_j^{t+1} - z_j^t\right\|^2 .$$

Now we plug in Lemma 3 with the fact that $\frac{1}{\beta} \geq 1$ and get:

$$\mathbb{E}[F(\boldsymbol{\alpha}^{t+1}, \mathbf{w}^{t+1}) + h^{t+1} - (F(\boldsymbol{\alpha}^t, \mathbf{w}^t) + h^t)]$$

$$\geq \left(\frac{1}{2\gamma} - \frac{L}{2} - \frac{L^2\eta}{2}\right) \mathbb{E}\left\|\mathbf{w}^{t+1} - \mathbf{w}^t\right\|^2$$

$$- \left(16L_g^2 G_f^4 \gamma + \frac{L^2\eta}{2} + \frac{96NG_g^2 G_f^2}{\mu^2\eta\beta} + \frac{96NG_g^2 G_f^2\beta}{\mu^2\eta}\right) \mathbb{E}\left\|\mathbf{w}^t - \mathbf{w}^{t-1}\right\|^2 - \frac{96NG_g^2 G_f^2}{\mu^2\eta\beta} \mathbb{E}\left\|\mathbf{w}^{t-1} - \mathbf{w}^{t-2}\right\|^2$$

$$+ \left(\frac{\mu}{2} - \frac{5}{2\eta} - \frac{1}{4\eta(1-\beta)^2}\right) \mathbb{E}\left\|\boldsymbol{\alpha}^{t+1} - \boldsymbol{\alpha}^t\right\|^2 + \left(\mu + \frac{7}{2\eta} - \frac{\eta L^2}{2} - \frac{2L^2}{\mu}\right) \mathbb{E}\left\|\boldsymbol{\alpha}^t - \boldsymbol{\alpha}^{t-1}\right\|^2$$

$$- (1-\beta)^2 \left(4L_g^2 G_f^2 \gamma + \eta G_g^2\right) \sum_{j=1}^{N} \mathbb{E}\left\|z_j^t - (f_{\widehat{\mathcal{T}}}(\mathbf{w}^{t-1}) - f_j(\mathbf{w}^{t-1}))\right\|^2$$

$$- \left(2\gamma G_g^2 + 8\beta^2 L_g^2 G_f^2 \gamma\right)\frac{\sigma^2}{B} - \left(1 - \frac{\beta}{2}\right)^2 \frac{4G_g^2}{\mu^2\eta} \sum_{j=1}^{N} \mathbb{E}\left\|z_j^t - z_j^{t-1}\right\|^2 .$$

Recall our definition of potential function $\hat{F}^{t+1}$

$$\hat{F}^{t+1} := F(\boldsymbol{\alpha}^{t+1}, \mathbf{w}^{t+1}) + h^{t+1} - \left(\frac{1}{4\gamma} + 16L_g^2 G_f^4 \gamma + \frac{\eta L^2}{2} + \frac{192NG_g^2 G_f^2}{\mu^2\eta\beta} + \frac{96NG_g^2 G_f^2\beta}{\mu^2\eta}\right) \left\|\mathbf{w}^{t+1} - \mathbf{w}^t\right\|^2$$

$$- \left(\frac{1}{8\gamma} + \frac{96NG_g^2 G_f^2}{\mu^2\eta\beta}\right) \left\|\mathbf{w}^t - \mathbf{w}^{t-1}\right\|^2 + \left(\frac{7}{2\eta} + \mu - \frac{\eta L^2}{2} - \frac{2L^2}{\mu}\right) \left\|\boldsymbol{\alpha}^{t+1} - \boldsymbol{\alpha}^t\right\|^2 .$$

We conclude that:

$$\mathbb{E}[\hat{F}^{t+1} - \hat{F}^t] \geq \left(\frac{1}{4\gamma} - \frac{L}{2} - \frac{3}{2}\eta L^2 - 16L_g^2 G_f^4 \gamma - \frac{192NG_g^2 G_f^2}{\mu^2\eta\beta} - \frac{96NG_g^2 G_f^2\beta}{\mu^2\eta}\right) \mathbb{E}\left\|\mathbf{w}^{t+1} - \mathbf{w}^t\right\|^2$$

$$+ \frac{1}{8\gamma} \mathbb{E}\left\|\mathbf{w}^t - \mathbf{w}^{t-1}\right\|^2 + \frac{1}{8\gamma} \mathbb{E}\left\|\mathbf{w}^{t-1} - \mathbf{w}^{t-2}\right\|^2$$

$$+ \left(\frac{1}{\eta} + \frac{3}{2}\mu - \frac{\eta L^2}{2} - \frac{1}{4\eta(1-\beta)^2} - \frac{2L^2}{\mu}\right) \mathbb{E}\left\|\boldsymbol{\alpha}^{t+1} - \boldsymbol{\alpha}^t\right\|^2$$

$$- (1-\beta)^2 \left(4L_g^2 G_f^2 \gamma + \eta G_g^2\right) \sum_{j=1}^{N} \mathbb{E}\left\|z_j^t - (f_{\widehat{\mathcal{T}}}(\mathbf{w}^{t-1}) - f_j(\mathbf{w}^{t-1}))\right\|^2$$

$$- \left(2\gamma G_g^2 + 8\beta^2 L_g^2 G_f^2 \gamma\right)\frac{\sigma^2}{B} - \left(1 - \frac{\beta}{2}\right)^2 \frac{4G_g^2}{\mu^2\eta} \sum_{j=1}^{N} \mathbb{E}\left\|z_j^t - z_j^{t-1}\right\|^2 .$$

$\square$

**Lemma 9.** *For Algorithm [1], under the assumptions of Theorem [2], define the following auxiliary quantity $\tilde{F}^{t+1}$:*

$$\tilde{F}^{t+1} := \hat{F}^{t+1} - \sum_{j=1}^{N} \mathbb{E} \left\| z_j^{t+1} - (f_{\widehat{\mathcal{T}}}(\mathbf{w}^t) - f_j(\mathbf{w}^t)) \right\|^2 - \frac{(1-\frac{\beta}{2})^2}{1-(1-\frac{\beta}{2})^2} \frac{4G_g^2}{\mu^2 \eta} \sum_{j=1}^{N} \mathbb{E} \left\| z_j^{t+1} - z_j^t \right\|^2 .$$

*Let C1, C2 be defined in Lemma [8]. If the following conditions hold*

$$\frac{1}{8\gamma} - 4(4C_1\gamma^2 L_g^2 N + C_2\eta^2 G_g^2 N^2 + N)(1-\beta)^2 G_f^2 - \frac{(1-\frac{\beta}{2})^2}{1-(1-\frac{\beta}{2})^2}\frac{4G_g^2}{\mu^2\eta}\frac{48}{\beta}G_f^2 \geq 0, \quad (15)$$

$$1 - \left(4C_1\gamma^2 L_g^2 + C_2\eta^2 G_g^2 + 4L_g^2 G_f^2\gamma + \eta G_g^2 + 1\right)(1-\beta)^2 \geq 0, \quad (16)$$

$$\frac{1}{8\gamma} - \frac{(1-\frac{\beta}{2})^2}{1-(1-\frac{\beta}{2})^2}\frac{4G_g^2}{\mu^2\eta}\frac{24}{\beta}G_f^2 \geq 0, \quad (17)$$

*then the following statement hold:*

$$\mathbb{E}[\tilde{F}^{t+1} - \tilde{F}^t] \geq \frac{C_1}{2}\gamma^2 \mathbb{E} \left\| \nabla_\mathbf{w} G(\boldsymbol{\alpha}^t, \mathbf{w}^t) \right\|^2 + \frac{C_2}{2}\eta^2 \mathbb{E} \left\| \nabla_\boldsymbol{\alpha} G(\boldsymbol{\alpha}^t, \mathbf{w}^t) \right\|^2$$
$$- \left(2C_1\gamma^2 L_g^2 N\beta^2 + 2C_2\eta^2 G_g^2 N^2\beta^2 + 4\gamma G_g^2 + 2\beta^2 L_g^2 G_f^2\gamma + 2\beta^2\right)\frac{\sigma^2}{B} .$$

*Proof.* According to Lemma [8]:

$$\mathbb{E}[\hat{F}^{t+1} - \hat{F}^t]$$

$$\geq C_1 \mathbb{E} \left\| \mathbf{w}^{t+1} - \mathbf{w}^t \right\|^2 + \frac{1}{8\gamma} \mathbb{E} \left\| \mathbf{w}^t - \mathbf{w}^{t-1} \right\|^2 + \frac{1}{8\gamma} \mathbb{E} \left\| \mathbf{w}^{t-1} - \mathbf{w}^{t-2} \right\|^2 + C_2 \mathbb{E} \left\| \boldsymbol{\alpha}^{t+1} - \boldsymbol{\alpha}^t \right\|^2$$

$$- (1-\beta)^2 \left(4L_g^2 G_f^2\gamma + \eta G_g^2\right) \sum_{j=1}^{N} \mathbb{E} \left\| z_j^t - (f_{\widehat{\mathcal{T}}}(\mathbf{w}^{t-1}) - f_j(\mathbf{w}^{t-1})) \right\|^2$$

$$- \left(2\gamma G_g^2 + 8\beta^2 L_g^2 G_f^2\gamma\right)\frac{\sigma^2}{B} - \left(1-\frac{\beta}{2}\right)^2 \frac{4G_g^2}{\mu^2\eta} \sum_{j=1}^{N} \mathbb{E} \left\| z_j^t - z_j^{t-1} \right\|^2 ,$$

where

$$C_1 = \frac{1}{4\gamma} - \frac{L}{2} - \frac{3}{2}\eta L^2 - 16L_g^2 G_f^4\gamma - \frac{192NG_g^2 G_f^2}{\mu^2\eta\beta} - \frac{96NG_g^2 G_f^2\beta}{\mu^2\eta},$$

$$C_2 = \frac{1}{\eta} + \frac{3}{2}\mu - \frac{\eta L^2}{2} - \frac{1}{4\eta(1-\beta)^2} - \frac{2L^2}{\mu}.$$

Now we plug in Lemma [5]:

$$\mathbb{E}[\hat{F}^{t+1} - \hat{F}^t]$$

$$\geq C_1 \left(\frac{1}{2}\gamma^2 \mathbb{E} \left\| \nabla_\mathbf{w} G(\boldsymbol{\alpha}^t, \mathbf{w}^t) \right\|^2 - 2\gamma^2 G_g^2\frac{\sigma^2}{B} - 4\gamma^2 G_f^2 L_g^2 \sum_{j=1}^{N} \alpha^t(j) \mathbb{E} \left\| z_j^{t+1} - (f_{\widehat{\mathcal{T}}}(\mathbf{w}^t) - \nabla f_j(\mathbf{w}^t)) \right\|^2 \right)$$

$$+ C_2 \left(\frac{1}{2}\eta^2 \mathbb{E} \left\| \nabla_\boldsymbol{\alpha} G(\boldsymbol{\alpha}^t, \mathbf{w}^t) \right\|^2 - \eta^2 G_g^2 \sum_{j=1}^{N} \mathbb{E} \left\| z_j^{t+1} - (f_{\widehat{\mathcal{T}}}(\mathbf{w}^t) - f_j(\mathbf{w}^t)) \right\|^2 \right)$$

$$- (1-\beta)^2 \left(4L_g^2 G_f^2\gamma + \eta G_g^2\right) \sum_{j=1}^{N} \mathbb{E} \left\| z_j^t - (f_{\widehat{\mathcal{T}}}(\mathbf{w}^{t-1}) - f_j(\mathbf{w}^{t-1})) \right\|^2 - \left(2\gamma G_g^2 + 8\beta^2 L_g^2 G_f^2\gamma\right)\frac{\sigma^2}{B}$$

$$- \left(1-\frac{\beta}{2}\right)^2 \frac{4G_g^2}{\mu^2\eta} \sum_{j=1}^{N} \mathbb{E} \left\| z_j^t - z_j^{t-1} \right\|^2 + \frac{1}{8\gamma} \mathbb{E} \left\| \mathbf{w}^t - \mathbf{w}^{t-1} \right\|^2 + \frac{1}{8\gamma} \mathbb{E} \left\| \mathbf{w}^{t-1} - \mathbf{w}^{t-2} \right\|^2$$

Plugging in Lemma 2 yields:

$$
\mathbb{E}[\hat{F}^{t+1} - \hat{F}^t] \geq \frac{C_1}{2}\gamma^2\,\mathbb{E}\left\|\nabla_{\mathbf{w}}G(\boldsymbol{\alpha}^t, \mathbf{w}^t)\right\|^2 + C_2\frac{1}{2}\eta^2\,\mathbb{E}\left\|\nabla_{\boldsymbol{\alpha}}G(\boldsymbol{\alpha}^t, \mathbf{w}^t)\right\|^2
$$

$$
- C_1 4\gamma^2 G_f^2 L_g^2 \sum_{j=1}^{N}\alpha^t(j)\left((1-\beta)^2\,\mathbb{E}\left\|z_j^t - (f_{\widehat{\mathcal{T}}}(\mathbf{w}^{t-1}) - f_j(\mathbf{w}^{t-1}))\right\|^2 + 4(1-\beta)^2 G_f^2\left\|\mathbf{w}^t - \mathbf{w}^{t-1}\right\|^2 + 2\beta^2\frac{\sigma^2}{B}\right)
$$

$$
- C_2\eta^2 G_g^2 \sum_{j=1}^{N}\left((1-\beta)^2\,\mathbb{E}\left\|z_j^t - (f_{\widehat{\mathcal{T}}}(\mathbf{w}^{t-1}) - f_j(\mathbf{w}^{t-1}))\right\|^2 + 4(1-\beta)^2 G_f^2\left\|\mathbf{w}^t - \mathbf{w}^{t-1}\right\|^2 + 2\beta^2\frac{\sigma^2}{B}\right)
$$

$$
- (1-\beta)^2\left(4L_g^2 G_f^2\gamma + \eta G_g^2\right)\sum_{j=1}^{N}\mathbb{E}\left\|z_j^t - (f_{\widehat{\mathcal{T}}}(\mathbf{w}^{t-1}) - f_j(\mathbf{w}^{t-1}))\right\|^2
$$

$$
- \left(2\gamma G_g^2 + 8\beta^2 L_g^2 G_f^2\gamma + 2C_1\gamma^2 G_g^2\right)\frac{\sigma^2}{B}
$$

$$
- \left(1 - \frac{\beta}{2}\right)^2\frac{4G_g^2}{\mu^2\eta}\sum_{j=1}^{N}\mathbb{E}\left\|z_j^t - z_j^{t-1}\right\|^2 + \frac{1}{8\gamma}\,\mathbb{E}\left\|\mathbf{w}^t - \mathbf{w}^{t-1}\right\|^2 + \frac{1}{8\gamma}\,\mathbb{E}\left\|\mathbf{w}^{t-1} - \mathbf{w}^{t-2}\right\|^2.
$$

Rearranging the terms yields:

$$
\mathbb{E}[\hat{F}^{t+1} - \hat{F}^t]
$$

$$
\geq \frac{C_1}{2}\gamma^2\,\mathbb{E}\left\|\nabla_{\mathbf{w}}G(\boldsymbol{\alpha}^t, \mathbf{w}^t)\right\|^2 + \frac{C_2}{2}\eta^2\,\mathbb{E}\left\|\nabla_{\boldsymbol{\alpha}}G(\boldsymbol{\alpha}^t, \mathbf{w}^t)\right\|^2
$$

$$
+ \left(\frac{1}{8\gamma} - (4C_1\gamma^2 L_g^2 + C_2\eta^2 G_g^2 N)4(1-\beta)^2 G_f^2\right)\mathbb{E}\left\|\mathbf{w}^t - \mathbf{w}^{t-1}\right\|^2
$$

$$
- \left(4C_1\gamma^2 L_g^2 + C_2\eta^2 G_g^2 + 4L_g^2 G_f^2\gamma + \eta G_g^2\right)(1-\beta)^2\sum_{j=1}^{N}\mathbb{E}\left\|z_j^t - (f_{\widehat{\mathcal{T}}}(\mathbf{w}^{t-1}) - f_j(\mathbf{w}^{t-1}))\right\|^2
$$

$$
+ \frac{1}{8\gamma}\,\mathbb{E}\left\|\mathbf{w}^{t-1} - \mathbf{w}^{t-2}\right\|^2 - \left(8C_1\gamma^2 G_f^2 L_g^2\beta^2 + 2C_2\eta^2 G_g^2\beta^2 + 2\gamma G_g^2 + 8\gamma G_f^2 L_g^2\beta^2 + 2C_1\gamma^2 G_g^2\right)\frac{\sigma^2}{B}
$$

$$
- \left(1 - \frac{\beta}{2}\right)^2\frac{4G_g^2}{\mu^2\eta}\sum_{j=1}^{N}\mathbb{E}\left\|z_j^t - z_j^{t-1}\right\|^2.
$$

Recall our definition of potential function $\tilde{F}^{t+1}$:

$$
\tilde{F}^{t+1} := \hat{F}^{t+1} - \sum_{j=1}^{N}\mathbb{E}\left\|z_j^{t+1} - (f_{\widehat{\mathcal{T}}}(\mathbf{w}^t) - f_j(\mathbf{w}^t))\right\|^2 - \frac{(1-\frac{\beta}{2})^2}{1-(1-\frac{\beta}{2})^2}\frac{4G_g^2}{\mu^2\eta}\sum_{j=1}^{N}\mathbb{E}\left\|z_j^{t+1} - z_j^t\right\|^2
$$

Hence we have:

$$\mathbb{E}[\tilde{F}^{t+1} - \tilde{F}^t]$$

$$\geq \frac{C_1}{2}\gamma^2 \,\mathbb{E}\left\|\nabla_{\mathbf{w}}G(\boldsymbol{\alpha}^t, \mathbf{w}^t)\right\|^2 + \frac{C_2}{2}\eta^2 \,\mathbb{E}\left\|\nabla_{\boldsymbol{\alpha}}G(\boldsymbol{\alpha}^t, \mathbf{w}^t)\right\|^2$$

$$+ \left(\frac{1}{8\gamma} - 4(4C_1\gamma^2 L_g^2 N + C_2\eta^2 G_g^2 N^2)(1-\beta)^2 G_f^2\right)\mathbb{E}\left\|\mathbf{w}^t - \mathbf{w}^{t-1}\right\|^2$$

$$- \sum_{j=1}^{N}\mathbb{E}\left\|z_j^{t+1} - (f_{\widehat{\mathcal{T}}}(\mathbf{w}^t) - f_j(\mathbf{w}^t))\right\|^2 + \sum_{j=1}^{N}\mathbb{E}\left\|z_j^{t} - (f_{\widehat{\mathcal{T}}}(\mathbf{w}^{t-1}) - f_j(\mathbf{w}^{t-1}))\right\|^2$$

$$- \left(4C_1\gamma^2 L_g^2 + C_2\eta^2 G_g^2 + 4L_g^2 G_f^2\gamma + \eta G_g^2\right)(1-\beta)^2 \sum_{j=1}^{N}\mathbb{E}\left\|z_j^{t} - (f_{\widehat{\mathcal{T}}}(\mathbf{w}^{t-1}) - f_j(\mathbf{w}^{t-1}))\right\|^2$$

$$+ \frac{1}{8\gamma}\mathbb{E}\left\|\mathbf{w}^{t-1} - \mathbf{w}^{t-2}\right\|^2 - \left(8C_1\gamma^2 G_f^2 L_g^2\beta^2 + 2C_2\eta^2 G_g^2\beta^2 + 2\gamma G_g^2 + 8\gamma G_f^2 L_g^2\beta^2 + 2C_1\gamma^2 G_g^2\right)\frac{\sigma^2}{B}$$

$$- \left(1 - \frac{\beta}{2}\right)^2 \frac{4G_g^2}{\mu^2\eta}\sum_{j=1}^{N}\mathbb{E}\left\|z_j^{t} - z_j^{t-1}\right\|^2$$

$$- \frac{(1-\frac{\beta}{2})^2}{1-(1-\frac{\beta}{2})^2}\frac{4G_g^2}{\mu^2\eta}\sum_{j=1}^{N}\mathbb{E}\left\|z_j^{t+1} - z_j^{t}\right\|^2 + \frac{(1-\frac{\beta}{2})^2}{1-(1-\frac{\beta}{2})^2}\frac{4G_g^2}{\mu^2\eta}\sum_{j=1}^{N}\mathbb{E}\left\|z_j^{t} - z_j^{t-1}\right\|^2.$$

Now we plug in Lemma 2 and 3:

$$\mathbb{E}[\tilde{F}^{t+1} - \tilde{F}^t]$$

$$\geq \frac{C_1}{2}\gamma^2 \,\mathbb{E}\left\|\nabla_{\mathbf{w}}G(\boldsymbol{\alpha}^t, \mathbf{w}^t)\right\|^2 + \frac{C_2}{2}\eta^2 \,\mathbb{E}\left\|\nabla_{\boldsymbol{\alpha}}G(\boldsymbol{\alpha}^t, \mathbf{w}^t)\right\|^2$$

$$+ \left(\frac{1}{8\gamma} - 4(4C_1\gamma^2 L_g^2 N + C_2\eta^2 G_g^2 N^2 + N)(1-\beta)^2 G_f^2 - \frac{(1-\frac{\beta}{2})^2}{1-(1-\frac{\beta}{2})^2}\frac{4G_g^2}{\mu^2\eta}\frac{48}{\beta}G_f^2\right)$$

$$\times \mathbb{E}\left\|\mathbf{w}^t - \mathbf{w}^{t-1}\right\|^2$$

$$+ \left(1 - \left(4C_1\gamma^2 L_g^2 + C_2\eta^2 G_g^2 + 4L_g^2 G_f^2\gamma + \eta G_g^2 + 1\right)(1-\beta)^2\right)$$

$$\times \sum_{j=1}^{N}\mathbb{E}\left\|z_j^{t} - (f_{\widehat{\mathcal{T}}}(\mathbf{w}^{t-1}) - f_j(\mathbf{w}^{t-1}))\right\|^2$$

$$+ \left(\frac{1}{8\gamma} - \frac{(1-\frac{\beta}{2})^2}{1-(1-\frac{\beta}{2})^2}\frac{4G_g^2}{\mu^2\eta}\frac{24}{\beta}G_f^2\right)\mathbb{E}\left\|\mathbf{w}^{t-1} - \mathbf{w}^{t-2}\right\|^2$$

$$- \left(8C_1\gamma^2 G_f^2 L_g^2\beta^2 + 2C_2\eta^2 G_g^2\beta^2 + 2\gamma G_g^2 + 8\gamma G_f^2 L_g^2\beta^2 + 2C_1\gamma^2 G_g^2 + 2\beta^2 N\right)\frac{\sigma^2}{B}$$

$$+ (1-\frac{\beta}{2})^2\frac{4G_g^2}{\mu^2\eta}\underbrace{\left(\frac{1}{1-(1-\frac{\beta}{2})^2} - \frac{(1-\frac{\beta}{2})^2}{1-(1-\frac{\beta}{2})^2} - 1\right)}_{=0}\sum_{j=1}^{N}\mathbb{E}\left\|z_j^{t} - z_j^{t-1}\right\|^2.$$

By our assumption in (15)-(17):

$$\frac{1}{8\gamma} - 4(4C_1\gamma^2 L_g^2 N + C_2\eta^2 G_g^2 N^2 + N)(1-\beta)^2 G_f^2 - \frac{(1-\frac{\beta}{2})^2}{1-(1-\frac{\beta}{2})^2}\frac{4G_g^2}{\mu^2\eta}\frac{48}{\beta}G_f^2 \geq 0,$$

$$1 - \left(4C_1\gamma^2 L_g^2 + C_2\eta^2 G_g^2 + 4L_g^2 G_f^2\gamma + \eta G_g^2 + 1\right)(1-\beta)^2 \geq 0,$$

$$\frac{1}{8\gamma} - \frac{(1-\frac{\beta}{2})^2}{1-(1-\frac{\beta}{2})^2}\frac{4G_g^2}{\mu^2\eta}\frac{24}{\beta}G_f^2 \geq 0,$$

then we can have the 'clean' bound:

$$\mathbb{E}[\tilde{F}^{t+1} - \tilde{F}^t] \geq \frac{C_1}{2}\gamma^2 \, \mathbb{E}\left\|\nabla_{\mathbf{w}}G(\boldsymbol{\alpha}^t, \mathbf{w}^t)\right\|^2 + \frac{C_2}{2}\eta^2 \, \mathbb{E}\left\|\nabla_{\boldsymbol{\alpha}}G(\boldsymbol{\alpha}^t, \mathbf{w}^t)\right\|^2$$
$$- \left(2C_1\gamma^2 L_g^2 N\beta^2 + 2C_2\eta^2 G_g^2 N^2\beta^2 + 4\gamma G_g^2 + 2\beta^2 L_g^2 G_f^2\gamma + 2\beta^2\right)\frac{\sigma^2}{B} \, .$$

□

## A.3 Proof of theorem 2

Summing the inequality from $t = 1$ to $T$ yields:

$$\frac{\mathbb{E}[\tilde{F}^T - \tilde{F}^0]}{T} + \left(2C_1\gamma^2 L_g^2 N\beta^2 + 2C_2\eta^2 G_g^2 N^2\beta^2 + 4\gamma G_g^2 + 2\beta^2 L_g^2 G_f^2\gamma + 2\beta^2\right)\frac{\sigma^2}{B}$$
$$\geq \frac{1}{T}\sum_{t=0}^{T-1} \frac{C_1}{2}\gamma^2 \, \mathbb{E}\left\|\nabla_{\mathbf{w}}G(\boldsymbol{\alpha}^t, \mathbf{w}^t)\right\|^2 + \frac{C_2}{2}\eta^2 \, \mathbb{E}\left\|\nabla_{\boldsymbol{\alpha}}G(\boldsymbol{\alpha}^t, \mathbf{w}^t)\right\|^2$$
$$\geq \min\left\{\frac{C_1}{2}\gamma^2, \frac{C_2}{2}\eta^2\right\}\frac{1}{T}\sum_{t=0}^{T-1} \mathbb{E}\left\|\nabla_{\mathbf{w}}G(\boldsymbol{\alpha}^t, \mathbf{w}^t)\right\|^2 + \mathbb{E}\left\|\nabla_{\boldsymbol{\alpha}}G(\boldsymbol{\alpha}^t, \mathbf{w}^t)\right\|^2$$

We compute the upper and lower bound of $C_1$ and $C_2$. For $C_1$

$$C_1 = \frac{1}{4\gamma} - \frac{L}{2} - \frac{3}{2}\eta L^2 - 16L_g^2 G_f^4\gamma - \frac{192NG_g^2 G_f^2}{\mu^2\eta\beta} - \frac{96NG_g^2 G_f^2\beta}{\mu^2\eta}$$

The upper bound $C_1 \leq \frac{1}{4\gamma}$ holds trivially. For lower bound, since we choose

$$\gamma = \min\left\{\frac{1}{20L}, \frac{1}{60\eta L^2}, \frac{1}{8\sqrt{10}L_g G_f^2}, \frac{\mu^2\eta\beta}{7680NG_g^2 G_f^2}\right\}$$

we know that $C_1 \geq \frac{1}{8\gamma}$.

For $C_2$:

$$C_2 = \frac{1}{\eta} + \frac{3}{2}\mu - \frac{\eta L^2}{2} - \frac{1}{4\eta(1-\beta)^2} - \frac{2L^2}{\mu}.$$

The upper bound $C_2 \leq \frac{1}{\eta} + \frac{3\mu}{2}$ holds trivially. For lower bound, since we choose:

$$\eta = \min\left\{\frac{\sqrt{2}}{3L}, \frac{\mu}{36L^2}\right\}, \beta = 0.1 \leq 1 - \frac{\sqrt{3}}{2}$$

it holds that $C_2 \geq \frac{1}{2\eta}$.

Since $\frac{1}{8\gamma} \leq C_1 \leq \frac{1}{4\gamma}$ and $\frac{1}{2\eta} \leq C_2 \leq \frac{1}{\eta} + \frac{3\mu}{2} \leq \frac{2}{\eta}$, we have:

$$\frac{\mathbb{E}[\tilde{F}^T - \tilde{F}^0]}{T} + \left(\frac{1}{2}\gamma L_g^2 N\beta^2 + \frac{4}{\eta}\eta^2 G_g^2 N^2\beta^2 + 4\gamma G_g^2 + 2\beta^2 L_g^2 G_f^2\gamma + 2\beta^2\right)\frac{\sigma^2}{B} \quad (18)$$

$$\geq \min\left\{\frac{1}{16}\gamma, \frac{1}{4}\eta\right\}\frac{1}{T}\sum_{t=0}^{T-1} \mathbb{E}\left\|\nabla_{\mathbf{w}}G(\boldsymbol{\alpha}^t, \mathbf{w}^t)\right\|^2 + \mathbb{E}\left\|\nabla_{\boldsymbol{\alpha}}G(\boldsymbol{\alpha}^t, \mathbf{w}^t)\right\|^2 \quad (19)$$

We then need to verify the conditions (15), (16) and (17) in Lemma 9 can hold under our choice of $\eta_{\mathbf{x}}$, $\eta_{\mathbf{y}}$ and $\beta$. To guarantee (15) holding, we need:

$$\gamma \leq \min\left\{\sqrt{\frac{1}{128L_g^2 N}}, \frac{1}{52G_f^2\left(2\eta G_g^2 N^2 + N\right)}, \frac{\mu^2\eta}{307200G_g^2 G_f^2}\right\}.$$

To guarantee (16) holding, we need:

$$\gamma \le \min\left\{\frac{1}{50L_g^2}, \frac{1}{200L_g^2 G_f^2}\right\}, \eta \le \frac{1}{100G_g^2}.$$

To guarantee (17) holding, we need:

$$\gamma \le \frac{\mu^2 \eta}{1080 G_g^2 G_f^2}$$

Next we examine how large the $\mathbb{E}[\tilde{F}^T - \tilde{F}^0]$ is. By definition of potential function, we have:

$$\mathbb{E}[\tilde{F}^T - \tilde{F}^0]$$

$$= \hat{F}^T - \sum_{j=1}^N \mathbb{E}\left\|z_j^T - (f_{\hat{\mathcal{T}}}(\mathbf{w}^{T-1}) - f_j(\mathbf{w}^{T-1}))\right\|^2 - \frac{(1-\frac{\beta}{2})^2}{1-(1-\frac{\beta}{2})^2}\frac{4G_g^2}{\mu^2\eta}\sum_{j=1}^N \mathbb{E}\left\|z_j^T - z_j^{T-1}\right\|^2$$

$$- \left(\hat{F}^0 - \sum_{j=1}^N \mathbb{E}\left\|z_j^0 - (f_{\hat{\mathcal{T}}}(\mathbf{w}^{-1}) - f_j(\mathbf{w}^{-1}))\right\|^2 - \frac{(1-\frac{\beta}{2})^2}{1-(1-\frac{\beta}{2})^2}\frac{4G_g^2}{\mu^2\eta}\sum_{j=1}^N \mathbb{E}\left\|z_j^0 - z_j^{-1}\right\|^2\right)$$

$$\le \hat{F}^T - \hat{F}^0 + \sum_{j=1}^N \mathbb{E}\left\|z_j^0 - (f_{\hat{\mathcal{T}}}(\mathbf{w}^{-1}) - f_j(\mathbf{w}^{-1}))\right\|^2 + \frac{(1-\frac{\beta}{2})^2}{\beta - \frac{\beta^2}{4}}\frac{4G_g^2}{\mu^2\eta}\sum_{j=1}^N \mathbb{E}\left\|z_j^0 - z_j^{-1}\right\|^2.$$

By our choice $z_j^0 = z_j^{-1} = f_{\hat{\mathcal{T}}}(\mathbf{w}^{-1}) - f_j(\mathbf{w}^{-1})$, $\mathbf{w}^0 = \mathbf{w}^{-1}$, we have

$$\mathbb{E}[\tilde{F}^T - \tilde{F}^0] \le \hat{F}^T - \hat{F}^0$$

Next we examine how large the $\mathbb{E}[\hat{F}^T - \hat{F}^0]$ is.

$$\mathbb{E}[\hat{F}^T - \hat{F}^0]$$

$$= F(\boldsymbol{\alpha}^T, \mathbf{w}^T) + s^T - \left(\frac{1}{4\gamma} + 4L_g^2 G_f^4 \gamma + \frac{\eta L^2}{2} + \frac{192NG_g^2}{\mu^2\eta\beta} + \frac{96NG_g^2 G_f^2 \beta}{\mu^2\eta}\right)\left\|\mathbf{w}^T - \mathbf{w}^{T-1}\right\|^2$$

$$- \left(\frac{1}{8\gamma} + \frac{96NG_g^2}{\mu^2\eta\beta}\right)\left\|\mathbf{w}^{T-1} - \mathbf{w}^{T-2}\right\|^2 + \left(\frac{7}{2\eta} + \mu - \frac{\eta L^2}{2}\right)\left\|\boldsymbol{\alpha}^T - \boldsymbol{\alpha}^{T-1}\right\|^2$$

$$- F(\boldsymbol{\alpha}^0, \mathbf{w}^0) - s^0 + \left(\frac{1}{4\gamma} + 4L_g^2 G_f^4 \gamma + \frac{\eta L^2}{2} + \frac{192NG_g^2}{\mu^2\eta\beta} + \frac{96NG_g^2 G_f^2 \beta}{\mu^2\eta}\right)\left\|\mathbf{w}^0 - \mathbf{w}^{-1}\right\|^2$$

$$+ \left(\frac{1}{8\gamma} + \frac{96NG_g^2}{\mu^2\eta\beta}\right)\left\|\mathbf{w}^{-1} - \mathbf{w}^{-2}\right\|^2 - \left(\frac{7}{2\eta} + \mu - \frac{\eta L^2}{2}\right)\left\|\boldsymbol{\alpha}^0 - \boldsymbol{\alpha}^{-1}\right\|^2$$

$$\le F_{\max} + \left(\frac{7}{2\eta} + \mu - \frac{\eta L^2}{2}\right)\left\|\boldsymbol{\alpha}^T - \boldsymbol{\alpha}^{T-1}\right\|^2$$

Notice that $\boldsymbol{\alpha}^{-1} = \boldsymbol{\alpha}^0$, $\mathbf{w}^0 = \mathbf{w}^{-1} = \mathbf{w}^{-2}$ we have

$$\mathbb{E}[\hat{F}^T - \hat{F}^0] \le F_{\max} + \left(\frac{7}{2\eta} + \mu - \frac{\eta L^2}{2}\right)\left\|\boldsymbol{\alpha}^T - \boldsymbol{\alpha}^{T-1}\right\|^2$$

.

According to updating rule,

$$\mathbb{E}\left\|\boldsymbol{\alpha}^T - \boldsymbol{\alpha}^{T-1}\right\|^2 = \mathbb{E}\left\|\eta \mathbf{g}_{\boldsymbol{\alpha}}^{T-1}\right\|^2 \le 2\eta^2 \mathbb{E}\left\|\nabla_{\boldsymbol{\alpha}} F(\boldsymbol{\alpha}^{t-1}, \mathbf{w}^{t-1})\right\|^2 + 2\eta^2 \sum_{j=1}^N G_g^2 \left\|z_j^t - (f_{\hat{\mathcal{T}}}(\mathbf{w}^{t-1}) - f_j(\mathbf{w}^{t-1}))\right\|^2$$

$$\le 2\eta^2 (2NB_g^2 + 2L^2) + 2\eta^2 NG_g^2 \left(4\frac{(1-\beta)^4}{1-(1-\beta)^2}\gamma^2 G_f^4 G_g^2 + 2\beta^2\frac{(1-\beta)^2}{1-(1-\beta)^2}\frac{\sigma^2}{B}\right)$$

$$\le 4\eta^2 (NB_g^2 + L^2) + 2\eta^2 NG_g^2 \left(14\gamma^2 G_f^4 G_g^2 + 2\frac{\sigma^2}{B}\right)$$

Putting pieces together we can have the upper bound of $\mathbb{E}[\tilde{F}^{T+1} - \tilde{F}^1]$:

$$\mathbb{E}[\tilde{F}^{T+1} - \tilde{F}^1] \le F_{\max} + \frac{9}{2\eta}\left(4\eta^2 (NB_g^2 + L^2) + 2\eta^2 NG_g^2\left(14\gamma^2 G_f^4 G_g^2 + 2\frac{\sigma^2}{B}\right)\right)$$

Plugging above bound back to (19) yields:

$$\frac{1}{T}\sum_{t=1}^T \left\|\nabla G(\boldsymbol{\alpha}^t, \mathbf{w}^t)\right\|^2$$

$$\le \max\left\{\frac{16}{\gamma}, \frac{4}{\eta}\right\} \cdot \frac{F_{\max} + \frac{9}{2}\left(4\eta(NB_g^2 + L^2) + 2\eta NG_g^2\left(14\gamma^2 G_f^4 G_g^2 + 2\frac{\sigma^2}{B}\right)\right)}{T}$$

$$+ \max\left\{\frac{16}{\gamma}, \frac{4}{\eta}\right\} \cdot \left(\frac{1}{2}\gamma L_g^2 N\beta^2 + 4\eta G_g^2 N^2 \beta^2 + 4\gamma G_g^2 + 2\beta^2 L_g^2 G_f^2 \gamma + 2\beta^2\right)\frac{\sigma^2}{B}$$

$$\le O\left(\frac{1}{\eta} \cdot \frac{F_{\max} + \left(\eta(NB_g^2 + L^2) + \eta NG_g^2\left(\gamma^2 G_f^4 G_g^2 + \frac{\sigma^2}{B}\right)\right)}{T}\right)$$

$$+ O\left(\frac{1}{\eta} \cdot \left(\gamma L_g^2 N\beta^2 + \eta G_g^2 N^2 \beta^2 + \gamma G_g^2 + \beta^2 L_g^2 G_f^2 \gamma + 2\beta^2\right)\frac{\sigma^2}{B}\right)$$

Define $\kappa = L/\mu$. Recall that we choose:

$$\eta = \Theta\left(\frac{1}{\kappa L}\right), \gamma = \Theta\left(\min\left\{\frac{1}{L}, \frac{1}{L_g G_f^2}, \frac{\mu^2}{NG_g^2 G_f^2 \kappa L}\right\}\right)$$

To guarantee RHS is less than $\epsilon^2$, we need:

$$T = \Omega\left(\max\left\{\frac{F_{\max}}{\eta\epsilon^2}, \frac{NB_g^2}{\epsilon^2}\right\}\right), B = \Theta\left(\max\left\{\frac{\mu^2 L_g^2 \sigma^2}{G_g^2 G_f^2 \epsilon^2}, \frac{G_g^2 N\sigma^2}{\epsilon^2}, \frac{\mu^2}{NG_f^2 \epsilon^2}, \frac{\mu^2 L_g^2 \sigma^2}{NG_g^2 \epsilon^2}, \frac{\kappa L\sigma^2}{\epsilon^2}\right\}\right),$$

which yields the total gradient complexity:

$$O\left(\frac{\kappa L F_{\max}}{\epsilon^2} \cdot \max\left\{\frac{\kappa L\sigma^2}{\epsilon^2}, \frac{\mu^2 L_g^2 \sigma^2}{G_g^2 G_f^2 \epsilon^2}, \frac{G_g^2 N\sigma^2}{\epsilon^2}, 1\right\}\right).$$

# B Proofs for $\mathbf{w}^*$ Approximation Algorithm

## B.1 Proof of Lemma 1

*Proof.* First, we define $\Phi(\boldsymbol{\alpha}, \mathbf{w}) = \sum_{j=1}^{N} \alpha(j) \cdot f_j(\mathbf{w})$. Indeed, $\Phi(\cdot, \mathbf{w})$ is $\sqrt{N} G_f$ smooth for all $\mathbf{w} \in \mathcal{W}$. To see this, we consider

$$\|\nabla_{\mathbf{w}} \Phi(\boldsymbol{\alpha}, \mathbf{w}) - \nabla_{\mathbf{w}} \Phi(\boldsymbol{\alpha}', \mathbf{w})\| = \left\| \sum_{i=1}^{N} \alpha_i \nabla f_i(\mathbf{w}) - \sum_{i=1}^{N} \alpha_i' \nabla f_i(\mathbf{w}) \right\|$$

$$= \left\| \sum_{i=1}^{N} (\alpha_i - \alpha_i') \nabla f_i(\mathbf{w}) \right\|$$

$$\leq \sum_{i=1}^{N} |\alpha_i - \alpha_i'| \max_{i \in [N]} \|\nabla f_i(\mathbf{w})\|$$

$$\leq \|\boldsymbol{\alpha} - \boldsymbol{\alpha}'\|_1 G_f$$

$$\leq \sqrt{N} G_f \|\boldsymbol{\alpha} - \boldsymbol{\alpha}'\|.$$

According to optimality conditions we have:

$$\langle \mathbf{w} - \mathbf{w}^*(\boldsymbol{\alpha}), \nabla_{\mathbf{w}} \Phi(\boldsymbol{\alpha}, \mathbf{w}^*(\boldsymbol{\alpha})) \rangle \geq 0,$$
$$\langle \mathbf{w} - \mathbf{w}^*(\boldsymbol{\alpha}'), \nabla_{\mathbf{w}} \Phi(\boldsymbol{\alpha}', \mathbf{w}^*(\boldsymbol{\alpha}')) \rangle \geq 0$$

Substituting $\mathbf{v}$ with $\mathbf{w}^*(\boldsymbol{\alpha}')$ and $\mathbf{w}^*(\boldsymbol{\alpha})$ in the above first and second inequalities respectively yields:

$$\langle \mathbf{w}^*(\boldsymbol{\alpha}') - \mathbf{w}^*(\boldsymbol{\alpha}), \nabla_{\mathbf{w}} \Phi(\boldsymbol{\alpha}, \mathbf{w}^*(\boldsymbol{\alpha})) \rangle \geq 0,$$
$$\langle \mathbf{w}^*(\boldsymbol{\alpha}) - \mathbf{w}^*(\boldsymbol{\alpha}'), \nabla_{\mathbf{w}} \Phi(\boldsymbol{\alpha}', \mathbf{w}^*(\boldsymbol{\alpha}')) \rangle \geq 0$$

Adding up the above two inequalities yields:

$$\langle \mathbf{w}^*(\boldsymbol{\alpha}') - \mathbf{w}^*(\boldsymbol{\alpha}), \nabla_{\mathbf{w}} \Phi(\boldsymbol{\alpha}, \mathbf{w}^*(\boldsymbol{\alpha})) - \nabla_{\mathbf{w}} \Phi(\boldsymbol{\alpha}', \mathbf{w}^*(\boldsymbol{\alpha}')) \rangle \geq 0, \tag{20}$$

Since $\Phi(\boldsymbol{\alpha}, \cdot)$ is $\mu$ strongly convex, we have:

$$\langle \mathbf{w}^*(\boldsymbol{\alpha}') - \mathbf{w}^*(\boldsymbol{\alpha}), \nabla_{\mathbf{w}} \Phi(\boldsymbol{\alpha}, \mathbf{w}^*(\boldsymbol{\alpha}')) - \nabla_{\mathbf{w}} \Phi(\boldsymbol{\alpha}, \mathbf{w}^*(\boldsymbol{\alpha})) \geq \mu \|\mathbf{v}^*(\boldsymbol{\alpha}') - \mathbf{w}^*(\boldsymbol{\alpha})\|^2. \tag{21}$$

Adding up (20) and (21) yields:

$$\langle \mathbf{w}^*(\boldsymbol{\alpha}') - \mathbf{w}^*(\boldsymbol{\alpha}), \nabla_{\mathbf{w}} \Phi(\boldsymbol{\alpha}, \mathbf{w}^*(\boldsymbol{\alpha}')) - \nabla_{\mathbf{w}} \Phi(\boldsymbol{\alpha}', \mathbf{w}^*(\boldsymbol{\alpha}')) \geq \mu \|\mathbf{w}^*(\boldsymbol{\alpha}') - \mathbf{w}^*(\boldsymbol{\alpha})\|^2$$

Finally, using $\sqrt{N} G_f$ smoothness of $\Phi$ will conclude the proof:

$$\sqrt{N} G_f \|\mathbf{w}^*(\boldsymbol{\alpha}') - \mathbf{w}^*(\boldsymbol{\alpha})\| \|\boldsymbol{\alpha} - \boldsymbol{\alpha}'\| \geq \mu \|\mathbf{w}^*(\boldsymbol{\alpha}') - \mathbf{w}^*(\boldsymbol{\alpha})\|^2$$

$$\iff \kappa^* \|\boldsymbol{\alpha} - \boldsymbol{\alpha}'\| \geq \|\mathbf{w}^*(\boldsymbol{\alpha}') - \mathbf{w}^*(\boldsymbol{\alpha})\|.$$

$\square$

## B.2 Preliminaries of the proof of theorem 3

In this section, we introduce some notational preliminaries for the proof for Theorem 3. Following the framework in [21], we introduce three models, neural network predictor, random feature predictor and kernel least square predictor. In the following sections we will use $w(j)$ to denote the $j$th coordinate of vector $\mathbf{w}$.

**Two-layer ReLU neural network (NN) predictor**    Recall that in Section 3.1 we consider a two layer vector-valued neural network $\mathbf{h}_{\boldsymbol{\theta}} : \mathbb{R}^N \to \mathbb{R}^d$:

$$\mathbf{h}_{\boldsymbol{\theta}}(\mathbf{x}) = \left[ \mathbf{a}_1^\top (\mathbf{U}^1 \mathbf{x})_+, ..., \mathbf{a}_d^\top (\mathbf{U}^d \mathbf{x})_+ \right],$$

where parameters of the *hidden layer* are matrices $\mathbf{U}^i \in \mathbb{R}^{m \times N}$, collectively captured by the parameter vector $\boldsymbol{\theta} = (\text{vec}(\mathbf{U}^1), \dots, \text{vec}(\mathbf{U}^d)) \in \mathbb{R}^{dmN}$, and we also define $\boldsymbol{\theta}^j = \text{vec}(\mathbf{U}^j) \in \mathbb{R}^{Nm}$. Here $\mathbf{a}_i \in \{\pm 1/\sqrt{m}\}^m$ are parameters of the *output layer*.

Next, induce the Neural Tangent Feature (NTF) operator, defined as

$$\phi_t^j = \begin{pmatrix} a_j(1)\mathbb{I}\{\mathbf{u}_t^{j\top}(1)\boldsymbol{\alpha}\}\boldsymbol{\alpha} \\ \vdots \\ a_j(m)\mathbb{I}\{\mathbf{u}_t^{j\top}(m)\boldsymbol{\alpha}\}\boldsymbol{\alpha} \end{pmatrix} \in \mathbb{R}^{dm} \qquad (j,t \in \mathbb{N})$$

where $\mathbf{u}_t^{j\top}(r)$ is $r$th row of $\mathbf{U}_t^j$. Note that we dropped dependence on $\boldsymbol{\alpha}$ on the left hand side — it is always assumed that the operator is applied to variable $\boldsymbol{\alpha}$. Also, note that the operator comes by differentiation of the neural network with respect to a $j$-th component of the hidden layer, that is $\phi_t^j(\boldsymbol{\alpha}) = \nabla_{\boldsymbol{\theta}^j}\mathbf{h}_{\boldsymbol{\theta}}(\boldsymbol{\alpha})|_{\boldsymbol{\theta}^j=\boldsymbol{\theta}_t^j}$. For convenience, we also introduce the following NTF matrix notation, given multiple input vectors $\boldsymbol{\alpha}_1, \ldots, \boldsymbol{\alpha}_n$:

$$\boldsymbol{\Phi}_t^j = \left[\phi_t^j(\boldsymbol{\alpha}_1), \ldots, \phi_t^j(\boldsymbol{\alpha}_n)\right] \in \mathbb{R}^{dm \times n} .$$

Finally, it is not hard to see that the use of NTF operator recovers the neural network predictor:

$$\mathbf{h}_t(\boldsymbol{\alpha}) = \begin{bmatrix} \phi_t^1(\boldsymbol{\alpha})^\top \boldsymbol{\theta}_t^1 \\ \vdots \\ \phi_t^d(\boldsymbol{\alpha})^\top \boldsymbol{\theta}_t^d \end{bmatrix}$$

Using the above definitions, we can write least squares objective as

$$\widehat{\mathcal{R}}(\boldsymbol{\theta}_t) = \frac{1}{n}\sum_{i=1}^n (\mathbf{h}_t(\boldsymbol{\alpha}_i) - \mathbf{w}^*(\boldsymbol{\alpha}_i))^2 = \frac{1}{n}\sum_{i=1}^n\sum_{j=1}^d \left(\phi_t^{j\top}(\boldsymbol{\alpha}_i)\boldsymbol{\theta}_t^j - w^*(\boldsymbol{\alpha}_i)(j)\right)^2 ,$$

and similarly for the pointwise loss function,

$$\widehat{\mathcal{R}}^j(\boldsymbol{\theta}_t^j) = \frac{1}{n}\sum_{i=1}^n \left(h_t^j(\boldsymbol{\alpha}_i) - w^*(\boldsymbol{\alpha}_i)(j)\right)^2 = \frac{1}{n}\sum_{i=1}^n \left(\phi_t^{j\top}(\boldsymbol{\alpha}_i)\boldsymbol{\theta}_t^j - w^*(\boldsymbol{\alpha}_i)(j)\right)^2 ,$$

and for the GD update rule:

$$\boldsymbol{\theta}_{t+1}^j = \boldsymbol{\theta}_t^j - \eta \mathbf{g}_t^j, \text{where } \mathbf{g}_t^j = \frac{1}{n}\sum_{i=1}^n \phi_t^j(\boldsymbol{\alpha}_i)(h_t^j(\boldsymbol{\alpha}_i) - w_i^t(j)) .$$

**Neural Tangent Feature (NTF) predictor** For parameter vector $\bar{\boldsymbol{\theta}}_t^j \in \mathbb{R}^{Nd}$ and $\bar{\boldsymbol{\theta}}_t = (\bar{\boldsymbol{\theta}}_t^1, \ldots, \bar{\boldsymbol{\theta}}_t^d)$ (the procedure for obtaining them is discussed later), we have the following NTF predictor defined as

$$\mathbf{h}_t^{\mathrm{rf}} = \begin{bmatrix} (\phi_0^1(\boldsymbol{\alpha}))^\top \bar{\boldsymbol{\theta}}_t^1 \\ \vdots \\ (\phi_0^d(\boldsymbol{\alpha}))^\top \bar{\boldsymbol{\theta}}_t^d \end{bmatrix} .$$

We also use $h_t^{\mathrm{rf},j}(\boldsymbol{\alpha})$ to indicate the $j$th coordinate of $\mathbf{h}_t^{\mathrm{rf}}(\boldsymbol{\alpha})$. Then, the objective of the NTF predictor is

$$\widehat{\mathcal{R}}^{\mathrm{rf}}(\boldsymbol{\theta}_t) = \frac{1}{n}\sum_{i=1}^n \left(\mathbf{h}_t^{\mathrm{rf}}(\boldsymbol{\alpha}_i) - \mathbf{w}^*(\boldsymbol{\alpha}_i)\right)^2 = \frac{1}{n}\sum_{i=1}^n\sum_{j=1}^d \left(\phi_0^{j\top}(\boldsymbol{\alpha}_i)\bar{\boldsymbol{\theta}}_t^j - w^*(\boldsymbol{\alpha}_i)(j)\right)^2 ,$$

its pointwise counterpart is

$$\widehat{\mathcal{R}}^{\mathrm{rf},j}(\boldsymbol{\theta}_t^j) = \frac{1}{n}\sum_{i=1}^n \left(h_t^{\mathrm{rf},j}(\boldsymbol{\alpha}_i) - w^*(\boldsymbol{\alpha}_i)(j)\right)^2 = \frac{1}{n}\sum_{i=1}^n \left(\phi_0^{j\top}(\boldsymbol{\alpha}_i)\bar{\boldsymbol{\theta}}_t^j - w^*(\boldsymbol{\alpha}_i)(j)\right)^2 ,$$

and GD update rule is defined as

$$\bar{\boldsymbol{\theta}}_{t+1} = \bar{\boldsymbol{\theta}}_t - \eta\nabla\widehat{\mathcal{R}}^{\mathrm{rf}}(\boldsymbol{\theta}_t) .$$

**Kernel Least Squares (KLS) predictor** In the following we will work with a kernel function:

**Proposition 3** (Kernel induced by Rectified Linear Unit (ReLU) activation)**.** *The following kernel function is called the NTK function induced by the ReLU activation:*

$$\kappa(\boldsymbol{\alpha}, \boldsymbol{\alpha}') = (\boldsymbol{\alpha}^\top \boldsymbol{\alpha}') \int_{\mathbb{R}^d} \mathbb{I}\left\{\mathbf{w}^\top \boldsymbol{\alpha} > 0\right\} \mathbb{I}\left\{\mathbf{w}^\top \boldsymbol{\alpha}' > 0\right\} \mathcal{N}(\mathrm{d}\mathbf{w} \mid \mathbf{0}, \mathbf{I}_d) \qquad (\boldsymbol{\alpha}, \boldsymbol{\alpha}' \in \mathbb{S}^{d-1}) .$$

*The following holds for $\kappa$:*

- *It is a reproducing kernel and has analytic form $\kappa(\boldsymbol{\alpha}, \boldsymbol{\alpha}') = (\boldsymbol{\alpha}^\top \boldsymbol{\alpha}')(\pi - \arccos(\boldsymbol{\alpha}^\top \boldsymbol{\alpha}'))$ [15].*

- $\sup_{\boldsymbol{\alpha}, \boldsymbol{\alpha}' \in \mathbb{S}^{d-1}} \kappa(\boldsymbol{\alpha}, \boldsymbol{\alpha}') \leq 1.$

- *Eigenvalues of $\kappa$ satisfy $\mu_k \leq C\, k^{-\frac{d}{2}}$ for $k \in \mathbb{N}$ [2].*

- *The NTK matrix is a symmetric matrix $\mathbf{K} \in \mathbb{R}^{n \times n}$ with entries $(\mathbf{K})_{i,j} = \kappa(\boldsymbol{\alpha}_i, \boldsymbol{\alpha}_j)$.*

*Throughout, $\mathcal{H}$ is the RKHS induced by $\kappa$.*

Having established the kernel function, we assume the following about matrix $\mathbf{K}$:

**Assumption 2** (Smallest eigenvalue of the NTK matrix)**.** *Assume that there exists fixed $\lambda_0 > 0$ such that $\mathbb{P}(\lambda_{\min}(\mathbf{K}) \geq \lambda_0) \geq 1 - \delta_{\lambda_0}$ and $\delta_{\lambda_0} \in [0, 1]$.*

Assumption 2 is fairly standard and often satisfied with sample-dependent lower bounds:

- In particular, [9] shows that $\lambda_0 > 0$ whenever no two distinct inputs are parallel.
- In a random design setting, [4, Lemma 5.2] show that when inputs are sampled from isotropic Gaussian and $n \leq d^C$ for some activation function-dependent $C$, $\lambda_0 = \Omega(d)$ with probability at least $1 - e^{-\mathcal{O}(n)}$.
- In a random design setting, [29] show that (for a certain well-behaved family of input distributions), $\lambda_{\min}(\mathbf{K}) = \Theta_{\mathbb{P}}(d)$ with high probability. More precisely, their Theorem 3.2 implies that we have $\lambda_{\min}(\mathbf{K}) \geq C\operatorname{polylog}(n, d)d$ with probability at least $1 - n^2 e^{-\mathcal{O}(\sqrt{d})}$.

In addition to the NTK matrix, we define its empirical counterpart for the $j$th component of the hidden layer, namely

$$\hat{\mathbf{K}}^j = (\boldsymbol{\Phi}_0^j)^\top \boldsymbol{\Phi}_0^j \in \mathbb{R}^{n \times n} .$$

Then, it is clear that $\mathbf{K} = \mathbb{E}[\hat{\mathbf{K}}^j \mid \text{data}]$.

Now, for some vector $\mathbf{c} \in \mathbb{R}^d$, we define the KLS predictor as

$$\mathbf{h}_t^{\text{kls}}(\boldsymbol{\alpha}) = \begin{bmatrix} \sum_{i=1}^n k(\boldsymbol{\alpha}_i, \boldsymbol{\alpha})c_i^1 \\ \vdots \\ \sum_{i=1}^n k(\boldsymbol{\alpha}_i, \boldsymbol{\alpha})c_i^d \end{bmatrix},$$

We also use $h_t^{\text{kls},j}(\boldsymbol{\alpha})$ to indicate the $j$th coordinate of $\mathbf{h}_t^{\text{kls}}(\boldsymbol{\alpha})$. The objective for KLS predictor is defined as:

$$\widehat{\mathcal{R}}^{\text{kls}}(\mathbf{c}_t) = \frac{1}{n} \sum_{i=1}^n \left(\mathbf{h}_t^{\text{kls}}(\boldsymbol{\alpha}_i) - \mathbf{w}^*(\boldsymbol{\alpha}_i)\right)^2$$

for the pointwise loss function as:

$$\widehat{\mathcal{R}}^{\text{kls},j}(\mathbf{c}_t^j) = \frac{1}{n} \sum_{i=1}^n \left(\mathbf{h}_t^{\text{kls},j}(\boldsymbol{\alpha}_i) - w^*(\boldsymbol{\alpha}_i)(j)\right)^2$$

and finally GD update rule is defined as:

$$\mathbf{c}_{t+1} = \mathbf{c}_t - \eta \nabla \widehat{\mathcal{R}}^{\text{kls}}(\mathbf{c}_t) .$$

**Proposition 4.** *Let $\mathbf{w}_i^t$ be iterates generated by Algorithm 2. Then the following statement holds true:*

$$\left\|\mathbf{w}_i^t - \mathbf{w}^*(\boldsymbol{\alpha}_i)\right\|^2 \leq (1 - \mu\gamma)^t D$$

### B.3 Properties of ReLU Networks

In this section we include standard lemmata for analysis of shallow neural networks, and where appropriate we account for the fact that we are working with vector-valued predictors.

**Proposition 5** (Activation patterns [21]). *Assume that initial parameters of a neural network are chosen as in section 3.1. Consider the set of indices of neurons that changed their activation patterns on input $\boldsymbol{\alpha}$, when $\boldsymbol{\theta}_0$ is replaced by some parameters $\tilde{\boldsymbol{\theta}} = (\tilde{\mathbf{u}}_1, \ldots, \tilde{\mathbf{u}}_m)$:*

$$P(\tilde{\boldsymbol{\theta}}, \boldsymbol{\alpha}) = \left\{ k \in [m] | \mathbb{I}\{\tilde{\mathbf{u}}^\top(k)\boldsymbol{\alpha} > 0\} - \mathbb{I}\{\mathbf{u}_0^\top(k)\boldsymbol{\alpha} > 0\} \neq 0 \right\} \qquad (\tilde{\boldsymbol{\theta}} \in \mathbb{R}^{Nm}, \boldsymbol{\alpha} \in \Delta^N).$$

*Then, for $\tilde{\boldsymbol{\theta}}$ whose components satisfy $\max_k \|\tilde{\mathbf{u}}_k - \mathbf{u}_{0,k}\| \leq \rho$ for some fixed $\rho \geq 0$, and any $\boldsymbol{\alpha} \in \Delta^N$, the following facts hold:*

(a) *For all $k \in P(\tilde{\boldsymbol{\theta}}, \boldsymbol{\alpha})$, $|\mathbf{u}_{0,k}^\top \boldsymbol{\alpha}| \leq \|\tilde{\mathbf{u}}_k - \mathbf{u}_{0,k}\|$.*

(b) $\mathbb{E}\,|P(\tilde{\boldsymbol{\theta}}, \boldsymbol{\alpha})| \leq m\rho$.

(c) *With probability at least $1 - 2e^{-\nu}$ for any $\nu > 0$, $|P(\tilde{\boldsymbol{\theta}}, \boldsymbol{\alpha})| \leq m\rho + \sqrt{m\nu}$.*

*Also*

(d) $\left\| \phi_{\tilde{\boldsymbol{\theta}}} - \phi_0 \right\| \leq \rho + \sqrt{\frac{\nu}{m}}$.

(e) $\left\| (\phi_{\tilde{\boldsymbol{\theta}}} - \phi_0)^\top \tilde{\boldsymbol{\theta}} \right\| \leq \rho(\sqrt{m}\rho + \sqrt{\nu})$.

**Proposition 6** (Bounded gradient). *The gradients of neural network are bounded:*

$$\|\nabla_{\boldsymbol{\theta}} \mathbf{h}_{\boldsymbol{\theta}}(\boldsymbol{\alpha})\|^2 \leq d, \quad \left\| \nabla \widehat{\mathcal{R}}(\boldsymbol{\theta}) \right\|^2 \leq 4d\widehat{\mathcal{R}}(\boldsymbol{\theta}).$$

*Proof.* The proof simply follows by definition:

$$\|\nabla_{\boldsymbol{\theta}} \mathbf{h}_{\boldsymbol{\theta}}(\boldsymbol{\alpha})\|^2 = \sum_{j=1}^{d} \left\| \phi^j(\boldsymbol{\alpha}) \right\|^2 = \sum_{j=1}^{d} \sum_{r=1}^{m} \frac{1}{m} \mathbb{I}\{\mathbf{u}_r^\top \boldsymbol{\alpha}\} \|\boldsymbol{\alpha}\|^2 \leq d$$

and

$$\left\| \nabla \widehat{\mathcal{R}}(\boldsymbol{\theta}) \right\|^2 = \left\| \frac{2}{n} \sum_{i=1}^{n} \nabla_{\boldsymbol{\theta}} \mathbf{h}_{\boldsymbol{\theta}}(\boldsymbol{\alpha}_i)(\mathbf{h}_{\boldsymbol{\theta}}(\boldsymbol{\alpha}_i) - \mathbf{w}^*(\boldsymbol{\alpha}_i)) \right\|^2$$

$$\leq 4\frac{1}{n} \sum_{i=1}^{n} \|\nabla_{\boldsymbol{\theta}} \mathbf{h}_{\boldsymbol{\theta}}(\boldsymbol{\alpha}_i)\|^2 \|\mathbf{h}_{\boldsymbol{\theta}}(\boldsymbol{\alpha}_i) - \mathbf{w}^*(\boldsymbol{\alpha}_i))\|^2 \leq 4d\widehat{\mathcal{R}}(\boldsymbol{\theta}).$$

$\square$

**Lemma 10** (Spectral Property of Empirical Kernel Matrix). *Given a set of parameter $\boldsymbol{\theta}_t = \{\mathbf{u}_t(1), ..., \mathbf{u}_t(m)\}$, if we assume $\max_{r \in [m]} \|\mathbf{u}_t(r) - \mathbf{u}_0(r)\| \leq \rho$, then the following statements about the spectral property of its random feature and empirical gram matrix hold true:*

(a) $\|\phi_t(\boldsymbol{\alpha}_i)\| \leq 1$, $\|\boldsymbol{\Phi}_t\| \leq \sqrt{n}$, $\|\boldsymbol{\Phi}_t - \boldsymbol{\Phi}_0\| \leq \sqrt{n(\rho + \sqrt{\frac{\nu}{m}})}$

(b) $\left\| \hat{\mathbf{K}}_t \right\| \leq \left\| \hat{\mathbf{K}}_0 \right\| + 2n^2(\rho + \sqrt{\frac{\nu}{m}})$,

(c) $\lambda_{\min}(\hat{\mathbf{K}}_t) \geq \lambda_{\min}(\hat{\mathbf{K}}_0) - 2n^2(\rho + \sqrt{\frac{\nu}{m}})$,

(d) *If we assume $m \geq \frac{64n^2 \log(n/\nu')}{\lambda_0^2}$, then $\mathbb{P}\left( \lambda_{\min}(\hat{\mathbf{K}}_0) \geq \frac{1}{2}\lambda_0 \right) \geq 1 - \nu'$, and $\mathbb{P}\left( \left\| \hat{\mathbf{K}}_0 \right\| \leq \frac{1}{2} \|\mathbf{K}_0\| \right) \geq 1 - \nu'$.*

*Proof.* We begin with proving $(a)$, the spectral norm $\phi_t(\boldsymbol{\alpha}_i)$

$$\|\phi_t(\boldsymbol{\alpha}_i)\|^2 \le \sum_{r=1}^{m} \left\|a(r)\mathbb{I}\left\{\mathbf{u}_t^\top(r)\boldsymbol{\alpha}_i\right\}\boldsymbol{\alpha}_i\right\|^2 \le 1$$

the spectral bound for $\boldsymbol{\Phi}_t$.

$$\|\boldsymbol{\Phi}_t\|^2 = \|(\phi_t(\boldsymbol{\alpha}_1),\ldots,\phi_t(\boldsymbol{\alpha}_n))\|^2 \le \sum_{i=1}^{n} \|\phi_t(\boldsymbol{\alpha}_i)\|_F^2 \le n,$$

so we know $\|\boldsymbol{\Phi}_t\| \le \sqrt{n}$.

For $\|\boldsymbol{\Phi}_t - \boldsymbol{\Phi}_0\|$, by definition we have:

$$\begin{aligned}
\|\boldsymbol{\Phi}_t - \boldsymbol{\Phi}_0\|^2 &\le \|\boldsymbol{\Phi}_t - \boldsymbol{\Phi}_0\|_F^2 \\
&\le \sum_{i=1}^{n} \|\phi_t(\boldsymbol{\alpha}_i) - \phi_0(\boldsymbol{\alpha}_i)\|^2 \\
&\le \sum_{i=1}^{n} \|\phi_t(\boldsymbol{\alpha}_i) - \phi_0(\boldsymbol{\alpha}_i)\|^2 \\
&\le \sum_{i=1}^{n}\sum_{r=1}^{m} |a(r)\mathbb{I}\{\mathbf{u}_t^\top(r)(\boldsymbol{\alpha}_i)\} - a(r)\mathbb{I}\{\mathbf{u}_0^\top(r)(\boldsymbol{\alpha}_i)\}|^2 \\
&\le \frac{1}{m}\sum_{i=1}^{n} |P(\boldsymbol{\theta}_t,\boldsymbol{\alpha}_i)| \\
&\le n(\rho + \sqrt{\frac{\nu}{m}})\,.
\end{aligned}$$

Now we show the bound on the norm of $\hat{\mathbf{K}}_t$. We first examine $\left\|\phi_t^\top(\boldsymbol{\alpha}_i)\phi_t(\boldsymbol{\alpha}_j) - \phi_0^\top(\boldsymbol{\alpha}_i)\phi_0(\boldsymbol{\alpha}_j)\right\|_F$.
Consider

$$\begin{aligned}
&\left|\phi_t^\top(\boldsymbol{\alpha}_i)\phi_t(\boldsymbol{\alpha}_j) - \phi_0^\top(\boldsymbol{\alpha}_i)\phi_0(\boldsymbol{\alpha}_j)\right| \\
&=\left|\sum_{r=1}^{m} a(r)\mathbb{I}\{\mathbf{u}_t^\top(r)\boldsymbol{\alpha}_i\} \cdot a(r)\mathbb{I}\{\mathbf{u}_t^\top(r)\boldsymbol{\alpha}_j\}\boldsymbol{\alpha}_i^\top\boldsymbol{\alpha}_j - \sum_{r=1}^{m} a(r)\mathbb{I}\{\mathbf{u}_0^\top(r)\boldsymbol{\alpha}_i\} \cdot a(r)\mathbb{I}\{\mathbf{u}_0^\top(r)\boldsymbol{\alpha}_j\}\boldsymbol{\alpha}_i^\top\boldsymbol{\alpha}_j\right| \\
&\le\frac{1}{m}\sum_{r=1}^{m} \left|\mathbb{I}\{\mathbf{u}_t^\top(r)\boldsymbol{\alpha}_i\} \cdot \mathbb{I}\{\mathbf{u}_t^\top(r)\boldsymbol{\alpha}_j\} - \mathbb{I}\{\mathbf{u}_0^\top(r)\boldsymbol{\alpha}_i\} \cdot \mathbb{I}\{\mathbf{u}_0^\top(r)\boldsymbol{\alpha}_j\}\right| |\boldsymbol{\alpha}_i^\top\boldsymbol{\alpha}_j| \\
&=\frac{1}{m}\sum_{r=1}^{m} \Big| \left(\mathbb{I}\{\mathbf{u}_t^\top(r)\boldsymbol{\alpha}_i\} - \mathbb{I}\{\mathbf{u}_0^\top(r)\boldsymbol{\alpha}_i\}\right) \cdot \mathbb{I}\{\mathbf{u}_t^\top(r)\boldsymbol{\alpha}_j\} - \mathbb{I}\{\mathbf{u}_0^\top(r)\boldsymbol{\alpha}_i\} \\
&\qquad\times \left(\mathbb{I}\{\mathbf{u}_0^\top(r)\boldsymbol{\alpha}_j\} - \mathbb{I}\{\mathbf{u}_t^\top(r)\boldsymbol{\alpha}_j\}\right)\Big| |\boldsymbol{\alpha}_i^\top\boldsymbol{\alpha}_j| \\
&=\frac{1}{m}\left(|P(\boldsymbol{\theta}_t,\boldsymbol{\alpha}_i)| + |P(\boldsymbol{\theta}_t,\boldsymbol{\alpha}_j)|\right) \\
&\le 2(\rho + \sqrt{\frac{\nu}{m}})\,.
\end{aligned}$$

Notice that due to triangle inequality we have $\left\|\hat{\mathbf{K}}_t\right\| \leq \left\|\hat{\mathbf{K}}_0\right\| + \left\|\hat{\mathbf{K}}_t - \hat{\mathbf{K}}_0\right\|$. Now we examine $\left\|\hat{\mathbf{K}}_t - \hat{\mathbf{K}}_0\right\|$:

$$\left\|\hat{\mathbf{K}}_t - \hat{\mathbf{K}}_0\right\| \leq \left\|\hat{\mathbf{K}}_t - \hat{\mathbf{K}}_0\right\|_F \tag{22}$$

$$= \sqrt{\sum_{i,j\in[n]} |\boldsymbol{\phi}_t^\top(\boldsymbol{\alpha}_i)\boldsymbol{\phi}_t(\boldsymbol{\alpha}_j) - \boldsymbol{\phi}_0^\top(\boldsymbol{\alpha}_i)\boldsymbol{\phi}_0(\boldsymbol{\alpha}_j)|^2} \tag{23}$$

$$\leq \sum_{i,j\in[n]} |\boldsymbol{\phi}_t^\top(\boldsymbol{\alpha}_i)\boldsymbol{\phi}_t(\boldsymbol{\alpha}_j) - \boldsymbol{\phi}_0^\top(\boldsymbol{\alpha}_i)\boldsymbol{\phi}_0(\boldsymbol{\alpha}_j)| \tag{24}$$

$$= 2n^2(\rho + \sqrt{\frac{\nu}{m}}). \tag{25}$$

At last, we prove statement (d). According to Weyl's inequality:

$$\lambda_{\min}(\hat{\mathbf{K}}_0) \geq \lambda_{\min}(\mathbf{K}) - \left\|\hat{\mathbf{K}}_0 - \mathbf{K}\right\|.$$

According to basic concentration inequality we know that the following statement holds with probability at least $1 - \nu$:

$$|\hat{\mathbf{K}}_0(i,j) - \mathbf{K}(i,j)| \leq \frac{2\sqrt{\log(1/\nu')}}{\sqrt{m}}$$

Taking union bound over all $n^2$ entries and summing over all entries yields:

$$\left\|\hat{\mathbf{K}}_0 - \mathbf{K}\right\| \leq \left\|\hat{\mathbf{K}}_0 - \mathbf{K}\right\|_F \leq \sum_{i,j\in[n]} |\hat{\mathbf{K}}_0(i,j) - \mathbf{K}(i,j)| \leq \frac{4n\sqrt{\log(n/\nu')}}{\sqrt{m}}$$

To ensure the RHS less than $\frac{1}{2}\lambda_0$, we need

$$m \geq \frac{64n^2 \log(n/\nu')}{\lambda_0^2}.$$

Last, since $\left\|\hat{\mathbf{K}}_0 - \mathbf{K}\right\| \leq \frac{1}{2}\lambda_0$, we know that it is also true $\left\|\hat{\mathbf{K}}_0 - \mathbf{K}\right\| \leq \frac{1}{2}\sigma_{\max} = \frac{1}{2}\|\mathbf{K}\|$. $\qquad \square$

**Lemma 11** (Deviation of Gradient Computed on Inexact Labels). *For Algorithm 2, the following statements hold true:*

$$\left\|\nabla_{\boldsymbol{\theta}_t^j}\widehat{\mathcal{R}}^j(\boldsymbol{\theta}_t^j) - \mathbf{g}_t^j\right\|^2 \leq 4(1 - \mu\gamma)^{Kt}D,$$

$$\left\|\boldsymbol{\Phi}_t^{j^\top}(\nabla\widehat{\mathcal{R}}^j(\boldsymbol{\theta}_t^j) - \mathbf{g}_t^j)\right\|^2 \leq \frac{4}{n}\left(\frac{3}{2}\|\mathbf{K}\| + 2n^2\left(\rho + \sqrt{\frac{\nu}{m}}\right)\right)^2 (1 - \mu\gamma)^{Kt}D.$$

*Proof.* Recall the definition of $\nabla_{\boldsymbol{\theta}_t}\widehat{\mathcal{R}}(\boldsymbol{\theta}_t)$ and $\mathbf{g}_t$:

$$\nabla_{\boldsymbol{\theta}_t}\widehat{\mathcal{R}}(\boldsymbol{\theta}_t) = \begin{bmatrix} \frac{2}{n}\sum_{i=1}^n \boldsymbol{\phi}_t^1(h_t^1(\boldsymbol{\alpha}_i) - \mathbf{w}^*(\boldsymbol{\alpha}_i)(1)) \\ \vdots \\ \frac{2}{n}\sum_{i=1}^n \boldsymbol{\phi}_t^d(h_t^d(\boldsymbol{\alpha}_i) - \mathbf{w}^*(\boldsymbol{\alpha}_i)(d)) \end{bmatrix} = \begin{bmatrix} \frac{2}{n}\boldsymbol{\Phi}_t^1(\mathbf{h}_t^1 - \mathbf{w}^*(1)) \\ \vdots \\ \frac{2}{n}\boldsymbol{\Phi}_t^d(\mathbf{h}_t^d - \mathbf{w}^*(d)) \end{bmatrix},$$

$$\mathbf{g}_t = \begin{bmatrix} \frac{2}{n}\sum_{i=1}^n \boldsymbol{\phi}_t^1(h_t^1(\boldsymbol{\alpha}_i) - \mathbf{w}_i^t(1)) \\ \vdots \\ \frac{2}{n}\sum_{i=1}^n \boldsymbol{\phi}_t^d(h_t^d(\boldsymbol{\alpha}_i) - \mathbf{w}_i^t(d)) \end{bmatrix} = \begin{bmatrix} \frac{2}{n}\boldsymbol{\Phi}_t^1(\mathbf{h}_t^1 - \mathbf{w}_i^t(1)) \\ \vdots \\ \frac{2}{n}\boldsymbol{\Phi}_t^d(\mathbf{h}_t^d - \mathbf{w}_i^t(d)) \end{bmatrix}$$

where $\mathbf{w}^t(j) = [w_1^t(j), ..., w_n^t(j)]^\top \in \mathbb{R}^n$ and $\mathbf{w}^*(j) = [w^*(\boldsymbol{\alpha}_1)(j), ..., w^*(\boldsymbol{\alpha}_n)(j)]^\top \in \mathbb{R}^n$. Hence

$$
\begin{aligned}
\left\| \nabla_{\boldsymbol{\theta}_t^j} \widehat{\mathcal{R}}^j(\boldsymbol{\theta}_t^j) - \mathbf{g}_t^j \right\|^2 &= \left\| \frac{2}{n} \boldsymbol{\Phi}_t^j (\mathbf{h}_t^j - \mathbf{w}^*(j)) - \frac{2}{n} \boldsymbol{\Phi}_t^j \left( \mathbf{h}_t^j - \mathbf{w}^t(j) \right) \right\|^2 \\
&= \left\| \frac{2}{n} \boldsymbol{\Phi}_t^j (\mathbf{w}^t(j) - \mathbf{w}^*(j)) \right\|^2 \\
&\leq \frac{4}{n^2} \max_{j \in [d]} \left\| \boldsymbol{\Phi}_t^j \right\|^2 \sum_{i=1}^n \left\| \mathbf{w}_i^t - \mathbf{w}^*(\boldsymbol{\alpha}_i) \right\|^2 \\
&\leq 4(1 - \mu\gamma)^{Kt} D \,.
\end{aligned}
$$

For the second result we have

$$
\begin{aligned}
\left\| \boldsymbol{\Phi}_t^{j\top} (\nabla \widehat{\mathcal{R}}^j(\boldsymbol{\theta}_t^j) - \mathbf{g}_t^j) \right\|^2 &= \frac{4}{n^2} \left\| \boldsymbol{\Phi}_t^\top \boldsymbol{\Phi}_t (\mathbf{h}_t^j - \mathbf{w}^*(j)) - \boldsymbol{\Phi}_t^{j\top} \boldsymbol{\Phi}_t^j (\mathbf{h}_t^j - \mathbf{w}^t(j)) \right\|^2 \\
&= \frac{4}{n^2} \left\| \boldsymbol{\Phi}_t^{j\top} \boldsymbol{\Phi}_t^j (\mathbf{w}^t(j) - \mathbf{w}^*(j)) \right\|^2 \\
&\leq \frac{4}{n^2} \left\| \boldsymbol{\Phi}_t^{j\top} \boldsymbol{\Phi}_t^j \right\|^2 n(1 - \mu\gamma)^{Kt} D \\
&\leq \frac{4}{n} \left( \left\| \hat{\mathbf{K}}_0 \right\| + 2n^2(\rho + \sqrt{\frac{\nu}{m}}) \right)^2 (1 - \mu\gamma)^{Kt} D \\
&\leq \frac{4}{n} \left( \frac{3}{2} \left\| \mathbf{K} \right\| + 2n^2(\rho + \sqrt{\frac{\nu}{m}}) \right)^2 (1 - \mu\gamma)^{Kt} D \,.
\end{aligned}
$$

$\square$

The following theorem shows that Algorithm 2 can converge on empirical risk defined in (4).

**Theorem 5** (Convergence of Algorithm). *For Algorithm 2, if we choose*

$$
m \geq \left( \frac{48nB_y}{\lambda_0} + \sqrt{\nu} \right)^2 \frac{256n^4}{\lambda_0^2},
$$

$$
\eta \leq \min \left\{ \frac{3n^3}{\lambda_0} \left( \frac{48nB_y}{\lambda_0} + \sqrt{\nu} \right), \frac{3\lambda_0 n^2}{256 \left( \frac{1}{2} \left\| \mathbf{K} \right\| + \frac{\lambda_0}{8} \right)^2} \right\},
$$

*and assume the inner iteration number of Algorithm 1 satisfying*

$$
K \geq \max \left\{ \kappa \log \left( \frac{\eta \lambda_0 n T^2 D}{16 B_y^2 n} \right), 2\kappa \log \left( \frac{\eta \lambda_0 \kappa \sqrt{D}}{4 B_y n} \right) \right\}, \tag{26}
$$

*then the following statement holds for any $t \in [T]$ and $j \in [d]$:*

$$
\widehat{\mathcal{R}}^j(\boldsymbol{\theta}_{t+1}^j) \leq \left( 1 - \frac{\eta \lambda_0}{4n} \right)^t \widehat{\mathcal{R}}^j(\boldsymbol{\theta}_0^j) + \frac{4nC (1 - \mu\gamma)^K D}{\eta \lambda_0},
$$

*with*

$$
C = \frac{\lambda_0^2 \eta^2}{24n^4} + \frac{4\eta^2 \left( \frac{1}{2} \left\| \mathbf{K} \right\| + \frac{\lambda_0}{8} \right)^2}{\frac{3}{8} n^2}.
$$

*Proof.* We prove by induction. We make the following inductive hypotheses:

(I)   $\widehat{\mathcal{R}}^j(\boldsymbol{\theta}_t^j) \leq \left( 1 - \frac{\eta \lambda_0}{4n} \right)^{t-1} \widehat{\mathcal{R}}^j(\boldsymbol{\theta}_0^j) + \sum_{s=0}^{t-1} \left( 1 - \frac{\eta \lambda_0}{4n} \right)^{t-1-s} (1 - \mu\gamma)^s C(1 - \mu\gamma)^K D,$

(II)   $\max_{r \in [m]} \left\| \mathbf{u}_t^j(r) - \mathbf{u}_t^j(r) \right\| \leq \rho := \frac{48nB_y}{\lambda_0 \sqrt{m}}.$

where $C$ is the constant to be determined later. First we are going to show hypothesis (**II**) holding for $t+1$. First, by our choice of $\eta$, we know $(1 - \mu\gamma) \le \left(1 - \frac{\eta\lambda_0}{4n}\right)$. Hence

$$\sum_{s=0}^{t-1} \left(1 - \frac{\eta\lambda_0}{4n}\right)^{t-1-s} (1 - \mu\gamma)^s \le \frac{4n}{\eta\lambda_0}.$$

Now we are going to bound maximal neuron drifting from initialization: for any $j \in [d]$

$$\left\|\mathbf{u}_{t+1}^j(r) - \mathbf{u}_0^j(r)\right\|$$

$$= \left\|\sum_{t'=0}^{t} \eta \mathbf{g}_{t'}^j(r)\right\| = \eta \left\|\sum_{t'=0}^{t} \frac{2}{n} \sum_{i=1}^{n} a_j(r) \mathbb{I}\left\{\mathbf{u}_t^j(r)^\top \boldsymbol{\alpha}_i\right\} \boldsymbol{\alpha}_i (h_t^j(\boldsymbol{\alpha}_i) - w_t^i(j))\right\|$$

$$\le \frac{2\eta}{\sqrt{m}} \sum_{t'=0}^{t} \frac{1}{n} \sum_{i=1}^{n} \left(|h_t^j(\boldsymbol{\alpha}_i) - w_*^i(j)| + |w_*^i(j) - w_t^i(j)|\right)$$

$$\le \frac{2\eta}{\sqrt{m}} \sum_{t'=0}^{t} \left(\sqrt{\frac{1}{n} \sum_{i=1}^{n} \left(h_t^j(\boldsymbol{\alpha}_i) - w_*^i(j)\right)^2} + \sqrt{\frac{1}{n} \sum_{i=1}^{n} \left(w_*^i(j) - w_t^i(j)\right)^2}\right)$$

$$\le \frac{2\eta}{\sqrt{m}} \sum_{t'=0}^{t} \left(\sqrt{\widehat{\mathcal{R}}^j(\boldsymbol{\theta}_{t'}^j)} + \sqrt{(1 - \mu\gamma)^{Kt'} D}\right)$$

$$\le 2\eta \sum_{t'=0}^{t} \left(\frac{\sqrt{\left(1 - \frac{\eta\lambda_0}{4n}\right)^{t'-1} \widehat{\mathcal{R}}^j(\boldsymbol{\theta}_0^j) + C\frac{4n(1-\mu\gamma)^K D}{\eta\lambda_0}}}{\sqrt{m}} + \frac{1}{\sqrt{m}}\sqrt{(1 - \mu\gamma)^{Kt'} D}\right)$$

$$\le 2\eta \left(\frac{\frac{8n}{\eta\lambda_0}\sqrt{\widehat{\mathcal{R}}(\boldsymbol{\theta}_0)} + t\sqrt{C\frac{4n(1-\mu\gamma)^K D}{\eta\lambda_0}}}{\sqrt{m}} + \frac{1}{\sqrt{m}\mu\gamma}\sqrt{(1 - \mu\gamma)^K D}\right),$$

where at last step we use the fact $\sqrt{1 - a} \le 1 - \frac{a}{2}$ for $a \in (0, 1)$, and triangle inequality to split terms inside square root. Since we choose $\gamma = \frac{1}{L}$, we have

$$\|\mathbf{u}_{t+1}(r) - \mathbf{u}_t(r)\| \le 2\eta \left(\frac{8nB_y}{\eta\lambda_0\sqrt{m}} + t\sqrt{C\frac{4n\left(1 - \frac{1}{\kappa}\right)^K D}{\eta\lambda_0 m}} + \frac{2\kappa\sqrt{\left(1 - \frac{1}{\kappa}\right)^K D}}{\sqrt{m}}\right)$$

According to our choice of $K$:

$$K \ge \max\left\{\kappa \log\left(\frac{\eta\lambda_0 nT^2 CD}{16B_y^2 n}\right), 2\kappa \log\left(\frac{\eta\lambda_0\kappa\sqrt{D}}{4B_y n}\right)\right\},$$

we have:

$$t\sqrt{C\frac{4n\left(1 - \frac{1}{\kappa}\right)^K D}{\eta\lambda_0 m}} \le \frac{8nB_y}{\eta\lambda_0\sqrt{m}},$$

$$\frac{2\kappa\sqrt{\left(1 - \frac{1}{\kappa}\right)^K D}}{\sqrt{m}} \le \frac{8nB_y}{\eta\lambda_0\sqrt{m}}.$$

Hence we conclude that:

$$\|\mathbf{u}_{t+1}(r) - \mathbf{u}_t(r)\| \le \frac{48nB_y}{\lambda_0\sqrt{m}}.$$

Now we switch to proving hypothesis (**I**). According to updating rule, we have

$$\boldsymbol{\theta}_{t+1}^j = \boldsymbol{\theta}_t^j - \eta \mathbf{g}_t^j$$

$$= \boldsymbol{\theta}_t^j - \eta \frac{2}{n} \boldsymbol{\Phi}_t^j (\mathbf{h}_t^j - \mathbf{w}^t(j))$$

where $\mathbf{w}^t(j) = [w_1^t(j), ..., w_n^t(j)]$. Hence

$$
\begin{aligned}
\widehat{\mathcal{R}}^j(\boldsymbol{\theta}_{t+1}^j) &= \frac{1}{n} \left\| \mathbf{\Phi}_{t+1}^j{}^\top \boldsymbol{\theta}_{t+1}^j - \mathbf{w}^*(j) \right\|^2 \\
&= \frac{1}{n} \left\| \mathbf{\Phi}_{t+1}^j{}^\top \boldsymbol{\theta}_{t+1}^j - \mathbf{\Phi}_t^j{}^\top \boldsymbol{\theta}_{t+1}^j + \mathbf{\Phi}_t^j{}^\top \boldsymbol{\theta}_{t+1}^j - \mathbf{w}^*(j) \right\|^2 \\
&\leq \inf_{p>0} \left(1 + \frac{1}{p}\right) \underbrace{\frac{1}{n} \left\| \mathbf{\Phi}_{t+1}^j{}^\top \boldsymbol{\theta}_{t+1}^j - \mathbf{\Phi}_t^j{}^\top \boldsymbol{\theta}_{t+1}^j \right\|^2}_{T_1} + (1+p) \underbrace{\frac{1}{n} \left\| \mathbf{\Phi}_t^j{}^\top \boldsymbol{\theta}_{t+1}^j - \mathbf{w}^*(j) \right\|^2}_{T_2}
\end{aligned}
$$
$$(27)$$

where $\mathbf{w}^*(j) = [w^*(\boldsymbol{\alpha}_1)(j), ..., w^*(\boldsymbol{\alpha}_n)(j)]$.

For $T_1$:

$$
T_1 = \frac{1}{n} \left\| \mathbf{\Phi}_{t+1}^j{}^\top \boldsymbol{\theta}_{t+1}^j - \mathbf{\Phi}_t^j{}^\top \boldsymbol{\theta}_{t+1}^j \right\|^2 = \frac{1}{n} \sum_{i=1}^n \left\| \sum_{r=1}^m a_j(r)(\mathbb{I}\{\mathbf{u}_{t+1}^j{}^\top(r)\boldsymbol{\alpha}_i\} - \mathbb{I}\{\mathbf{u}_t^j{}^\top(r)\boldsymbol{\alpha}_i\})\mathbf{u}_t^j{}^\top(r)\boldsymbol{\alpha}_i \right\|^2
$$

Notice that:

$$
\begin{aligned}
&\left| \sum_{r=1}^m a_j(r)(\mathbb{I}\{\mathbf{u}_{t+1}^j(r)^\top \boldsymbol{\alpha}_i\} - \mathbb{I}\{\mathbf{u}_t^j(r)^\top \boldsymbol{\alpha}_i\})\mathbf{u}_t^\top(r)\boldsymbol{\alpha}_i \right| \\
&\leq \frac{1}{\sqrt{m}} \sum_{r=1}^m \left| (\mathbb{I}\{\mathbf{u}_{t+1}^j(r)^\top \boldsymbol{\alpha}_i\} - \mathbb{I}\{\mathbf{u}_t^j(r)^\top \boldsymbol{\alpha}_i\})\mathbf{u}_t^\top(r)\boldsymbol{\alpha}_i \right| \\
&\leq \frac{1}{\sqrt{m}} \sum_{r=1}^m \left| (\mathbb{I}\{\mathbf{u}_{t+1}^j(r)^\top \boldsymbol{\alpha}_i\} - \mathbb{I}\{\mathbf{u}_t^j(r)^\top \boldsymbol{\alpha}_i\}) \right| \left\| \mathbf{u}_t^j{}^\top(r) \right\| \\
&\leq \frac{1}{\sqrt{m}} (|P(\boldsymbol{\theta}_{t+1}, \boldsymbol{\alpha}_i)| + |P(\boldsymbol{\theta}_t, \boldsymbol{\alpha}_i)|) \left\| \mathbf{u}_t^\top(r) - \mathbf{u}_{t+1}^\top(r) \right\|.
\end{aligned}
$$

According to updating rule, we have:

$$
\begin{aligned}
\left\| \mathbf{u}_t^j{}^\top(r) - \mathbf{u}_{t+1}^j{}^\top(r) \right\| &= \left\| \eta \mathbf{g}_t^j(r) \right\| \\
&\leq \eta \left\| \frac{2}{n} \sum_{i=1}^n \mathbb{I}\left\{ \mathbf{u}_t^j{}^\top(r)\boldsymbol{\alpha}_i \right\} \boldsymbol{\alpha}_i^\top a_j(r)(h_t^j(\boldsymbol{\alpha}_i) - w_i^t(j)) \right\| \\
&\quad + \eta \left\| \frac{2}{n} \sum_{i=1}^n \mathbb{I}\left\{ \mathbf{u}_t^j(r)^\top \boldsymbol{\alpha}_i \right\} \boldsymbol{\alpha}_i^\top a_j(r)(w^*(\boldsymbol{\alpha}_i)(j) - w_i^t(j)) \right\| \\
&\leq \frac{2\eta \sqrt{\widehat{\mathcal{R}}^j(\boldsymbol{\theta}_t^j)}}{\sqrt{m}} + \frac{2\eta}{\sqrt{m}} \sqrt{(1 - \mu\gamma)^{Kt} D}
\end{aligned}
$$

Since we prove that $\max_{r\in[m]} \left\| \mathbf{u}_{t+1}^j(r) - \mathbf{u}_t^j(r) \right\| \leq \rho$, we have:

$$
|P(\boldsymbol{\theta}^{t+1}, \boldsymbol{\alpha}_i)| \leq m\rho + \sqrt{m}\nu.
$$

Putting pieces together yields:

$$
\begin{aligned}
T_1 &\leq \frac{4}{m} (m\rho + \sqrt{m}\nu)^2 \left( \frac{2\eta \sqrt{\widehat{\mathcal{R}}^j(\boldsymbol{\theta}_t^j)}}{\sqrt{m}} + \frac{2\eta}{\sqrt{m}} \sqrt{(1 - \mu\gamma)^{Kt} D} \right)^2 \\
&\leq 4 (\sqrt{m}\rho + \sqrt{\nu})^2 \left( \frac{4\eta^2 \widehat{\mathcal{R}}(\boldsymbol{\theta}^t)}{m} + \frac{4\eta^2}{m}(1 - \mu\gamma)^{Kt} D \right).
\end{aligned}
$$
$$(28)$$

For $T_2$:

$$\frac{1}{n}\left\|\boldsymbol{\Phi}_t^{j\top}\boldsymbol{\theta}_{t+1}^j - \mathbf{w}^*(j)\right\|^2$$

$$= \frac{1}{n}\left\|\boldsymbol{\Phi}_t^{j\top}(\boldsymbol{\theta}_t^j - \eta\mathbf{g}_t^j) - \mathbf{w}^*(j)\right\|^2$$

$$= \frac{1}{n}\left\|\boldsymbol{\Phi}_t^{j\top}(\boldsymbol{\theta}_t^j - \frac{2}{n}\eta\boldsymbol{\Phi}_t^j(\mathbf{h}_t^j - \mathbf{w}^*(j))) - \mathbf{w}^*(j) - \boldsymbol{\Phi}_t^{j\top}(\frac{2}{n}\eta\boldsymbol{\Phi}_t^j(\mathbf{w}^t(j) - \mathbf{w}^*(j)))\right\|^2$$

$$\leq \inf_{q>0}\left(1 + \frac{1}{q}\right)\frac{1}{n}\left\|\left(\mathbf{I} - \frac{2}{n}\eta\boldsymbol{\Phi}_t^{j\top}\boldsymbol{\Phi}_t\right)(\mathbf{h}_t^j - \mathbf{w}^*(j))\right\|^2 + (1+q)\eta^2\frac{1}{n}\left\|\frac{2}{n}\boldsymbol{\Phi}_t^{j\top}\boldsymbol{\Phi}_t(\mathbf{w}^t(j) - \mathbf{w}^*(j))\right\|^2$$

$$\leq \inf_{q>0}\left(1 + \frac{1}{q}\right)\left(1 - \frac{2\eta\lambda_{\min}(\hat{\mathbf{K}}_t)}{n}\right)^2\widehat{\mathcal{R}}^j(\boldsymbol{\theta}_t^j) + (1+q)\frac{4\eta^2}{n^3}\left\|\boldsymbol{\Phi}_t^{j\top}\boldsymbol{\Phi}_t^j(\mathbf{w}^t(j) - \mathbf{w}^*(j))\right\|^2$$

We choose $q = \frac{n}{2\eta\lambda_{\min}(\hat{\mathbf{K}}_t)}$, and the fact that $(1+a)(1-a)^2 \leq (1-a)$:

$$T_2 = \frac{1}{n}\left\|\boldsymbol{\Phi}_t^{j\top}\boldsymbol{\theta}_{t+1}^j - \mathbf{w}^*(j)\right\|^2$$

$$\leq \left(1 - \frac{2\eta\lambda_{\min}(\hat{\mathbf{K}}_t)}{n}\right)\widehat{\mathcal{R}}^j(\boldsymbol{\theta}_t^j) + \left(1 + \frac{n}{2\eta\lambda_{\min}(\hat{\mathbf{K}}_t)}\right)\frac{4\eta^2}{n^3}\left\|\boldsymbol{\Phi}_t^{j\top}\boldsymbol{\Phi}_t^j(\mathbf{w}^t(j) - \mathbf{w}^*(j))\right\|^2$$

$$\leq \left(1 - \frac{2\eta\lambda_{\min}(\hat{\mathbf{K}}_t)}{n}\right)\widehat{\mathcal{R}}^j(\boldsymbol{\theta}_t^j)$$

$$+ \left(1 + \frac{n}{2\eta\lambda_{\min}(\hat{\mathbf{K}}_t)}\right)\frac{4\eta^2}{n^3}\left(\left\|\hat{\mathbf{K}}_0\right\| + 2n^2(\rho + \sqrt{\frac{\nu}{m}})\right)^2(1-\mu\gamma)^{Kt}D$$

Since we choose $\eta \leq \frac{n}{2\lambda_{\min}(\hat{\mathbf{K}}_t)}$, we can re-write the above inequality as :

$$T_2 \leq \left(1 - \frac{2\eta\lambda_{\min}(\hat{\mathbf{K}}_t^j)}{n}\right)\widehat{\mathcal{R}}^j(\boldsymbol{\theta}_t^j) + \frac{4\eta}{n^2\lambda_{\min}(\hat{\mathbf{K}}_t)}\left(\left\|\hat{\mathbf{K}}_0\right\| + 2n^2(\rho + \sqrt{\frac{\nu}{m}})\right)^2(1-\mu\gamma)^{Kt}D$$

$$\leq \left(1 - \frac{2\eta\lambda_{\min}(\hat{\mathbf{K}}_0)}{n} + 4n\eta\left(\rho + \sqrt{\frac{\nu}{m}}\right)\right)\widehat{\mathcal{R}}^j(\boldsymbol{\theta}_t^j) + \frac{4\eta\left(\left\|\hat{\mathbf{K}}_0\right\| + 2n^2(\rho + \sqrt{\frac{\nu}{m}})\right)^2(1-\mu\gamma)^{Kt}D}{n^2\left(\lambda_{\min}(\hat{\mathbf{K}}_0) - 2n^2(\rho + \sqrt{\frac{\nu}{m}})\right)}$$

Plugging $T_1$ and $T_2$ back to (27), and using the eigenvalue lower bound from Lemma 10 (d) yields:

$$\widehat{\mathcal{R}}^j(\boldsymbol{\theta}_{t+1}^j) \leq \inf_{p>0}(1+p)\,4\left(\sqrt{m}\rho + \sqrt{\nu}\right)^2\left(\frac{4\eta^2\widehat{\mathcal{R}}(\boldsymbol{\theta}^t)}{m} + \frac{4\eta^2}{m}(1-\mu\gamma)^{Kt}D\right)$$

$$+ \left(1 + \frac{1}{p}\right)\left[\left(1 - \frac{\eta\lambda_0}{n} + 4n\eta\left(\rho + \sqrt{\frac{\nu}{m}}\right)\right)\widehat{\mathcal{R}}^j(\boldsymbol{\theta}_t^j)\right.$$

$$\left. + \frac{4\eta\left(\left\|\hat{\mathbf{K}}_0\right\| + 2n^2(\rho + \sqrt{\frac{\nu}{m}})\right)^2(1-\mu\gamma)^{Kt}D}{n^2\left(\frac{1}{2}\lambda_0 - 2n^2(\rho + \sqrt{\frac{\nu}{m}})\right)}\right].$$

Now we plug in $\rho = \frac{48nB_y}{\lambda_0 \sqrt{m}}$

$$\widehat{\mathcal{R}}^j(\boldsymbol{\theta}_{t+1}^j) \leq \inf_{p>0} (1+p) \, 4 \left( \sqrt{m} \frac{48nB_y}{\lambda_0 \sqrt{m}} + \sqrt{\nu} \right)^2 \left( \frac{4\eta^2 \widehat{\mathcal{R}}^j(\boldsymbol{\theta}_t^j)}{m} + \frac{4\eta^2}{m} (1-\mu\gamma)^{Kt} D \right)$$

$$+ \left( 1 + \frac{1}{p} \right) \left[ \left( 1 - \frac{\eta\lambda_0}{n} + 4n\eta \left( \frac{48nB_y}{\lambda_0 \sqrt{m}} + \sqrt{\frac{\nu}{m}} \right) \right) \widehat{\mathcal{R}}(\boldsymbol{\theta}^t) \right.$$

$$\left. + \frac{4\eta \left( \left\| \hat{\mathbf{K}}_0 \right\| + 2n^2 (\frac{48nB_y}{\lambda_0 \sqrt{m}} + \sqrt{\frac{\nu}{m}}) \right)^2 (1-\mu\gamma)^{Kt} D}{n^2 \left( \frac{1}{2}\lambda_0 - 2n^2 (\frac{48nB_y}{\lambda_0 \sqrt{m}} + \sqrt{\frac{\nu}{m}}) \right)} \right]. \quad (29)$$

According to our choice of $m$:

$$m \geq \left( \frac{48nB_y}{\lambda_0} + \sqrt{\nu} \right)^2 \frac{256n^4}{\lambda_0^2}$$

we know that

$$4n\eta \left( \frac{48nB_y}{\lambda_0 \sqrt{m}} + \sqrt{\frac{\nu}{m}} \right) \leq \frac{\eta\lambda_0}{4n},$$

$$4 \left( \sqrt{m} \frac{48nB_y}{\lambda_0 \sqrt{m}} + \sqrt{\nu} \right)^2 \frac{4\eta^2 \widehat{\mathcal{R}}(\boldsymbol{\theta}^t)}{m} \leq \frac{\eta^2 \lambda_0^2}{16n^4} \widehat{\mathcal{R}}(\boldsymbol{\theta}^t),$$

$$2n^2 \left( \frac{48nB_y}{\lambda_0 \sqrt{m}} + \sqrt{\frac{\nu}{m}} \right) \leq \frac{\lambda_0}{8}.$$

Plugging above inequality back to Eq.(29) yields:

$$\widehat{\mathcal{R}}(\boldsymbol{\theta}^{t+1}) \leq \inf_{p>0} (1+p) \left( 4 \left( \sqrt{m} \frac{48nB_y}{\lambda_0 \sqrt{m}} + \sqrt{\nu} \right)^2 \left( \frac{4\eta^2}{m} (1-\mu\gamma)^{Kt} D \right) + \frac{\eta^2 \lambda_0^2}{16n^4} \widehat{\mathcal{R}}(\boldsymbol{\theta}^t) \right)$$

$$+ \left( 1 + \frac{1}{p} \right) \left( 1 - \frac{3\eta\lambda_0}{4n} \right) \widehat{\mathcal{R}}(\boldsymbol{\theta}^t) + \left( 1 + \frac{1}{p} \right) \frac{4\eta \left( \left\| \hat{\mathbf{K}}_0 \right\| + \frac{\lambda_0}{8} \right)^2 (1-\mu\gamma)^{Kt} D}{\frac{3}{8}\lambda_0 n^2}.$$

According to the fact that:

$$\left( 1 - \frac{1}{a} \right) \left( 1 + \frac{1}{2a-1} \right) = \frac{2a-2}{2a-1} \leq 1 - \frac{1}{2a}$$

we choose $p = \frac{8n}{3\eta\lambda_0} - 1$, and it yields:

$$\widehat{\mathcal{R}}(\boldsymbol{\theta}^{t+1}) \leq \frac{8n}{3\eta\lambda_0} \left( 4 \left( \frac{48nB_y}{\lambda_0} + \sqrt{\nu} \right)^2 \left( \frac{4\eta^2}{m} (1-\mu\gamma)^{Kt} D \right) + \frac{\eta^2 \lambda_0^2}{16n^4} \widehat{\mathcal{R}}(\boldsymbol{\theta}^t) \right)$$

$$+ \left( 1 - \frac{3\eta\lambda_0}{8n} \right) \widehat{\mathcal{R}}(\boldsymbol{\theta}^t) + \left( 1 + \frac{1}{\frac{8n}{3\eta\lambda_0} - 1} \right) \frac{4\eta \left( \left\| \hat{\mathbf{K}}_0 \right\| + \frac{\lambda_0}{8} \right)^2 (1-\mu\gamma)^{Kt} D}{\frac{3}{8}\lambda_0 n^2}.$$

Using the fact $\eta \le \frac{n}{\lambda_0}$ yields:

$$\widehat{\mathcal{R}}(\boldsymbol{\theta}^{t+1}) \le \frac{8n}{3\eta\lambda_0} \left(4\left(\frac{48dnB_y}{\lambda_0} + \sqrt{\nu}\right)^2 \left(\frac{4\eta^2}{m}(1-\mu\gamma)^{Kt}D\right)\right)$$

$$+ \left(1 - \frac{3\eta\lambda_0}{8n} + \frac{\eta\lambda_0}{6n^3}\right) \widehat{\mathcal{R}}(\boldsymbol{\theta}^t) + \frac{8\eta\left(\left\|\hat{\mathbf{K}}_0\right\| + \frac{\lambda_0}{8}\right)^2 (1-\mu\gamma)^{Kt}D}{\frac{3}{8}\lambda_0 n^2}$$

$$\le \left(\frac{48nB_y}{\lambda_0} + \sqrt{\nu}\right)^2 \left(\frac{128n\eta}{3\lambda_0 m}(1-\mu\gamma)^{Kt}D\right)$$

$$+ \left(1 - \frac{3\eta\lambda_0}{8n} + \frac{\eta\lambda_0}{6n^3}\right) \widehat{\mathcal{R}}(\boldsymbol{\theta}^t) + \frac{8\eta\left(\left\|\hat{\mathbf{K}}_0\right\| + \frac{\lambda_0}{8}\right)^2 (1-\mu\gamma)^{Kt}D}{\frac{3}{8}\lambda_0 n^2}.$$

Since $n^2 \ge \frac{4}{3}$, we know $\frac{\eta\lambda_0}{6n^3} \le \frac{\eta\lambda_0}{8n}$. Hence we have:

$$\widehat{\mathcal{R}}(\boldsymbol{\theta}^{t+1})$$

$$\le \left(\frac{48dnB_y}{\lambda_0} + \sqrt{\nu}\right)^2 \left(\frac{128n\eta}{3\lambda_0 m}(1-\mu\gamma)^{Kt}D\right)$$

$$+ \left(1 - \frac{\eta\lambda_0}{4n}\right) \widehat{\mathcal{R}}(\boldsymbol{\theta}^t) + \frac{8\eta\left(\frac{1}{2}\|\mathbf{K}\| + \frac{\lambda_0}{8}\right)^2 (1-\mu\gamma)^{Kt}D}{\frac{3}{8}\lambda_0 n^2}$$

$$= \left(1 - \frac{\eta\lambda_0}{4n}\right) \widehat{\mathcal{R}}(\boldsymbol{\theta}^t) + \frac{4n}{\eta\lambda_0}\left(\left(\frac{48nB_y}{\lambda_0} + \sqrt{\nu}\right)^2 \frac{32\eta^2}{3m} + \frac{2\eta^2\left(\frac{1}{2}\|\mathbf{K}\| + \frac{\lambda_0}{8}\right)^2}{\frac{3}{8}n^2}\right)(1-\mu\gamma)^{Kt}D.$$

Plugging lower bound of $m \ge \left(\frac{48nB_y}{\lambda_0} + \sqrt{\nu}\right)^2 \frac{256n^4}{\lambda_0^2}$ yields:

$$\widehat{\mathcal{R}}(\boldsymbol{\theta}^{t+1}) \le \left(1 - \frac{\eta\lambda_0}{4n}\right) \widehat{\mathcal{R}}(\boldsymbol{\theta}^t) + \frac{4n}{\eta\lambda_0}\left(\frac{\lambda_0^2\eta^2}{24n^4} + \frac{2\eta^2\left(\frac{1}{2}\|\mathbf{K}\| + \frac{\lambda_0}{8}\right)^2}{\frac{3}{8}n^2}\right)(1-\mu\gamma)^{Kt}D.$$

Finally, according to inductive hypothesis (I), we plug in the convergence rate for $\widehat{\mathcal{R}}(\boldsymbol{\theta}^t)$ and get

$$\widehat{\mathcal{R}}(\boldsymbol{\theta}^{t+1}) \le \left(1 - \frac{\eta\lambda_0}{4n}\right)^t \widehat{\mathcal{R}}(\boldsymbol{\theta}_0) + \left(1 - \frac{\eta\lambda_0}{4n}\right)\sum_{s=0}^{t-1}\left(1 - \frac{\eta\lambda_0}{4n}\right)^{t-1-s}(1-\mu\gamma)^s C(1-\mu\gamma)^K D$$

$$+ \frac{4n}{\eta\lambda_0}\left(\frac{\lambda_0^2\eta^2}{24n^4} + \frac{2\eta^2\left(\frac{1}{2}\|\mathbf{K}\| + \frac{\lambda_0}{8}\right)^2}{\frac{3}{8}n^2}\right)(1-\mu\gamma)^{Kt}D$$

$$\le \left(1 - \frac{\eta\lambda_0}{4n}\right)^t \widehat{\mathcal{R}}(\boldsymbol{\theta}_0) + C\sum_{s=0}^{t-1}\left(1 - \frac{\eta\lambda_0}{4n}\right)^{t-s}(1-\mu\gamma)^s(1-\mu\gamma)^K D$$

$$+ \frac{4n}{\eta\lambda_0}\left(\frac{\lambda_0^2\eta^2}{24n^4} + \frac{2\eta^2\left(\frac{1}{2}\|\mathbf{K}\| + \frac{\lambda_0}{8}\right)^2}{\frac{3}{8}n^2}\right)\left(1 - \frac{\eta\lambda_0}{4n}\right)^{t-t}(1-\mu\gamma)^t(1-\mu\gamma)^K D$$

Now we just need to ensure that

$$\frac{\lambda_0^2\eta^2}{24n^4} + \frac{2\eta^2\left(\frac{1}{2}\|\mathbf{K}\| + \frac{\lambda_0}{8}\right)^2}{\frac{3}{8}n^2} \le C.$$

We hence choose $C$ as:

$$C = \frac{\lambda_0^2\eta^2}{24n^4} + \frac{4\eta^2\left(\frac{1}{2}\|\mathbf{K}\| + \frac{\lambda_0}{8}\right)^2}{\frac{3}{8}n^2}$$

Hence we proved that:

$$\widehat{\mathcal{R}}(\boldsymbol{\theta}^{t+1}) \leq \left(1 - \frac{\eta\lambda_0}{4n}\right)^t \widehat{\mathcal{R}}(\boldsymbol{\theta}_0) + C\sum_{s=0}^{t}\left(1 - \frac{\eta\lambda_0}{4n}\right)^{t-s}(1-\mu\gamma)^s(1-\mu\gamma)^K D$$

$$\leq \left(1 - \frac{\eta\lambda_0}{4n}\right)^t \widehat{\mathcal{R}}(\boldsymbol{\theta}_0) + \frac{4nC\,(1-\mu\gamma)^K D}{\eta\lambda_0}.$$

with

$$C = \frac{\lambda_0^2\eta^2}{24n^4} + \frac{4\eta^2\left(\frac{1}{2}\|\mathbf{K}\| + \frac{\lambda_0}{8}\right)^2}{\frac{3}{8}n^2}.$$

$\square$

## B.4 Relations Between Different Predictors

In this section, we are going to relate the predictions of neural network, random feature predictor and kernel OLS predictor.

**Lemma 12** (Neural Network Predictor and Random Feature Predictor Coupling). *Let* $\tilde{n} := \left(\frac{48nB_y}{\lambda_0\sqrt{m}} + \sqrt{\frac{\nu}{m}}\right)$. *For Algorithm 2, the following statement holds for any* $j \in [d]$ *and* $t \in [T]$:

$$\left\|\mathbf{h}_{t+1}^j - \mathbf{h}_{t+1}^{\mathrm{rf},j}\right\|$$

$$\leq \left(1 - \frac{\eta\lambda_0}{4n}\right)^{\frac{t}{2}}\frac{16}{\lambda_0}\left(\left(\frac{3}{2}\|\mathbf{K}\| + 2n^2\tilde{n} + 2\sqrt{n}n\tilde{n}\right)\sqrt{(1-\mu\gamma)^K D} + \left(2\sqrt{n}n^2\tilde{n} + 2\sqrt{n}n\tilde{n}\right)\sqrt{\widehat{\mathcal{R}}(\boldsymbol{\theta}_0)}\right)$$

$$+ \left(2\sqrt{n}\tilde{n} + 4\sqrt{n}n\tilde{n}\right)\frac{8n}{\lambda_0}\sqrt{\frac{4nC\,(1-\mu\gamma)^K D}{\eta\lambda_0}}.$$

*Proof.* Notice the following canonical decomposition:

$$h_{t+1}^j(\boldsymbol{\alpha}) - h_{t+1}^{\mathrm{rf},j}(\boldsymbol{\alpha})$$
$$= \boldsymbol{\phi}_{t+1}^{j}{}^\top\boldsymbol{\theta}_{t+1}^j - \boldsymbol{\phi}_0^j{}^\top\bar{\boldsymbol{\theta}}_{t+1}^j$$
$$= \boldsymbol{\phi}_t^j{}^\top\boldsymbol{\theta}_{t+1}^j + (\boldsymbol{\phi}_{t+1}^j{}^\top - \boldsymbol{\phi}_t^j{}^\top)\boldsymbol{\theta}_{t+1}^j - \boldsymbol{\phi}_0^j{}^\top\bar{\boldsymbol{\theta}}_{t+1}^j$$
$$= \boldsymbol{\phi}_t^j{}^\top(\boldsymbol{\theta}_t^j - \eta\mathbf{g}_t^j) + (\boldsymbol{\phi}_{t+1}^j{}^\top - \boldsymbol{\phi}_t^j{}^\top)\boldsymbol{\theta}_{t+1}^j - \boldsymbol{\phi}_0^j{}^\top\bar{\boldsymbol{\theta}}_{t+1}^j$$
$$= \boldsymbol{\phi}_t^j{}^\top(\boldsymbol{\theta}_t^j - \eta\nabla_{\boldsymbol{\theta}_t^j}\widehat{\mathcal{R}}^j(\boldsymbol{\theta}_t^j) - \eta(\mathbf{g}_t^j - \nabla_{\boldsymbol{\theta}_t^j}\widehat{\mathcal{R}}^j(\boldsymbol{\theta}_t^j))) + (\boldsymbol{\phi}_{t+1}^j{}^\top - \boldsymbol{\phi}_t^j{}^\top)\boldsymbol{\theta}_{t+1}^j - \boldsymbol{\phi}_0^j{}^\top(\bar{\boldsymbol{\theta}}_t^j - \eta\nabla_{\bar{\boldsymbol{\theta}}_t^j}\mathcal{R}^{\mathrm{rf},j}(\bar{\boldsymbol{\theta}}_t^j))$$
$$= h_t^j(\boldsymbol{\alpha}) - h_t^{\mathrm{rf},j}(\boldsymbol{\alpha}) + (\boldsymbol{\phi}_{t+1}^j{}^\top - \boldsymbol{\phi}_t^j{}^\top)\boldsymbol{\theta}_{t+1}^j + \eta\boldsymbol{\phi}_t^j{}^\top(\nabla_{\boldsymbol{\theta}_t^j}\widehat{\mathcal{R}}(\boldsymbol{\theta}_t^j) - \mathbf{g}_t^j)$$
$$\quad + \eta(\boldsymbol{\phi}_0^j{}^\top\nabla_{\boldsymbol{\theta}_t^j}\widehat{\mathcal{R}}^{\mathrm{rf},j}(\boldsymbol{\theta}_t^j) - \boldsymbol{\phi}_t^j{}^\top\nabla_{\boldsymbol{\theta}_t^j}\widehat{\mathcal{R}}^j(\boldsymbol{\theta}_t^j)).$$

Define $r_t(\boldsymbol{\alpha}) = h_t^j(\boldsymbol{\alpha}) - w_*(\boldsymbol{\alpha})(j)$, and $r_t^{\mathrm{rf}}(\boldsymbol{\alpha}) = h_t^{\mathrm{rf},j}(\boldsymbol{\alpha}) - w_*(\boldsymbol{\alpha})(j)$. Notice that

$$\boldsymbol{\phi}_0^j{}^\top\nabla_{\boldsymbol{\theta}_t^j}\widehat{\mathcal{R}}^{\mathrm{rf},j}(\boldsymbol{\theta}_t^j) - \boldsymbol{\phi}_t^j{}^\top\nabla_{\boldsymbol{\theta}_t^j}\widehat{\mathcal{R}}^j(\boldsymbol{\theta}_t^j) = \boldsymbol{\phi}_0^j{}^\top(\boldsymbol{\alpha})\frac{2}{n}\sum_{i=1}^n\boldsymbol{\phi}_0^j(\boldsymbol{\alpha}_i)r_t^{\mathrm{rf}}(\boldsymbol{\alpha}_i) - \boldsymbol{\phi}_t^j{}^\top(\boldsymbol{\alpha})\frac{2}{n}\sum_{i=1}^n\boldsymbol{\phi}_t(\boldsymbol{\alpha}_i)r_t(\boldsymbol{\alpha}_i)$$

$$= -\boldsymbol{\phi}_0^j{}^\top(\boldsymbol{\alpha})\frac{2}{n}\sum_{i=1}^n\boldsymbol{\phi}_0(\boldsymbol{\alpha}_i)(h_t(\boldsymbol{\alpha}_i) - h_t^{\mathrm{rf}}(\boldsymbol{\alpha}_i))$$

$$+ \left(\boldsymbol{\phi}_0^j{}^\top(\boldsymbol{\alpha})\frac{2}{n}\sum_{i=1}^n\boldsymbol{\phi}_0^j(\boldsymbol{\alpha}_i) - \boldsymbol{\phi}_t^j{}^\top(\boldsymbol{\alpha})\frac{2}{n}\sum_{i=1}^n\boldsymbol{\phi}_t^j(\boldsymbol{\alpha}_i)\right)r_t(\boldsymbol{\alpha}_i)$$

We define the following stacked vector representation :

$$\mathbf{h}_{t+1}^j := [h_{t+1}^j{}^\top(\boldsymbol{\alpha}_1), ..., h_{t+1}^j{}^\top(\boldsymbol{\alpha}_n)]^\top \in \mathbb{R}^n,$$
$$\mathbf{r}_t := [r_t^\top(\boldsymbol{\alpha}_1), ..., r_t^\top(\boldsymbol{\alpha}_n)]^\top \in \mathbb{R}^n.$$

Hence we can write the pointwise difference into the following compact form:

$$\mathbf{h}_{t+1}^j - \mathbf{h}_{t+1}^{\mathrm{rf},j} = \left(\mathbf{I} - \frac{2\eta}{n}\boldsymbol{\Phi}_0^j{}^\top\boldsymbol{\Phi}_0^j\right)(\mathbf{h}_t^j - \mathbf{h}_t^{\mathrm{rf},j})$$
$$+ \underbrace{(\boldsymbol{\Phi}_{t+1}^j{}^\top - \boldsymbol{\Phi}_t^j{}^\top)\boldsymbol{\theta}_{t+1}^j}_{T_1}$$
$$+ \eta\,\underbrace{\boldsymbol{\Phi}_t^j{}^\top(\nabla_{\boldsymbol{\theta}_t^j}\widehat{\mathcal{R}}^j(\boldsymbol{\theta}_t^j) - \mathbf{g}_t^j)}_{T_2}$$
$$+ \eta\,\underbrace{\frac{2}{n}\left(\boldsymbol{\Phi}_0^j{}^\top\boldsymbol{\Phi}_0^j - \boldsymbol{\Phi}_t^j{}^\top\boldsymbol{\Phi}_t^j\right)\mathbf{r}_t}_{T_3}.$$

Unrolling the recursion to $t = 0$ yields:

$$\mathbf{h}_{t+1}^j - \mathbf{h}_{t+1}^{\mathrm{rf},j} = \sum_{s=0}^t (\mathbf{I} - \frac{2\eta}{n}\hat{\mathbf{K}}_0^j)^{t-s}(\boldsymbol{\Phi}_{s+1}^j{}^\top - \boldsymbol{\Phi}_s^j{}^\top)\boldsymbol{\theta}_{t+1}^j + \eta\sum_{s=0}^t(\mathbf{I} - \frac{2\eta}{n}\hat{\mathbf{K}}_0^j)^{t-s}\boldsymbol{\Phi}_s^j{}^\top(\nabla_{\boldsymbol{\theta}_t^j}\widehat{\mathcal{R}}(\boldsymbol{\theta}_s^j) - \mathbf{g}_s^j)$$
$$+ \eta\sum_{s=0}^t(\mathbf{I} - \frac{2\eta}{n}\hat{\mathbf{K}}_0^j)^{t-s}\frac{2}{n}\left(\hat{\mathbf{K}}_0^j - \hat{\mathbf{K}}_s^j\right)\mathbf{r}_s.$$

Taking spectral norm on both sides yields:

$$\left\|\mathbf{h}_{t+1}^j - \mathbf{h}_{t+1}^{\mathrm{rf},j}\right\| = \underbrace{\left\|\sum_{s=0}^t(\mathbf{I} - \frac{2\eta}{n}\hat{\mathbf{K}}_0^j)^{t-s}(\boldsymbol{\Phi}_{s+1}^j{}^\top - \boldsymbol{\Phi}_s^j{}^\top)\boldsymbol{\theta}_{t+1}^j\right\|}_{T_1}$$
$$+ \eta\,\underbrace{\left\|\sum_{s=0}^t(\mathbf{I} - \frac{2\eta}{n}\hat{\mathbf{K}}_0^j)^{t-s}\boldsymbol{\Phi}_s^j{}^\top(\nabla_{\boldsymbol{\theta}_t^j}\widehat{\mathcal{R}}^j(\boldsymbol{\theta}_s^j) - \mathbf{g}_s^j)\right\|}_{T_2}$$
$$+ \eta\,\underbrace{\left\|\sum_{s=0}^t(\mathbf{I} - \frac{2\eta}{n}\hat{\mathbf{K}}_0^j)^{t-s}\frac{2}{n}\left(\hat{\mathbf{K}}_0^j - \hat{\mathbf{K}}_s^j\right)\mathbf{r}_s\right\|}_{T_3}.$$

We first bound $T_1$ as follows:

$$\left\|\sum_{s=0}^t\left(\mathbf{I} - \frac{2\eta}{n}\hat{\mathbf{K}}_0^j\right)^{t-s}(\boldsymbol{\Phi}_{s+1}^j{}^\top - \boldsymbol{\Phi}_s^j{}^\top)\boldsymbol{\theta}_{t+1}^j\right\| \leq \sum_{s=0}^t\left\|(\mathbf{I} - \frac{2\eta}{n}\hat{\mathbf{K}}_0^j)^{t-s}\right\|\left\|(\boldsymbol{\Phi}_{s+1}^j{}^\top - \boldsymbol{\Phi}_s^j{}^\top)\boldsymbol{\theta}_{t+1}^j\right\|$$
$$\leq \sum_{s=0}^t\left(1 - \frac{2\eta}{n}\lambda_{\min}(\hat{\mathbf{K}}_0^j)\right)^{t-s}\left\|(\boldsymbol{\Phi}_{s+1}^j{}^\top - \boldsymbol{\Phi}_s^j{}^\top)\boldsymbol{\theta}_{t+1}^j\right\|$$
$$\leq \sum_{s=0}^t\left(1 - \frac{\lambda_0\eta}{n}\right)^{t-s}\left\|(\boldsymbol{\Phi}_{s+1}^j{}^\top - \boldsymbol{\Phi}_s^j{}^\top)\boldsymbol{\theta}_{t+1}^j\right\|$$

where we apply triangle inequality and Cauchy-Schwartz inequality. To bound $\left\|(\boldsymbol{\Phi}_{s+1}^\top - \boldsymbol{\Phi}_s^\top)\boldsymbol{\theta}_{t+1}\right\|$, we evoke Eq.(28)):

$$\left\|(\boldsymbol{\Phi}_{s+1}^{j}{}^\top - \boldsymbol{\Phi}_s^{j}{}^\top)\boldsymbol{\theta}_{t+1}^j\right\| \leq 2\sqrt{n}\left(\frac{48nB_y}{\lambda_0} + \sqrt{\nu}\right)\left(\frac{2\eta\sqrt{\widehat{\mathcal{R}}^j(\boldsymbol{\theta}_s^j)}}{\sqrt{m}} + \frac{2\eta}{\sqrt{m}}\sqrt{(1-\mu\gamma)^{Ks}D}\right)$$

Now we plug in the convergence rate from Theorem 5, and obtain

$$\left\|(\boldsymbol{\Phi}_{s+1}^{j}{}^\top - \boldsymbol{\Phi}_s^{j}{}^\top)\boldsymbol{\theta}_{t+1}^j\right\|$$
$$\leq 2\sqrt{n}\left(\frac{48nB_y}{\lambda_0} + \sqrt{\nu}\right)\left(\frac{2\eta\sqrt{\left(1-\frac{\eta\lambda_0}{4n}\right)^s\widehat{\mathcal{R}}^j(\boldsymbol{\theta}_0^j) + \frac{4nC(1-\mu\gamma)^K D}{\eta\lambda_0}}}{\sqrt{m}} + \frac{2\eta}{\sqrt{m}}\sqrt{(1-\mu\gamma)^{Ks}D}\right)$$

Putting pieces together yields:

$$T_1 \leq 2\sqrt{n}\left(\frac{48nB_y}{\lambda_0} + \sqrt{\nu}\right)\sum_{s=0}^{t}(1-\frac{\lambda_0\eta}{n})^{t-s}$$
$$\times\left(\frac{2\eta\sqrt{\left(1-\frac{\eta\lambda_0}{4n}\right)^s\widehat{\mathcal{R}}(\boldsymbol{\theta}_0) + \frac{4nC(1-\mu\gamma)^K D}{\eta\lambda_0}}}{\sqrt{m}} + \frac{2\eta}{\sqrt{m}}\sqrt{(1-\mu\gamma)^{Ks}D}\right)$$
$$\leq \frac{4\eta\sqrt{n}}{\sqrt{m}}\left(\frac{48nB_y}{\lambda_0} + \sqrt{\nu}\right)\sum_{s=0}^{t}(1-\frac{\lambda_0\eta}{n})^{t-s}$$
$$\times\left(\sqrt{\left(1-\frac{\eta\lambda_0}{4n}\right)^s\widehat{\mathcal{R}}(\boldsymbol{\theta}_0) + \frac{4nC\left(1-\mu\gamma\right)^K D}{\eta\lambda_0}} + \sqrt{(1-\mu\gamma)^{Ks}D}\right)$$
$$\leq \frac{4\eta\sqrt{n}}{\sqrt{m}}\left(\frac{48nB_y}{\lambda_0} + \sqrt{\nu}\right)(1-\frac{\lambda_0\eta}{n})^{\frac{t}{2}}$$
$$\times\sum_{s=0}^{t}(1-\frac{\lambda_0\eta}{n})^{\frac{t}{2}-s}\left(\left(1-\frac{\eta\lambda_0}{4n}\right)^{\frac{s}{2}}\sqrt{\widehat{\mathcal{R}}(\boldsymbol{\theta}_0)} + \sqrt{\frac{4nC\left(1-\mu\gamma\right)^K D}{\eta\lambda_0}} + (1-\mu\gamma)^{\frac{s}{2}}\sqrt{(1-\mu\gamma)^K D}\right).$$

Using the fact that $1-\frac{\lambda_0\eta}{n} \leq 1-\frac{\lambda_0\eta}{4n}$ and our choice of learning rate such that $1-\mu\gamma \leq 1-\frac{\lambda_0\eta}{4n}$, we have:

$$T_1 \leq \frac{\sqrt{n}4\eta}{\sqrt{m}}\left(\frac{48nB_y}{\lambda_0} + \sqrt{\nu}\right)$$
$$\times\left((1-\frac{\lambda_0\eta}{n})^{\frac{t}{2}}\frac{8n}{\lambda_0\eta}\sqrt{\widehat{\mathcal{R}}^j(\boldsymbol{\theta}_0^j)} + \frac{4n}{\lambda_0\eta}\sqrt{\frac{4nC\left(1-\mu\gamma\right)^K D}{\eta\lambda_0}} + (1-\frac{\lambda_0\eta}{n})^{\frac{t}{2}}\frac{8n}{\lambda_0\eta}\sqrt{(1-\mu\gamma)^K D}\right).$$

For $T_2$, similarly, we apply triangle and Cauchy-Schwartz inequality:

$$T_2 \leq \eta\sum_{s=0}^{t}\left(1-\frac{\eta\lambda_0}{n}\right)^{t-s}\left\|\boldsymbol{\Phi}_s^{j}{}^\top(\nabla_{\boldsymbol{\theta}_t^j}\widehat{\mathcal{R}}^j(\boldsymbol{\theta}_s^j) - \mathbf{g}_s^j)\right\|$$

To bound $\left\|\boldsymbol{\Phi}_s^{j}{}^\top(\nabla_{\boldsymbol{\theta}_t^j}\widehat{\mathcal{R}}^j(\boldsymbol{\theta}_s^j) - \mathbf{g}_s)\right\|$, we evoke Lemma 11:

$$\left\| \mathbf{\Phi}_s^{j\top} (\nabla_{\boldsymbol{\theta}_t^j} \widehat{\mathcal{R}}^j(\boldsymbol{\theta}_s^j) - \mathbf{g}_s^j) \right\| \leq \frac{2}{n} \left( \left\| \hat{\mathbf{K}}_0^j \right\| + 2n^2(\rho + \sqrt{\frac{\nu}{m}}) \right) \sqrt{(1 - \mu\gamma)^{Ks} D}.$$

Putting the above inequality back to $T_2$ yields:

$$T_2 \leq \eta \sum_{s=0}^{t} \left( 1 - \frac{\eta\lambda_0}{n} \right)^{t-s} \frac{2}{n} \left( \frac{3}{2} \|\mathbf{K}\| + 2n^2(\rho + \sqrt{\frac{\nu}{m}}) \right) \sqrt{(1 - \mu\gamma)^{Ks} D}$$

$$\leq \eta \left( 1 - \frac{\eta\lambda_0}{4n} \right)^{\frac{t}{2}} \sum_{s=0}^{t} \left( 1 - \frac{\eta\lambda_0}{4n} \right)^{\frac{t}{2} - \frac{s}{2}} \frac{2}{n} \left( \frac{3}{2} \|\mathbf{K}\| + 2n^2 \left( \frac{48nB_y}{\lambda_0\sqrt{m}} + \sqrt{\frac{\nu}{m}} \right) \right) \sqrt{(1 - \mu\gamma)^K D}$$

$$\leq \left( 1 - \frac{\eta\lambda_0}{4n} \right)^{\frac{t}{2}} \frac{16}{\lambda_0} \left( \frac{3}{2} \|\mathbf{K}\| + 2n^2 \left( \frac{48nB_y}{\lambda_0\sqrt{m}} + \sqrt{\frac{\nu}{m}} \right) \right) \sqrt{(1 - \mu\gamma)^K D}$$

For $T_3$, we also apply Cauchy-Schwartz and get:

$$T_3 \leq \eta \sum_{s=0}^{t} (1 - \frac{\eta\lambda_0}{n})^{t-s} \left\| \frac{2}{n} \left( \hat{\mathbf{K}}_0^j - \hat{\mathbf{K}}_s^j \right) \mathbf{r}_s \right\|$$

$$\leq \eta \frac{2}{n} \sum_{s=0}^{t} (1 - \frac{\eta\lambda_0}{n})^{t-s} \left\| \left( \hat{\mathbf{K}}_0^j - \hat{\mathbf{K}}_s^j \right) \right\| \|\mathbf{r}_s\|$$

$$\leq \eta \frac{2}{n} \sum_{s=0}^{t} (1 - \frac{\eta\lambda_0}{n})^{t-s} 2n^2 (\rho + \sqrt{\frac{\nu}{m}}) \sqrt{n} \sqrt{\widehat{\mathcal{R}}^j(\boldsymbol{\theta}_t^j)}$$

where at last step we plug in empirical gram matrix bound from Lemma 10. Now it suffices to plug in bound for $\widehat{\mathcal{R}}(\boldsymbol{\theta}_t)$

$$T_3 \leq \eta \frac{2}{n} \sum_{s=0}^{t} (1 - \frac{\eta\lambda_0}{n})^{t-s} 2n^2 (\rho + \sqrt{\frac{\nu}{m}}) \sqrt{n} \sqrt{\left( 1 - \frac{\eta\lambda_0}{4n} \right)^s \widehat{\mathcal{R}}^j(\boldsymbol{\theta}_0^j) + \frac{4n(1 - \mu\gamma)^K D}{\eta\lambda_0}}$$

$$\leq \eta \frac{2}{n} 2n^2 (\rho + \sqrt{\frac{\nu}{m}}) \sqrt{n} \left( (1 - \frac{\eta\lambda_0}{n})^{\frac{t}{2}} \sum_{s=0}^{t} (1 - \frac{\eta\lambda_0}{n})^{\frac{t}{2} - \frac{s}{2}} \sqrt{\widehat{\mathcal{R}}^j(\boldsymbol{\theta}_0^j)} + \frac{8n}{\eta\lambda_0} \sqrt{\frac{4n(1 - \mu\gamma)^K D}{\eta\lambda_0}} \right)$$

$$\leq 4\eta\sqrt{n}n \left( \frac{48nB_y}{\lambda_0\sqrt{m}} + \sqrt{\frac{\nu}{m}} \right) \left( (1 - \frac{\eta\lambda_0}{4n})^{\frac{t}{2}} \frac{8n}{\eta\lambda_0} \sqrt{\widehat{\mathcal{R}}^j(\boldsymbol{\theta}_0^j)} + \frac{8n}{\eta\lambda_0} \sqrt{\frac{4n(1 - \mu\gamma)^K D}{\eta\lambda_0}} \right).$$

Putting pieces together yields:

$$\left\| \mathbf{h}_{t+1}^j - \mathbf{h}_{t+1}^{\text{rf},j} \right\| \leq \frac{\sqrt{n}4\eta}{\sqrt{m}} \left( \frac{48nB_y}{\lambda_0} + \sqrt{\nu} \right)$$

$$\times \left( (1 - \frac{\lambda_0\eta}{4n})^{\frac{t}{2}} \frac{8n}{\lambda_0\eta} \sqrt{\widehat{\mathcal{R}}^j(\boldsymbol{\theta}_0^j)} + \frac{4n}{\lambda_0\eta} \sqrt{\frac{4nC(1 - \mu\gamma)^K D}{\eta\lambda_0}} + (1 - \frac{\lambda_0\eta}{4n})^{\frac{t}{2}} \frac{8n}{\lambda_0\eta} \sqrt{(1 - \mu\gamma)^K D} \right)$$

$$+ \left( 1 - \frac{\eta\lambda_0}{4n} \right)^{\frac{t}{2}} \frac{16}{\lambda_0} \left( \frac{3}{2} \|\mathbf{K}\| + 2n^2 \left( \frac{48nB_y}{\lambda_0\sqrt{m}} + \sqrt{\frac{\nu}{m}} \right) \right) \sqrt{(1 - \mu\gamma)^K D}$$

$$+ 4\eta\sqrt{n}n \left( \frac{48nB_y}{\lambda_0\sqrt{m}} + \sqrt{\frac{\nu}{m}} \right) \left( (1 - \frac{\eta\lambda_0}{4n})^{\frac{t}{2}} \frac{8n}{\eta\lambda_0} \sqrt{\widehat{\mathcal{R}}^j(\boldsymbol{\theta}_0^j)} + \frac{8n}{\eta\lambda_0} \sqrt{\frac{4n(1 - \mu\gamma)^K D}{\eta\lambda_0}} \right).$$

Let $\tilde{n} := \left( \frac{48nB_y}{\lambda_0\sqrt{m}} + \sqrt{\frac{\nu}{m}} \right)$

$$\left\| \mathbf{h}_{t+1}^{j} - \mathbf{h}_{t+1}^{\mathrm{rf},j} \right\| \leq 4\sqrt{n}\tilde{n}\Bigg( \left(1 - \frac{\eta\lambda_0}{4n}\right)^{\frac{t}{2}} \frac{8n}{\lambda_0}\sqrt{\widehat{\mathcal{R}}^j(\boldsymbol{\theta}_0^j)}$$

$$+ \frac{4n}{\lambda_0}\sqrt{\frac{4nC\left(1-\mu\gamma\right)^K D}{\eta\lambda_0}} + \left(1 - \frac{\eta\lambda_0}{4n}\right)^{\frac{t}{2}} \frac{8n}{\lambda_0}\sqrt{(1-\mu\gamma)^K D}\Bigg)$$

$$+ \left(1 - \frac{\eta\lambda_0}{4n}\right)^{\frac{t}{2}} \frac{16}{\lambda_0}\left(\frac{3}{2}\|\mathbf{K}\| + 2n^2\tilde{n}\right)\sqrt{(1-\mu\gamma)^K D}$$

$$+ 4\sqrt{n}n\tilde{n}\Bigg( \left(1 - \frac{\eta\lambda_0}{4n}\right)^{\frac{t}{2}} \frac{8n}{\lambda_0}\sqrt{\widehat{\mathcal{R}}^j(\boldsymbol{\theta}_0^j)} + \frac{8n}{\lambda_0}\sqrt{\frac{4nC\left(1-\mu\gamma\right)^K D}{\eta\lambda_0}}\Bigg)$$

$$\leq \left(1 - \frac{\eta\lambda_0}{4n}\right)^{\frac{t}{2}} \frac{16}{\lambda_0}\left(\left(\frac{3}{2}\|\mathbf{K}\| + 2n^2\tilde{n} + 2\sqrt{n}n\tilde{n}\right)\sqrt{(1-\mu\gamma)^K D} + \left(2\sqrt{n}n^2\tilde{n} + 2\sqrt{n}n\tilde{n}\right)\sqrt{\widehat{\mathcal{R}}(\boldsymbol{\theta}_0)}\right)$$

$$+ \left(2\sqrt{n}\tilde{n} + 4\sqrt{n}n\tilde{n}\right)\frac{8n}{\lambda_0}\sqrt{\frac{4nC\left(1-\mu\gamma\right)^K D}{\eta\lambda_0}}.$$

$\square$

**Lemma 13** (Neural network parameter and RF parameter coupling). *For Algorithm 2, the following statement holds for any $j \in [d]$ and $t \in [T]$:*

$$\left\| \boldsymbol{\theta}_t^j - \bar{\boldsymbol{\theta}}_t^j \right\|^2 \leq O\left(\frac{n^2}{\lambda_0^2}\left(\frac{48nB_y}{\lambda_0\sqrt{m}} + \sqrt{\frac{\nu}{m}}\right) + \frac{n^6}{\lambda_0^4}\left(\frac{48nB_y}{\lambda_0\sqrt{m}} + \sqrt{\frac{\nu}{m}}\right)^2\right)$$

$$+ O\left(\eta^3 T^2 \left(\frac{48nB_y}{\lambda_0\sqrt{m}} + \sqrt{\frac{\nu}{m}}\right)^2 \frac{n^3(1-\frac{1}{\kappa})^K D}{\lambda_0^3}(\lambda_{\max} + \lambda_0)^2 + \frac{(1-\frac{1}{\kappa})^K D}{\lambda_0^2}\right).$$

*Proof.* According to updating rule of neural network and random feature predictor, we have

$$\boldsymbol{\theta}_{t+1}^j - \bar{\boldsymbol{\theta}}_{t+1}^j$$
$$= \boldsymbol{\theta}_t^j - \bar{\boldsymbol{\theta}}_t^j - \eta(\mathbf{g}_t^j - \nabla_{\bar{\boldsymbol{\theta}}_t^j}\mathcal{R}^{\mathrm{rf},j}(\bar{\boldsymbol{\theta}}_t^j))$$
$$= \boldsymbol{\theta}_t^j - \bar{\boldsymbol{\theta}}_t^j - \eta\frac{2}{n}\sum_{i=1}^n(\boldsymbol{\phi}_t^j(\boldsymbol{\alpha}_i)\mathbf{r}_t^j(\boldsymbol{\alpha}_i) - \boldsymbol{\phi}_0^j(\boldsymbol{\alpha}_i)\mathbf{r}_t^{\mathrm{rf},j}(\boldsymbol{\alpha}_i)) - \eta(\mathbf{g}_t^j - \nabla_{\boldsymbol{\theta}_t^j}\mathcal{R}^j(\bar{\boldsymbol{\theta}}_t^j))$$
$$= \boldsymbol{\theta}_t^j - \bar{\boldsymbol{\theta}}_t^j - \eta\frac{2}{n}\sum_{i=1}^n(\boldsymbol{\phi}_t^j(\boldsymbol{\alpha}_i)\mathbf{r}_t^j(\boldsymbol{\alpha}_i) - \boldsymbol{\phi}_0(\boldsymbol{\alpha}_i)\mathbf{r}_t^{\mathrm{rf},j}(\boldsymbol{\alpha}_i)) - \eta(\mathbf{g}_t^j - \nabla_{\boldsymbol{\theta}_t^j}\mathcal{R}^j(\bar{\boldsymbol{\theta}}_t^j))$$
$$= \boldsymbol{\theta}_t^j - \bar{\boldsymbol{\theta}}_t^j - \eta\frac{2}{n}\sum_{i=1}^n(\boldsymbol{\phi}_t^j(\boldsymbol{\alpha}_i) - \boldsymbol{\phi}_0^j(\boldsymbol{\alpha}_i))\mathbf{r}_t(\boldsymbol{\alpha}_i) - \eta\frac{2}{n}\sum_{i=1}^n\boldsymbol{\phi}_0^j(\boldsymbol{\alpha}_i)(\mathbf{r}_t(\boldsymbol{\alpha}_i) - \mathbf{r}_t^{\mathrm{rf}}(\boldsymbol{\alpha}_i)) - \eta(\mathbf{g}_t^j - \nabla_{\boldsymbol{\theta}_t^j}\mathcal{R}(\bar{\boldsymbol{\theta}}_t^j))$$
$$= \boldsymbol{\theta}_t^j - \bar{\boldsymbol{\theta}}_t^j - \eta\frac{2}{n}(\boldsymbol{\Phi}_t^j(\boldsymbol{\alpha}_i) - \boldsymbol{\Phi}_0^j(\boldsymbol{\alpha}_i))\mathbf{r}_t - \eta\frac{2}{n}\boldsymbol{\Phi}_0^j(\boldsymbol{\alpha}_i)(\mathbf{h}_t^j - \mathbf{h}_t^{\mathrm{rf},j}) - \eta(\mathbf{g}_t^j - \nabla_{\boldsymbol{\theta}_t^j}\mathcal{R}^j(\bar{\boldsymbol{\theta}}_t^j)).$$

Now taking norm on both side yields:

$$\left\| \boldsymbol{\theta}_{t+1}^j - \bar{\boldsymbol{\theta}}_{t+1}^j \right\| = \left\| \boldsymbol{\theta}_t^j - \bar{\boldsymbol{\theta}}_t^j \right\| + \eta\frac{2}{n}\underbrace{\left\| (\boldsymbol{\Phi}_t^j - \boldsymbol{\Phi}_0^j)\mathbf{r}_t^j \right\|}_{T_1} + \eta\frac{2}{n}\underbrace{\left\| \boldsymbol{\Phi}_0^j(\boldsymbol{\alpha}_i)(\mathbf{h}_t^j - \mathbf{h}_t^{\mathrm{rf},j}) \right\|}_{T_2} + \eta\underbrace{\left\| \mathbf{g}_t^j - \nabla_{\boldsymbol{\theta}_t^j}\mathcal{R}(\bar{\boldsymbol{\theta}}_t^j) \right\|}_{T_3}.$$

For $T_1$:

$$T_1 \leq \left\|(\mathbf{\Phi}_t^j - \mathbf{\Phi}_0^j)\right\| \left\|\mathbf{r}_t^j\right\|$$

$$\leq \sqrt{n(\rho + \sqrt{\frac{\nu}{m}})}\sqrt{n}\sqrt{\widehat{\mathcal{R}}^j(\boldsymbol{\theta}_t^j)}$$

$$\leq \sqrt{n(\rho + \sqrt{\frac{\nu}{m}})}\sqrt{n}\sqrt{\left(1 - \frac{\eta\lambda_0}{4n}\right)^t \widehat{\mathcal{R}}^j(\boldsymbol{\theta}_0^j) + \frac{4n(1-\mu\gamma)^K D}{\eta\lambda_0}}.$$

where we plug in results from Lemma 10 and Theorem 5.

For $T_2$:

$$T_2 \leq \left\|\mathbf{\Phi}_0^j\right\| \left\|\mathbf{h}_t^j - \mathbf{h}_t^{\mathrm{rf},j}\right\|$$

$$\leq \sqrt{n}\left(1 - \frac{\eta\lambda_0}{4n}\right)^{\frac{t}{2}} \frac{16}{\lambda_0}\left(\left(\frac{3}{2}\|\mathbf{K}\| + 2n^2\tilde{n} + 2\sqrt{n}n\tilde{n}\right)\sqrt{(1-\mu\gamma)^K D} + \left(2\sqrt{n}n^2\tilde{n} + 2\sqrt{n}n\tilde{n}\right)\sqrt{\widehat{\mathcal{R}}(\boldsymbol{\theta}_0)}\right)$$

$$+ \sqrt{n}\left(2\sqrt{n}\tilde{n} + 4\sqrt{n}n\tilde{n}\right)\frac{8n}{\lambda_0}\sqrt{\frac{4nC(1-\mu\gamma)^K D}{\eta\lambda_0}}.$$

For $T_3$, we evoke Lemma 11:

$$T_3 \leq 2\sqrt{(1-\mu\gamma)^{Kt}D} \leq 2\sqrt{(1 - \frac{n\lambda_0}{4n})^t(1-\mu\gamma)^K D}.$$

Putting pieces together we have:

$$\left\|\boldsymbol{\theta}_{t+1} - \bar{\boldsymbol{\theta}}_{t+1}\right\|$$

$$= \left\|\boldsymbol{\theta}_t^j - \bar{\boldsymbol{\theta}}_t^j\right\| + \frac{2\eta}{n}\sqrt{n(\rho + \sqrt{\frac{\nu}{m}})}\sqrt{n}\sqrt{\left(1 - \frac{\eta\lambda_0}{4n}\right)^t \widehat{\mathcal{R}}(\boldsymbol{\theta}_0) + \frac{4n(1-\mu\gamma)^K D}{\eta\lambda_0}}$$

$$+ \frac{2\eta}{n}\sqrt{n}\left(1 - \frac{\eta\lambda_0}{4n}\right)^{\frac{t}{2}} \frac{16}{\lambda_0}$$

$$\times \left(\left(\frac{3}{2}\|\mathbf{K}\| + 2n^2\tilde{n} + 2\sqrt{n}n\tilde{n}\right)\sqrt{(1-\mu\gamma)^K D} + \left(2\sqrt{n}n^2\tilde{n} + 2\sqrt{n}n\tilde{n}\right)\sqrt{\widehat{\mathcal{R}}(\boldsymbol{\theta}_0)}\right)$$

$$+ \frac{2\eta}{n}\sqrt{n}\left(2\sqrt{n}\tilde{n} + 4\sqrt{n}n\tilde{n}\right)\frac{8n}{\lambda_0}\sqrt{\frac{4nC(1-\mu\gamma)^K D}{\eta\lambda_0}}$$

$$+ \frac{2\eta}{n}2\sqrt{(1 - \frac{\eta\lambda_0}{4n})^t(1-\mu\gamma)^K D}.$$

Conducting telescoping sum yields:

$$\left\|\boldsymbol{\theta}_{t+1} - \bar{\boldsymbol{\theta}}_{t+1}\right\|$$

$$= \frac{2\eta}{n}\sum_{s=0}^{t+1} n\sqrt{(\rho + \sqrt{\frac{\nu}{m}})}\sqrt{\left(1 - \frac{\eta\lambda_0}{4n}\right)^s \widehat{\mathcal{R}}^j(\boldsymbol{\theta}_0^j) + \frac{4nC\left(1-\mu\gamma\right)^K D}{\eta\lambda_0}}$$

$$+ \frac{2\eta}{n}\sum_{s=0}^{t+1}\sqrt{n}\left(1 - \frac{\eta\lambda_0}{4n}\right)^{\frac{s}{2}}\frac{16}{\lambda_0}$$

$$\times \left(\left(\frac{3}{2}\|\mathbf{K}\| + 2n^2\tilde{n} + 2\sqrt{n}n\tilde{n}\right)\sqrt{(1-\mu\gamma)^K D} + \left(2\sqrt{n}n^2\tilde{n} + 2\sqrt{n}n\tilde{n}\right)\sqrt{\widehat{\mathcal{R}}(\boldsymbol{\theta}_0)}\right)$$

$$+ \frac{2\eta}{n}\sum_{s=0}^{t+1}\left(2n\tilde{n} + 4n^2\tilde{n}\right)\frac{8n}{\lambda_0}\sqrt{\frac{4nC\left(1-\mu\gamma\right)^K D}{\eta\lambda_0}}$$

$$+ \frac{2\eta}{n}\sum_{s=0}^{t+1} 2\sqrt{(1-\frac{\eta\lambda_0}{4n})^s(1-\mu\gamma)^K D}$$

$$\leq n\sqrt{\tilde{n}}\left(\frac{16}{\lambda_0}\sqrt{\widehat{\mathcal{R}}(\boldsymbol{\theta}_0)}\right) + \eta T\left(\sqrt{\tilde{n}} + \left(2n\tilde{n} + 4n^2\tilde{n}\right)\frac{16}{\lambda_0}\right)\sqrt{\frac{4nC\left(1-\mu\gamma\right)^K D}{\eta\lambda_0}}$$

$$+ \frac{256\sqrt{n}}{\lambda_0^2}\left(\left(\frac{3}{2}\|\mathbf{K}\| + 2n^2\tilde{n} + 2\sqrt{n}n\tilde{n}\right)\sqrt{(1-\mu\gamma)^K D} + \left(2\sqrt{n}n^2\tilde{n} + 2\sqrt{n}n\tilde{n}\right)\sqrt{\widehat{\mathcal{R}}(\boldsymbol{\theta}_0)}\right)$$

$$+ \frac{32}{\lambda_0}\sqrt{(1-\mu\gamma)^K D}$$

Recall that $\tilde{n} = \left(\frac{48nB_y}{\lambda_0\sqrt{m}} + \sqrt{\frac{\nu}{m}}\right)$ and $C = \frac{\lambda_0^2 n^2}{24n^4} + \frac{4\eta^2\left(\frac{1}{2}\|\mathbf{K}\| + \frac{\lambda_0}{8}\right)^2}{\frac{3}{8}n^2}$, we conclude that:

$$\left\|\boldsymbol{\theta}_{t+1} - \bar{\boldsymbol{\theta}}_{t+1}\right\|^2$$

$$\leq O\left(\frac{n^2}{\lambda_0^2}\left(\frac{48nB_y}{\lambda_0\sqrt{m}} + \sqrt{\frac{\nu}{m}}\right) + \eta^2 T^2\left(\frac{48nB_y}{\lambda_0\sqrt{m}} + \sqrt{\frac{\nu}{m}}\right)^2\frac{n^4}{\lambda_0^2}\frac{n(1-\frac{1}{\kappa})^K D}{\eta\lambda_0}\frac{\eta^2(\lambda_{\max} + \lambda_0)^2}{n^2}\right)$$

$$+ O\left(\frac{n}{\lambda_0^4}n^4\left(\frac{48nB_y}{\lambda_0\sqrt{m}} + \sqrt{\frac{\nu}{m}}\right)^2(1-\frac{1}{\kappa})^K D + \frac{n}{\lambda_0^4}n^5\left(\frac{48nB_y}{\lambda_0\sqrt{m}} + \sqrt{\frac{\nu}{m}}\right)^2 + \frac{(1-\frac{1}{\kappa})^K D}{\lambda_0^2}\right)$$

$$= O\left(\frac{n^2}{\lambda_0^2}\left(\frac{48nB_y}{\lambda_0\sqrt{m}} + \sqrt{\frac{\nu}{m}}\right) + \eta^3 T^2\left(\frac{48nB_y}{\lambda_0\sqrt{m}} + \sqrt{\frac{\nu}{m}}\right)^2\frac{n^3(1-\frac{1}{\kappa})^K D}{\lambda_0^3}(\lambda_{\max} + \lambda_0)^2\right)$$

$$+ O\left(\frac{n^6}{\lambda_0^4}\left(\frac{48nB_y}{\lambda_0\sqrt{m}} + \sqrt{\frac{\nu}{m}}\right)^2 + \frac{(1-\frac{1}{\kappa})^K D}{\lambda_0^2}\right)$$

$$\square$$

**Theorem 6.** *For Algorithm 2, the following statement holds true for any $j \in [d]$, $t \in [T]$:*

$$\sup_{\boldsymbol{\alpha}\in\Delta^N}\left(h_T^j(\boldsymbol{\alpha}) - h_T^{\mathrm{kls},j}(\boldsymbol{\alpha})\right)^2$$

$$\leq 2\left(\frac{48nB_y}{\lambda_0\sqrt{m}}\right)^2(\frac{48nB_y}{\lambda_0} + \sqrt{\nu})^2 + O\left(\frac{n^2}{\lambda_0^2}\left(\frac{48nB_y}{\lambda_0\sqrt{m}} + \sqrt{\frac{\nu}{m}}\right) + \frac{n^6}{\lambda_0^4}\left(\frac{48nB_y}{\lambda_0\sqrt{m}} + \sqrt{\frac{\nu}{m}}\right)^2\right)$$

$$+ O\left(\eta^3 T^2\left(\frac{48nB_y}{\lambda_0\sqrt{m}} + \sqrt{\frac{\nu}{m}}\right)^2\frac{n^3(1-\frac{1}{\kappa})^K D}{\lambda_0^3}(\lambda_{\max} + \lambda_0)^2 + \frac{(1-\frac{1}{\kappa})^K D}{\lambda_0^2}\right)$$

*If we choose $K \geq \kappa\log\left(\frac{\eta^3 T^2 n^3 D}{\lambda_0^3}\right)$, then we have:*

$$\sup_{\boldsymbol{\alpha}\in\Delta^N}\left(h_T^j(\boldsymbol{\alpha}) - h_T^{\mathrm{kls},j}(\boldsymbol{\alpha})\right)^2 \leq O\left(\max\left\{\frac{\mathrm{poly}_3(n)}{\sqrt{m}}, \frac{\mathrm{poly}_8(n)}{m}\right\}\right).$$

*Proof.* We consider the following canonical decomposition:

$$\left(h^j(\boldsymbol{\alpha}) - h^{\mathrm{kls},j}(\boldsymbol{\alpha})\right)^2 \le 2\left(h_t^j(\boldsymbol{\alpha}) - h_t^{\mathrm{rf},j}(\boldsymbol{\alpha})\right)^2 + 2\left(h_t^{\mathrm{rf},j}(\boldsymbol{\alpha}) - h_t^{\mathrm{kls},j}(\boldsymbol{\alpha})\right)^2$$

For the first term, we

$$\left\|h_t^j(\boldsymbol{\alpha}) - h_t^{\mathrm{rf},j}(\boldsymbol{\alpha})\right\|^2$$

$$\le 2\left\|\boldsymbol{\phi}_t^j(\boldsymbol{\alpha})^\top \boldsymbol{\theta}_t^j - \boldsymbol{\phi}_0^j(\boldsymbol{\alpha})^\top \boldsymbol{\theta}_t^j\right\|^2 + 2\left\|\boldsymbol{\phi}_0^j(\boldsymbol{\alpha})^\top (\boldsymbol{\theta}_t^j - \bar{\boldsymbol{\theta}}_t^j)\right\|^2$$

$$\le 2\rho^2(\sqrt{m}\rho + \sqrt{\nu})^2 + O\left(\frac{n^2}{\lambda_0^2}\left(\frac{48nB_y}{\lambda_0\sqrt{m}} + \sqrt{\frac{\nu}{m}}\right) + \frac{n^6}{\lambda_0^4}\left(\frac{48nB_y}{\lambda_0\sqrt{m}} + \sqrt{\frac{\nu}{m}}\right)^2\right)$$

$$+ O\left(\eta^3 T^2 \left(\frac{48nB_y}{\lambda_0\sqrt{m}} + \sqrt{\frac{\nu}{m}}\right)^2 \frac{n^3(1-\frac{1}{\kappa})^K D}{\lambda_0^3}(\lambda_{\max} + \lambda_0)^2 + \frac{(1-\frac{1}{\kappa})^K D}{\lambda_0^2}\right)$$

where we plug in Proposition 5 (e) to bound $\left\|\boldsymbol{\phi}_t^j(\boldsymbol{\alpha})^\top \boldsymbol{\theta}_t^j - \boldsymbol{\phi}_0^j(\boldsymbol{\alpha})^\top \boldsymbol{\theta}_t^j\right\|$, and use Lemma 13 to bound $\left\|\boldsymbol{\phi}_0^j(\boldsymbol{\alpha})^\top (\boldsymbol{\theta}_t^j - \bar{\boldsymbol{\theta}}_t^j)\right\|$. Plugging in $\rho = \frac{48nB_y}{\lambda_0\sqrt{m}}$ will complete bounding first term:

$$\left\|h_t^j(\boldsymbol{\alpha}) - h_t^{\mathrm{rf},j}(\boldsymbol{\alpha})\right\|^2$$

$$\le 2\left(\frac{48nB_y}{\lambda_0\sqrt{m}}\right)^2 (\frac{48nB_y}{\lambda_0} + \sqrt{\nu})^2 + O\left(\frac{n^2}{\lambda_0^2}\left(\frac{48nB_y}{\lambda_0\sqrt{m}} + \sqrt{\frac{\nu}{m}}\right) + \frac{n^6}{\lambda_0^4}\left(\frac{48nB_y}{\lambda_0\sqrt{m}} + \sqrt{\frac{\nu}{m}}\right)^2\right)$$

$$+ O\left(\eta^3 T^2 \left(\frac{48nB_y}{\lambda_0\sqrt{m}} + \sqrt{\frac{\nu}{m}}\right)^2 \frac{n^3(1-\frac{1}{\kappa})^K D}{\lambda_0^3}(\lambda_{\max} + \lambda_0)^2 + \frac{(1-\frac{1}{\kappa})^K D}{\lambda_0^2}\right)$$

For the second term, we use the intermediate result from [21]

$$\left\|\mathbf{h}_t^{\mathrm{rf},j}(\boldsymbol{\alpha}) - \mathbf{h}_t^{\mathrm{kls},j}(\boldsymbol{\alpha})\right\| \le \sqrt{\frac{\nu}{m}}B_y\frac{n}{\lambda_0}\left(\frac{24n}{\lambda_0} + \frac{1}{2}\right)$$

□

**Lemma 14** (Approximation of Lipschitz functions on the ball [2, Proposition 6]). *For $R$ larger than a constant $c$ that depends only on $N$, for any function $f^\star : \mathbb{R}^N \to \mathbb{R}$ such that for all $\mathbf{x}, \tilde{\mathbf{x}} \in \mathbb{B}_q^N$, $\sup_{\mathbf{x}\in\mathbb{B}_q^N} |f^\star(\mathbf{x})| \le \Lambda$ and $|f^\star(\mathbf{x}) - f^\star(\tilde{\mathbf{x}})| \le \Lambda\|\mathbf{x} - \tilde{\mathbf{x}}\|_q$, there exists $h \in \mathcal{H}$, such that $\|h\|_{\mathcal{H}}^2 \le R$ and*

$$\sup_{\mathbf{x}\in\mathbb{B}_q^d(1)} |f^\star(\mathbf{x}) - h(\mathbf{x})| \le A(R), \qquad A(R) = c\Lambda\left(\frac{\sqrt{R}}{\Lambda}\right)^{-\frac{2}{d-2}}\ln\left(\frac{\sqrt{R}}{\Lambda}\right).$$

**Theorem 7** ([32, Theorem 1]). *Let $\mathcal{F} = \left\{f : \mathbb{S}^{d-1} \to [0, b]\right\}$ for some $b < \infty$. Then with probability at least $1 - e^{-\nu}, \nu > 0$, for all $f \in \mathcal{F}$ simultaneously*

$$\|f\|_2^2 \le \|f\|_n^2 + O\left(\|f\|_n\left(\widehat{\mathfrak{R}}(\mathcal{F}) + \sqrt{\frac{b\nu}{n}}\right) + \widehat{\mathfrak{R}}^2(\mathcal{F}) + \frac{b\nu}{n}\right),$$

*where the worst-case empirical Rademacher complexity is defined as*

$$\widehat{\mathfrak{R}}(\mathcal{F}) = \sup_{\mathbf{x}_1,\ldots,\mathbf{x}_n\in\mathbb{S}^{d-1}} \mathbb{E}_{\boldsymbol{\varepsilon}} \sup_{f\in\mathcal{F}} \left|\frac{1}{n}\sum_{i=1}^n \varepsilon_i f(\mathbf{x}_i)\right|. \qquad (\varepsilon_1,\ldots,\varepsilon_n \overset{\mathrm{iid}}{\sim} \mathrm{unif}\{\pm 1\})$$

**Lemma 15.** *[Rademacher complexity of kernel predictor class [3] Lemma 22] Consider the following function class with bounded RKHS norm:*

$$\mathcal{F} = \left\{h(\mathbf{x}) = \sum_{i=1}^n c_i k(\mathbf{x}, \mathbf{x}_i) : h \in \mathcal{H}, \|h\|_{\mathcal{H}} \le B\right\}.$$

*Then its empirical Rademacher complexity is bounded by:*

$$\widehat{\mathfrak{R}}(\mathcal{F}) \le O\left(\frac{B}{\sqrt{n}}\right).$$

**Lemma 16** (Excess risk of virtual KLS predictor learnt on optimal KLS approximator). *Let $h_*^{\text{kls},j}$ be the optimal approximator of $f^{*,j}$ in RKHS, and $\tilde{h}_t^{\text{kls},j}(\boldsymbol{\alpha})$ be KLS predictor trained by GD on sample $\{\boldsymbol{\alpha}_i, h_*^{\text{kls},j}(\boldsymbol{\alpha}_i)\}_{i=1}^n$. Then the following excess risk bound holds with probability at least $1 - e^{-\nu}$:*

$$\mathbb{E}\left\|\tilde{h}_{t+1}^{\text{kls},j}(\boldsymbol{\alpha}) - h_*^{\text{kls},j}(\boldsymbol{\alpha})\right\|^2 \leq \left(1 - \frac{2\lambda_0^2 \eta}{n}\right)^t + O\left(\left(1 - \frac{2\lambda_0^2 \eta}{n}\right)^t \left(\frac{R}{\sqrt{n}} + \sqrt{\frac{\nu}{n}}\right) + \frac{R^2}{n} + \frac{\nu}{n}\right),$$

*Proof.* Given a fixed function $h_*^{\text{kls},j}$ with $\left\|h_*^{\text{kls},j}\right\|_{\mathcal{H}}^2 \leq R$, we consider the following class:

$$\widetilde{\mathcal{H}} = \left\{\tilde{h}(\mathbf{x}) = h(\mathbf{x}) - h_*^{\text{kls},j}(\mathbf{x}) : h \in \mathcal{H}, \|h\|_{\mathcal{H}} \leq B\right\}$$

According to constant shift property of Rademacher complexity [33] and Lemma 15, we know

$$\widehat{\mathfrak{R}}(\widetilde{\mathcal{H}}) \leq O\left(\frac{B}{\sqrt{n}} + \frac{\sqrt{R}}{\sqrt{n}}\right).$$

According to Theorem 7, we have

$$\mathbb{E}\left\|\tilde{h}_t^{\text{kls},j}(\boldsymbol{\alpha}) - h_*^{\text{kls},j}(\boldsymbol{\alpha})\right\|^2 \leq \frac{1}{n}\sum_{i=1}^n \left\|\tilde{h}_t^{\text{kls},j}(\boldsymbol{\alpha}_i) - h_*^{\text{kls},j}(\boldsymbol{\alpha}_i)\right\|^2$$

$$+ O\left(\frac{1}{n}\sum_{i=1}^n \left\|\tilde{h}_t^{\text{kls},j}(\boldsymbol{\alpha}_i) - h_*^{\text{kls},j}(\boldsymbol{\alpha}_i)\right\|^2 \left(\frac{B + \sqrt{R}}{\sqrt{n}} + \sqrt{\frac{\nu}{n}}\right) + \frac{(B + \sqrt{R})^2}{n} + \frac{\nu}{n}\right),$$

Now, it suffices to bound (i) empirical risk term $\frac{1}{n}\sum_{i=1}^n \left\|\tilde{h}_t^{\text{kls},j}(\boldsymbol{\alpha}_i) - h_*^{\text{kls},j}(\boldsymbol{\alpha}_i)\right\|^2$ and (ii) RKHS norm of $\tilde{h}_t^{\text{kls},j}$.

For (i), we define $\widehat{\mathcal{R}}_t = \frac{1}{n}\sum_{i=1}^n \left\|\tilde{h}_t^{\text{kls},j}(\boldsymbol{\alpha}_i) - h_*^{\text{kls},j}(\boldsymbol{\alpha}_i)\right\|^2$ and recall the updating rule of kernel OLS predictor:

$$\tilde{\mathbf{h}}_{t+1}^{\text{kls},j} - \tilde{\mathbf{h}}_*^{\text{kls},j} = \mathbf{K}\mathbf{c}_{t+1}^j - \tilde{\mathbf{h}}_*^{\text{kls},j} = \mathbf{K}\mathbf{c}_t^j - \mathbf{K}\frac{2\eta}{n}\mathbf{K}^\top(\tilde{\mathbf{h}}_t^{\text{kls},j} - \tilde{\mathbf{h}}_*^{\text{kls},j}) - \tilde{\mathbf{h}}_*^{\text{kls},j}$$

$$= \left(\mathbf{I} - \frac{2\eta}{n}\mathbf{K}^\top\mathbf{K}\right)(\tilde{\mathbf{h}}_t^{\text{kls},j} - \tilde{\mathbf{h}}_*^{\text{kls},j})$$

$$= \left(\mathbf{I} - \frac{2\eta}{n}\mathbf{K}^\top\mathbf{K}\right)^t (\tilde{\mathbf{h}}_0^{\text{kls},j} - \tilde{\mathbf{h}}_*^{\text{kls},j}).$$

Hence we know that:

$$\widehat{\mathcal{R}}_{t+1} = \left(\mathbf{I} - \frac{2\eta}{n}\mathbf{K}\mathbf{K}^\top\right)^t \widehat{\mathcal{R}}_0.$$

(ii) RKHS norm of $\tilde{h}_t^{\text{kls},j}$

$$\left\|\tilde{h}_t^{\text{kls},j}\right\|_{\mathcal{H}} = \left\|\left(\mathbf{I} - \left(\mathbf{I} - \frac{2\eta}{n}\mathbf{K}\right)^t\right)\mathbf{y}_*\right\|_{\mathbf{K}^{-1}}$$

$$= \left\|\left(\mathbf{I} - \left(\mathbf{I} - \frac{2\eta}{n}\mathbf{K}\right)^t\right)\mathbf{K}^{-1}\mathbf{K}\mathbf{c}_*\right\|$$

$$\leq \left\|\mathbf{I} - \left(\mathbf{I} - \frac{2\eta}{n}\mathbf{K}\right)^t\right\|\|\mathbf{K}^{-1}\mathbf{K}\mathbf{c}_*\|$$

$$= \left\|\mathbf{I} - \left(\mathbf{I} - \frac{2\eta}{n}\mathbf{K}\right)^t\right\|\left\|\tilde{h}_*^{\text{kls},j}\right\|_{\mathcal{H}} \leq \sqrt{R}.$$

$\square$

**Lemma 17.** *[21, Lemma 8] Let $f_t^\kappa, \tilde{f}_t^\kappa$ be GD-trained KLS predictors (as discussed in appendix B.2) given training samples $(\mathbf{x}_i, y_i)_{i=1}^n$ and $(\mathbf{x}_i, \tilde{y}_i)_{i=1}^n$ respectively, where $y_i = f^\star(\mathbf{x}_i) + \varepsilon_i$ and $\tilde{y}_i = h(\mathbf{x}_i) + \varepsilon_i$ with $\|h\|_{\mathcal{H}}^2 \le R$ characterized by lemma 14. Then, with $A(R)$ defined in lemma 14, for any $t \in \mathbb{N}$,*

$$\|f_t^\kappa - \tilde{f}_t^\kappa\|_n^2 \le A(R) .$$

## B.5 Proof of Theorem 3

*Proof.* We notice the following risk decomposition:

$$
\mathbb{E}\left\|h_T^j(\boldsymbol{\alpha}) - w_j^*(\boldsymbol{\alpha})\right\|^2 \le 4\,\mathbb{E}\left\|h_T^j(\boldsymbol{\alpha}) - h_T^{\mathrm{kls},j}(\boldsymbol{\alpha})\right\|^2 + 4\,\mathbb{E}\left\|h_T^{\mathrm{kls},j}(\boldsymbol{\alpha}) - \tilde{h}_T^{\mathrm{kls},j}(\boldsymbol{\alpha})\right\|^2
$$

$$
+ 4\,\mathbb{E}\left\|\tilde{h}_t^{\mathrm{kls},j}(\boldsymbol{\alpha}) - h_*^{\mathrm{kls},j}(\boldsymbol{\alpha})\right\|^2 + 4\,\mathbb{E}\left\|h_*^{\mathrm{kls},j}(\boldsymbol{\alpha}) - w_j^*(\boldsymbol{\alpha})\right\|^2
$$

$$
\le O\left(\max\left\{\frac{\mathrm{poly}_3(n)}{\sqrt{m}}, \frac{\mathrm{poly}_8(n)}{m}\right\}\right) + O\left(\left(1 + \frac{c}{\lambda_0^2}\right)A(R)^2 + \frac{c}{\sqrt{n}}\right)
$$

$$
+ \left(1 - \frac{2\lambda_0^2\eta}{n}\right)^T + O\left(\frac{R}{n}\right) + O\left(\Lambda\left(\frac{\sqrt{R}}{\Lambda}\right)^{-\frac{2}{N-2}}\ln\left(\frac{\sqrt{R}}{\Lambda}\right)\right),
$$

where respectively we plug in the NN-KLS coupling result (Theorem 6), Lemma 17, excess risk of KLS (Lemma 16) and approximation error of RKHS to Lipschitz function (Lemma 14). By choosing $m \ge \Omega(n^{8+\frac{2}{2+N}})$, $T \ge \Omega\left(\frac{n}{N\lambda_0^2\eta}\log(n)\right)$ and optimizing over $R$, we recover the rate:

$$
\mathbb{E}\left\|h_t^j(\boldsymbol{\alpha}) - w_j^*(\boldsymbol{\alpha})\right\|^2 \le O\left(\Lambda^2 n^{-\frac{2}{2+N}}\right).
$$

Hence by summing over all coordinates $j \in [d]$ we conclude the result:

$$
\mathbb{E}\|\mathbf{h}_t(\boldsymbol{\alpha}) - \mathbf{w}^*(\boldsymbol{\alpha})\|^2 \le O\left(\Lambda^2 d n^{-\frac{2}{2+N}}\right)
$$

where in our case lemma 1 implies $\Lambda = \kappa^*$. Finally we lower bound $\lambda_0$ using [29, Theorem 3.2] which implies that $\lambda_0 = \Omega(\mathrm{polylog}(n, N)N)$ with probability at least $1 - n^2 e^{-O(\sqrt{N})}$ for a distribution on a simplex. $\qquad\square$

## C  Regret Analysis of Algorithm 4

In this section, we are going to provide the proof for Theorem 4. Let us first introduce the following definitions that will be used in our proof.

**Definition 2** (Covering and Packing Numbers). *An $\varepsilon$-cover of a set $S$ w.r.t. some metric $\|\cdot\|$ is a set $\{\mathbf{x}'_1, \ldots, \mathbf{x}'_n\} \subseteq S$ such that for each $\mathbf{x} \in S$ there exists $i \in \{1, \ldots, n\}$ such that $\|\mathbf{x} - \mathbf{x}'_i\| \le \varepsilon$. The covering number $\mathcal{N}(S, \varepsilon, \|\cdot\|)$ is the smallest cardinality of a $\varepsilon$-cover. An $\varepsilon$-packing of a set $S$ w.r.t. some metric $\|\cdot\|$ is a set $\{\mathbf{x}'_1, \ldots, \mathbf{x}'_m\} \subseteq S$ such that for any distinct $i, j \in \{1, \ldots, m\}$, we have $\|\mathbf{x}'_i - \mathbf{x}'_j\| > \varepsilon$. The packing number $\mathcal{M}(S, \varepsilon, \|\cdot\|)$ is the largest cardinality of a $\varepsilon$-packing.*

It is well known that $\mathcal{M}(S, 2\varepsilon, \|\cdot\|) \le \mathcal{N}(S, \varepsilon, \|\cdot\|) \le \mathcal{M}(S, \varepsilon, \|\cdot\|)$.

**Definition 3** (Metric dimension). *The metric space $(\mathcal{X}, \|\cdot\|)$ had metric dimension $d$, if there exists a constant $C_d$ such that for all $\varepsilon > 0$, $\mathcal{X}$ has an $\varepsilon$-cover at most $C_d \varepsilon^{-d}$.*

*Proof.* We start from the standard analysis framework for online non-parametric regression [13, 18, 20]. Let $S_t$ be the value of the variable $S$ at the end of time $t$. Hence $S_0 = \varnothing$. The functions $\pi_t : \mathcal{X} \to \{1, \ldots, t\}$ for $t = 1, 2, \ldots$ map each data point $\mathbf{x}$ to its closest center (in norm $\|\cdot\|$) in $S_{t-1}$,

$$\pi_t(\boldsymbol{\alpha}) = \begin{cases} \arg\min_{s \in S_{t-1}} \|\boldsymbol{\alpha} - \boldsymbol{\alpha}_s\| & \text{if } S_{t-1} \ne \varnothing \\ t & \text{otherwise.} \end{cases}$$

The set $T_s$ contain all data points $\boldsymbol{\alpha}_t$ that at time $t$ belonged to the ball with center $\boldsymbol{\alpha}_s$ and radius $\varepsilon_t$,

$$T_s = \{t \,:\, \|\boldsymbol{\alpha}_t - \boldsymbol{\alpha}_s\| \le \varepsilon_t,\, t = s, \ldots, T\} \ .$$

Finally, $\mathbf{w}^\star_s$ is the best fixed prediction for all examples $(\boldsymbol{\alpha}_t, \mathbf{w}_t)$ such that $t \in T_s$,

$$\mathbf{w}^\star_s = \arg\min_{\mathbf{w} \in \mathcal{W}} \sum_{t \in T_s} \ell_t(\mathbf{w}) = \frac{1}{|T_s|} \sum_{t \in T_s} \mathbf{w}_t \ . \tag{30}$$

We proceed by decomposing the regret into estimation and approximation cumulative errors,

$$R_T(f) = \sum_{t=1}^T \Big( \ell_t(\hat{\mathbf{w}}_t) - \ell_t\big(f(\boldsymbol{\alpha}_t)\big) \Big) = \sum_{t=1}^T \Big( \ell_t(\hat{\mathbf{w}}_t) - \ell_t\big(\mathbf{w}^\star_{\pi_t(\boldsymbol{\alpha}_t)}\big) \Big) + \sum_{t=1}^T \Big( \ell_t\big(\mathbf{w}^\star_{\pi_t(\boldsymbol{\alpha}_t)}\big) - \ell_t\big(f(\boldsymbol{\alpha}_t)\big) \Big) \ .$$

The estimation term is bounded as

$$\sum_{t=1}^T \Big( \ell_t(\hat{\mathbf{w}}_t) - \ell_t\big(\mathbf{w}^\star_{\pi_t(\boldsymbol{\alpha}_t)}\big) \Big) = \sum_{s \in S_T} \sum_{t \in T_s} \Big( \ell_t(\hat{\mathbf{w}}_t) - \ell_t(\mathbf{w}^\star_s) \Big)$$

We study the cumulative regret within a fixed ball $T_s$. We use $T_s(t)$ to denote the set of all points belong to ball with center $\mathbf{w}_s$, at time $t$. That is,

$$T_s(t) = \{j : j \in T_s, j \le t\}.$$

One property of $T_s(t)$ is that, if $\mathbf{x}_t \in T_s$, then $|T_s(t)| = |T_s(t-1)| + 1$. We further define $\tilde{\mathbf{w}}_t = \frac{1}{|T_s(t)|} \sum_{j \in T_s(t)} \mathbf{w}^*(\boldsymbol{\alpha}_j)$. According to [6, Lemma 3.1], we have:

$$\sum_{t \in T_s} \ell_t(\hat{\mathbf{w}}_t) - \ell_t(\mathbf{w}^\star_s) \le \sum_{t \in T_s} \ell_t(\hat{\mathbf{w}}_t) - \ell_t(\tilde{\mathbf{w}}_t)$$

Recall that $\hat{\mathbf{w}}_t = \frac{1}{|T_s(t-1)|} \sum_{j \in T_s(t-1)} \mathbf{w}_j$. Then,

$$\mathbb{E}[\ell_t(\hat{\mathbf{w}}_t) - \ell_t(\tilde{\mathbf{w}}_t)] \le 4D\, \mathbb{E} \, \|\hat{\mathbf{w}}_t - \ell_t(\tilde{\mathbf{w}}_t)\|$$

$$= 4D\, \mathbb{E} \left\| \frac{1}{|T_s(t-1)|} \sum_{j \in T_s(t-1)} \mathbb{I}\{Z_j = 1\} \mathbf{w}_j - \frac{1}{|T_s(t)|} \sum_{j \in T_s(t)} \mathbf{w}^*(\boldsymbol{\alpha}_j) \right\|$$

$$\le 4D\, \mathbb{E} \left\| \frac{1}{|T_s(t-1)|} \sum_{j \in T_s(t-1)} \mathbb{I}\{Z_j = 1\} (\mathbf{w}_j - \mathbf{w}^*(\boldsymbol{\alpha}_j)) \right\|$$

$$+ 4D\, \mathbb{E} \left\| \frac{1}{|T_s(t-1)|} \sum_{j \in T_s(t-1)} \mathbb{I}\{Z_j = 1\} \mathbf{w}^*(\boldsymbol{\alpha}_j) - \frac{1}{|T_s(t)|} \sum_{j \in T_s(t)} \mathbf{w}^*(\boldsymbol{\alpha}_j) \right\|$$

According to the convergence rate of GD on strongly-convex and smooth function, the first term is bounded by $p(1 - \frac{1}{\kappa})^K D$. For second term:

$$\mathbb{E}\left\|\frac{1}{|T_s(t-1)|}\sum_{j\in T_s(t-1)}\mathbb{I}\{Z_j = 1\}\mathbf{w}^*(\boldsymbol{\alpha}_j) - \frac{1}{|T_s(t)|}\sum_{j\in T_s(t)}\mathbf{w}^*(\boldsymbol{\alpha}_j)\right\|$$

$$= \mathbb{E}\left\|\sum_{j\in T_s(t-1)}\left(\frac{\mathbb{I}\{Z_j = 1\}}{|T_s(t-1)|} - \frac{1}{|T_s(t)|}\right)\mathbf{w}^*(\boldsymbol{\alpha}_j)\right\|$$

$$\leq \sum_{j\in T_s(t-1)}\mathbb{E}_{Z_j}\left\|\left(\frac{\mathbb{I}\{Z_j = 1\}}{|T_s(t-1)|} - \frac{1}{|T_s(t)|}\right)\mathbf{w}^*(\boldsymbol{\alpha}_j)\right\|$$

$$\leq \sum_{j\in T_s(t-1)}\left(p\frac{1}{|T_s(t-1)||T_s(t)|} + (1-p)\frac{1}{|T_s(t)|}\right)D$$

$$\leq \left(p\frac{1}{|T_s(t)|} + (1-p)\right)D$$

where at the second last step we use the fact that $|T_s(t)| = |T_s(t-1)| + 1$.

$$\sum_{t\in T_s}\ell_t(\hat{\mathbf{w}}_t) - \ell_t(\mathbf{w}_s^\star) \leq \sum_{t\in T_s}\ell_t(\hat{\mathbf{w}}_t) - \ell_t(\tilde{\mathbf{w}}_t)$$

$$\leq 4D\sum_{t\in T_s}\left(p\frac{1}{|T_s(t)|} + (1-p)\right)D + 4pD|T_s|(1 - \frac{1}{\kappa})^K D$$

We further sum above statement over all $s \in S_T$:

$$\mathbb{E}\left[\sum_{s\in S_T}\sum_{t\in T_s}\ell_t(\hat{\mathbf{w}}_t) - \ell_t(\mathbf{w}_s^\star)\right] \leq p\sum_{s\in S_T}\left(\ln(|T_s|) + (1-p)TD + 4pD|T_s|(1 - \frac{1}{\kappa})^K D\right)$$

$$\leq p|S_T|\ln(T) + (1-p)TD + 4pDT(1 - \frac{1}{\kappa})^K D$$

The first inequality is a known bound on the regret under square loss [6, page 43]. We upper bound the size of the final packing $S_T$ as

$$|S_T| \leq \mathcal{M}\big(B, \varepsilon_T, \|\cdot\|\big) \leq C_N \varepsilon_T^{-N}$$

Thus,

$$\sum_{t=1}^{T}\left(\ell_t(\hat{\mathbf{w}}_t) - \ell_t\big(\mathbf{w}_{\pi_t(\boldsymbol{\alpha}_t)}^\star\big)\right) \leq pC_N\varepsilon_T^{-N}\ln(T) + (1-p)TD + 4pDT(1 - \frac{1}{\kappa})^K D \qquad (31)$$

Next, we bound the approximation term. Using (30) we have

$$\sum_{t=1}^{T}\left(\ell_t\big(\mathbf{w}_{\pi_t(\boldsymbol{\alpha}_t)}^\star\big) - \ell_t\big(f(\boldsymbol{\alpha}_t)\big)\right) \leq \sum_{t=1}^{T}\left(\ell_t\big(f(\boldsymbol{\alpha}_{\pi_t(\boldsymbol{\alpha}_t)})\big) - \ell_t\big(f(\boldsymbol{\alpha}_t)\big)\right). \qquad (32)$$

Note that $\ell_t$ is $4D$-Lipschitz because $\mathbf{y}_t, \hat{\mathbf{y}}_t \in \mathbb{B}_2(D)$. Hence,

$$\ell_t\big(f(\boldsymbol{\alpha}_{\pi_t(\boldsymbol{\alpha}_t)})\big) - \ell_t\big(f(\boldsymbol{\alpha}_t)\big) \leq 4D\big|f(\boldsymbol{\alpha}_{\pi_t(\boldsymbol{\alpha}_t)}) - f(\boldsymbol{\alpha}_t)\big| \leq 4D\kappa^*\|\boldsymbol{\alpha}_{\pi_t(\boldsymbol{\alpha}_t)} - \boldsymbol{\alpha}_t\| \leq 4D\varepsilon_t$$

by $\kappa^*$-Lipschitzness of $f$. Thus, putting all together,

$$\sum_{t=1}^{T}\ell_t(\hat{\mathbf{w}}_t) - \sum_{t=1}^{T}\ell_t(f(\boldsymbol{\alpha}_t)) \leq 8\ln(eT)C_N\varepsilon_T^{-N} + 4D\kappa^*\sum_{t=1}^{T}\varepsilon_t$$

and recalling that $\varepsilon_t = t^{-\frac{1}{1+N}}$ we have

$$\sum_{t=1}^{T} t^{-\frac{1}{1+N}} \leq \sum_{t=1}^{T} t^{-\frac{1}{1+N}} \leq \int_0^T \tau^{-\frac{1}{1+N}} \, \mathrm{d}\tau = \left(1 + \frac{1}{N}\right) T^{\frac{N}{1+N}} \leq 2T^{\frac{N}{1+N}}$$

which completes the proof.

$\square$

