# OpenReview forum: "Mixture Weight Estimation and Model Prediction in Multi-source Multi-target Domain Adaptation"
_NeurIPS.cc/2023/Conference — NeurIPS 2023 poster_

### Official Review · Reviewer_Ety2 · 2023-07-05

**Soundness:** 3 good
**Presentation:** 2 fair
**Contribution:** 2 fair
**Rating:** 4
**Confidence:** 3

**Summary:**

This paper motivates the mixture weight estimation problem using Multi-source Multi-target Domain Adaptation problem. More specifically, this paper considers how to estimate the optimal mixture of sources, given a target domain; also, when there are multiple target domains, how to solve empirical risk minimization (ERM) for each target using a possibly unique mixture of data sources in a computationally efficient manner. This paper tackles both problems by constructing new efficient algorithms with a convergence guarantee.

**Strengths:**

This paper provides a rigorous theoretical analysis of the optimization algorithm proposed in the paper and considers both offline and online settings for the problem.

**Weaknesses:**

This paper focuses more on the theoretical analysis of a specific optimization problem instead of addressing the Multi-source Multi-target Domain Adaptation (M2DA) problem.

1. The bound in Theorem 1 can be quite loose; therefore, minimizing the right-hand side does not necessarily result in good weights in the target domain. As the right-hand side of this bound is minimized using empirical data, will there be an extra overfitting issue? Is there a way to bound this gap?

2. If some training data from the target domain is available, i.e., $\hat{\mathcal{T}}$, why not use these target data in the training of $h$ so that you have N+1 dimensional domain weights?

3. The relaxation from (1) to (2) seems arbitrary, especially the removal of the concave function of the square root.

4. There are no experimental results to verify the effectiveness of the proposed algorithm, not even on synthetic data.

**Questions:**

I appreciate the theoretical contribution made in the paper by analyzing the convergence of the proposed stochastic corrected gradient descent ascent algorithm. However, the current way of presenting these results is nothing but motivating the specific convex-nonconcave minimax problem using a contrived application of M2DA, which seems to be quite off. I would suggest the authors revise the paper by focusing on the theoretical contribution made to the convex-nonconcave minimax problem considered in eq (2) and the co-component ERM problems in eq. (3), which are interesting in their own right. However, this may require a significant rewrite of the paper and even a change in the title.

**Limitations:**

There are no experimental results to verify the effectiveness of the proposed algorithm, not even on synthetic data.

---

> ### Author Rebuttal · Authors · 2023-08-10
>
> **This paper is more like optimization not M2DA paper**
> We are afraid that we have to respectfully disagree with you on this point. As we mentioned in global rebuttal, the mixing domain type of multi-source learning algorithm typically contains two parts: finding good mixing weights and solving ERM. Our primary goal is to find good mixing weights, and minimax algorithm is just the technique to achieve this. As for the second part of our paper, we consider a novel but very important setting, where there exist multiple target domains to adapt. This setting motivates a new theory topic, co-component ERM, and we provide two possible ways to solve this.
>
> **Theorem 1 can be quite loose, and Eq. 1 uses empirical risk which will result in overfitting**
> First, we would argue that Theorem 1 is not loose. It recovers the optimal generalization risk which is subroot in terms of number of samples.  We agree with you that
> Eq. 1 is an empirical estimation of RHS of Theorem 1, where we use the empirical risk to replace the population risk in Theorem 1. The difference between empirical risk and population risk can be bounded by Rademacher complexity. We drop the Rademacher complexity terms for two reasons: first, it scales as $\sqrt{\frac{1}{m_i}}$, subroot function of number of samples, which is already captured by the second term in Eq. 1. Then, computing Rademacher complexity is usually expensive. Since it is not a dominating term, we drop it for computationally convenience, following [KL19]. Adding the complexity term to our objective, it will still be a convex-nonconcave problem, and our algorithm 1 still works. At last, optimizing with empirical discrepancy is a very standard technique in domain adaptation fields, for example, see [MMA09], [BBCLPV10], [ZLLJ19].
>
> **If some training data from the target domain is available, why not use target data as source as well?**
> We agree with you that we can use target data as one additional source. Actually, our theory applies to this setting, by just changing the $N$ source to $N+1$ source, and the rest of the analysis still works.
>
> **The relaxation from (1) to (2) seems arbitrary, especially the removal of the concave function of the square root.**
> Indeed the square root term in (1) is convex in alpha, and we relaxed a convex object to strongly convex objective. This operation is quite standard in optimization, e.g., relaxing a l2 norm to squared l2 norm.
>
> **No experiment**
> As we mentioned in global answer, we provide experiments with two layer MLP on MNIST dataset in rebuttal pdf.
>
> [KL19] N. Konstantinov and C. H. Lampert. Robust learning from untrusted sources. In Interna-
> tional Conference on Machine Learing (ICML), pages 3488–3498. PMLR, 2019
>
> [MMA09] Mansour, Yishay, Mehryar Mohri, and Afshin Rostamizadeh. "Domain adaptation: Learning bounds and algorithms." arXiv preprint arXiv:0902.3430 (2009).
>
> [BBCLPV10] Ben-David, Shai, John Blitzer, Koby Crammer, Alex Kulesza, Fernando Pereira, and Jennifer Wortman Vaughan. "A theory of learning from different domains." Machine learning 79 (2010): 151-175.
>
> [ZLLJ19] Zhang, Yuchen, Tianle Liu, Mingsheng Long, and Michael Jordan. "Bridging theory and algorithm for domain adaptation." In International conference on machine learning, pp. 7404-7413. PMLR, 2019.

---

> ### Author Response · Authors · 2023-08-20
>
> Dear Reviewer Ety2,
>
> We want to thank you for your constructive suggestions and thoughtful reviews, which are valuable to improving our paper.
>
> We understand that we are not supposed to bother reviewers, but as the deadline is approaching, as a follow-up on our rebuttal, we would like to kindly remind you that the close date of the discussion is approaching. We hope to use this open response window to discuss the paper, answer follow-up questions, and improve the quality of our paper. Have you gotten a chance to read our rebuttal, in which we tried our best to address your concerns? We want to make sure that you found our responses solid and convincing. Notice that we already provided some additional experiments results in our response to Reviewer 9T2K, and we would be more than happy to provide more information or clarification.
>
> The authors

---

> > ### Comment · Reviewer_Ety2 · 2023-08-21
> > **Thanks for the response**
> >
> > The rebuttal addresses my concerns about weaknesses 2 and 3. I would increase my score by 1.
> >
> > If the goal of this paper is to solve M2DA, then the significance of the theoretical contributions will be discounted due to relaxations and simplifications, i.e., the analysis only focuses on the optimization of an upper bound. And the empirical results should demonstrate the effectiveness of the proposed method. The added experiments on MINIST are inspiring but not sufficient enough. More complicated datasets for multisource domain adaptation and something beyond two-layer NN should be added. I agree with reviewer 3rHn that
> > "A more focused way of presenting the paper would be to put the theory part with sufficient experiments in the main content while putting the rest of the theoretical results as an addition."
> >
> > In summary, I think that the changes required for publication are too significant for me to recommend acceptance in this review process.

---

> > > ### Author Response · Authors · 2023-08-21
> > > **Additional experiments on Office dataset**
> > >
> > > Thanks for your feedback! We also appreciate your constructive suggestions, and we would like to address your remaining concerns as follows:
> > >
> > > 1. Optimizing to minimize the generalization bound is a natural and common technique to obtain a better model in domain adaptation. See the seminal works [1][2][3][4]. In all of the above works, they derived the generalization bound of domain adaptation, and minimize the right hand side of generalization bound to yield a good model.
> > >
> > > 2. We conduct more experiments on Office dataset, which is a widely-used domain adaptation dataset. The dataset contains three subdataset: amazon, webcam and dslr collected from different scenarios.
> > > Domain generation:
> > >
> > > |   Group  | Domain per group | samples per domain |
> > > |-----------|-----------|-----------|
> > > | Amazon  |  5 |500 |
> > > | Webcam  |5 | 100 |
> > > | dslr  |5 |100 |
> > >
> > > Results:
> > >
> > > |        | Target (Amazon) |Target (Webcam) |
> > > |-----------|-----------|-----------|
> > > | Average ERM | 86.25% |  83.75 % |
> > > | Pure target training | 58.75%   |42.5 %|
> > > | Our method | **91.25%** |**90 %** |
> > >
> > > Due to the time limit, we only post partial results. We will add more target setting, as well as ResNet results in revised version. Thanks for your suggestion again and hopefully it can further mitigate your concerns.
> > >
> > >
> > > References:
> > >
> > > [1] Ben-David, Shai, John Blitzer, Koby Crammer, Alex Kulesza, Fernando Pereira, and Jennifer Wortman Vaughan. "A theory of learning from different domains." Machine learning 79 (2010): 151-175.
> > >
> > > [2] Mansour, Yishay, Mehryar Mohri, and Afshin Rostamizadeh. "Domain adaptation: Learning bounds and algorithms." arXiv preprint arXiv:0902.3430 (2009).
> > >
> > > [3] Zhang, Yuchen, Tianle Liu, Mingsheng Long, and Michael Jordan. "Bridging theory and algorithm for domain adaptation." In International conference on machine learning, pp. 7404-7413. PMLR, 2019.
> > >
> > > [4] N. Konstantinov and C. H. Lampert. Robust learning from untrusted sources. In Interna- tional Conference on Machine Learing (ICML), pages 3488–3498. PMLR, 2019

---

### Official Review · Reviewer_T5Nt · 2023-07-08

**Soundness:** 3 good
**Presentation:** 4 excellent
**Contribution:** 3 good
**Rating:** 7
**Confidence:** 3

**Summary:**

Authors formulate the problem of optimizing mixture weights given the target domain as a compositional convex-concave minimax optimization problem. Then, they propose a stochastic descent ascent algorithm for solving the problem, which improves upon previous method of [31] by allowing stochastic updates. Then, authors address the second problem of: given a large number of target domain distributions, how to efficiently find model weights for all of target distributions? This is discussed in both offline and online setting. In the offline setting, a two-layer ReLU neural network is trained to output model weights; the model is essentially a "hypernetwork" that outputs model weights. In the online setting, nonparametric online regression method is adapted for the problem.

**Strengths:**

Authors identify an important instance of a convex-nonconcave minimax problem. The learning of mixture weights has become both theoretically and empirically important topic of research. For example, methods like Group DRO (Sagawa et al https://arxiv.org/abs/1911.08731 ) and DoReMi (Xie et al, https://arxiv.org/abs/2305.10429 ) have been drawing attention. The reduction of this learning problem to an abstract convex-nonconcave optimization problem will facilitate the adoption of techniques from a broader optimization literature.

Authors also push the state of the art on convex-nonconcave minimax problem by developing a stochastic version of the algorithm, adopting recently developed techniques such as stochastic corrected gradients.

**Weaknesses:**

The first half of the paper (mixture weight optimization) and the second half (model weight prediction) are related, but the connection is not very strong. These two ideas could've made good two separate papers. While making these results a single paper made this paper very rich with technical content, but on the other hand, readers of the main body of the paper shall learn much less than what they would with two separate papers. The rationale of authors seems to be that mixture weights for the second problem could be found from the algorithm from first half, but in this case, we already know model weights; thus, the second problem is not very well motivated from the first problem.

There is no numerical experiments in this paper, and hence results in the paper are not numerically validated. Also, this makes the practical utility of proposed algorithms less clear.

**Questions:**

The offline version of the algorithm (Section 3.1) is quite different from the online version of the algorithm (Section 3.2). Would it be meaningful to compare them against each other? For ex, the offline version of the algorithm could be applied to online setting by occasionally computing the label, and the online version of the algorithm could be applied to offline setting as a baseline?

In line 139, two relaxations in 139-144 are explained as standard. But within which literature are these standard? Can authors provide references for such relaxations?

Can stochastic versions of the Algorithm 2 exist (in terms of both $i$ and data points), which can alleviate the dependency on $M$?

**Limitations:**

What are alternative formulations for mixture weight estimation (1), and how would convex-nonconcave formulation compare against them? Practically, wouldn't it be too pessimistic to consider the supremum over $\mathcal{H}$?

---

> ### Author Rebuttal · Authors · 2023-08-10
>
> Thank you for your valuable comments! We will try to address your concerns as follows.
>
> **The connection between two parts of the paper**
> We agree with you that the two parts of the paper already have their own independent interests. The reason we put them into one paper is that the two parts put together solved the multi-source multi-target domain adaptation problems, which is a complete story.
>
> **No experiments**
> As we mentioned in global answer, we provide experiments with two layer MLP on MNIST dataset in rebuttal pdf.
>
> **The two relaxations**
> Our first relaxation is relaxing absolute value by $\sqrt{x^2+c}$, which is used in non-smooth optimization paper, for example [CLY23]. Our second relaxation is similar to relaxing the l2 norm to squred l2 norm, which is widely used in practice, when we wish to regularize the norm of the model.
>
> **Can stochastic versions of the Algorithm 2 exist (in terms of both
>  and data points), which can alleviate the dependency on M
> ?**
> Thanks for raising this interesting point. We believe this is feasible, by enabling the sampling idea in the dynamic of Algorithm 2. We will seriously consider it as a promising follow-up work.
>
> **Alternative formulations for mixture weight estimation.**
> To our best knowledge, optimizing Eq.1 is the only mixture weight estimation method with theoretical support. However, we believe there must be other ways to optimize for mixture weights, which we leave as promising open problem.
>
> **Would it be too pessimistic to consider the supremum over whole hyothesis class?**
> The supremum over hypothesis class is used to achieve uniform convergence generalization bound, but we also notice that there are some localization technique, e.g., local Rademacher complexity that could enable us to do finer generalization analysis by studying the subset of hypotheses with small risk. In domain adaptation field, a localized discrepancy measure is proposed by zhang2020localized. We believe applying this idea to multi-source learning setting will be a very interesting open problem.
>
> [CLY23] Chen, Xiaojun, Lingfeng Niu, and Yaxiang Yuan. "Optimality conditions and a smoothing trust region newton method for nonlipschitz optimization." SIAM Journal on Optimization 23.3 (2013): 1528-1552.

---

> > ### Comment · Reviewer_T5Nt · 2023-08-13
> > **Thanks for answers**
> >
> > Thank you very much for thoughtful answers to my questions. These make sense, and I agree they would be better addressed in another subsequent paper. Good luck pursuing these directions.

---

> > > ### Author Response · Authors · 2023-08-13
> > > **Thanks for your feedback**
> > >
> > > Thank you so much for your comments. We are truly grateful for your encouraging feedback and insightful suggestions!

---

### Official Review · Reviewer_3rHn · 2023-07-11

**Soundness:** 2 fair
**Presentation:** 2 fair
**Contribution:** 2 fair
**Rating:** 5
**Confidence:** 1

**Summary:**

This paper is about the multi-source multi-target domain adaptation problem. The authors formulate a minimax algorithm to find the mixture weights of source domains. Furthermore, the authors extend it to the scenario of multi-target domains and introduce the co-component ERM problem. For this problem, this paper proposes algorithms to efficiently solve co-component ERM problems, in offline and online fashions.

**Strengths:**

This paper extensively and theoretically studied the domain adaptation problem from the minimax optimization perspective.

The paper gives the convergence analysis of the proposed algorithms.

For the multi-target domain scenario, the authors provide solutions to both the offline and online settings. The solution is more efficient than training each target domain adaptation independently.

**Weaknesses:**

This paper is fully theoretical. The research focus of this paper, i.e., domain adaptation, has many open benchmark datasets and thus it would be better to experimentally evaluate the proposed methods.

In line 148, it is mentioned that a deterministic algorithm exists in the literature, while this paper is focused on the stochastic one. It would be good to elaborate more on the benefits of the stochastic one and meanwhile experimental compare them.

**Questions:**

As for the minimax optimization, in Eq.(1) and Eq.(2), why are the model/hypothesis parameters $\omega$ to maximize the difference between the target and source domain? Should it be the minimization?

**Limitations:**

No potential negative societal impact is found in this work.

---

> ### Author Rebuttal · Authors · 2023-08-10
>
> Thank you for your valuable comments! We will try to address your concerns as follows.
>
> **No experiments**
>
> As we mentioned in global answer, we provide experiments with two layer MLP on MNIST dataset in rebuttal pdf.
>
> **Comparison with deterministic convex-nonconcave optimization**
> In Xu et al 2023, they achieves convergence rate of $O(\kappa_F^2/\epsilon^2)$ which is faster than ours $O(\kappa_F^4/\epsilon^4)$.
> However, the problem of the deterministic algorithm is that, in practice it is too expensive to compute the full gradient, and hard to implement since we do not have enough memory to directly compute full-batch gradient.
>
> **Why use maximization over hypothesis space in Eq.1 and Eq.2**
> We use the maximization over hypothesis space is because our bound in Theorem 1 is a uniform convergence bound, which holds for every hypothesis in the class $\mathcal{H}$.

---

> > ### Comment · Reviewer_3rHn · 2023-08-21
> > **Thanks for answers. Score remained.**
> >
> > Thanks for the authors' answers and the added experimental evaluation, which all help better understand the paper!
> >
> > I prefer to keep the score for the following reasons.
> >
> > 1. I agree with other reviewers that Sec. 2 and Sec.3 look a bit disconnected, and thus the paper is a bit overwhelmed by the theoretical results, while the motivation of the corresponding problem formulations looks weak.
> >
> > A more focused way of presenting the paper would be to put the theory part with sufficient experiments in the main content while putting the rest of the theoretical results as an addition.
> >
> > 2. Though the authors provide the experimental evaluation during the rebuttal, considering several different problem setups proposed in this paper, it is still unclear whether a sufficient experimental evaluation is doable for all these problem setups, and also whether the experiments for these setups are realistic.
> >
> > Meanwhile, since this paper is not the focus of my research areas, I suggest the AC considering the low confidence score of my review.

---

### Official Review · Reviewer_9T2K · 2023-07-20

**Soundness:** 3 good
**Presentation:** 2 fair
**Contribution:** 2 fair
**Rating:** 6
**Confidence:** 3

**Summary:**

Authors propose a new way to compute mixture coefficients for combining multiple empirical risk minimization objectives (w.r.t. different sources in domain adaptation) in a way that takes into account the relation to a new target domain. As an application, the authors consider the multi-source multi-target domain adaptation scenario and solve it by predicting (the weights of) new target classifiers from the mixture weights.

**Strengths:**

- The considered phase transition is interesting.
- The weights could be provided by a domain expert who is, e.g., certain about a physical relation between the domains.
- Convergence of the algorithm as extension of [31] is interesting.


**Weaknesses:**

- No empirical intuition if the algorithm can be implemented with reasonable effort. I consider the result as of purely theoretical interest.
- The error in the computation of the weights $\alpha_i$ is not taken into account in the error rates results. This could dominate the convergence rates results.
- My impression is that, if we are able to solve Eq. (1) efficiently, then also a separate training for each target domain should give us a comperable accuracy. I don't see any argument, neither theoretical or practical, which guarantees that the proposed lagorithm improves the separate learning (which is possible with labels in the target domain).

**Questions:**

- For computing $\alpha$ in Eq. (1), one needs to compute differences in empirical risks (on target vs. source datasets). Errors in this difference seem to aggregate to errors in the solution for $\alpha$. How does this effect the final prediction of the target model from mixture weights?
- In meta-learning there is the approach of representing novel domains by "meta-features" and then map these meta-features to the hyper-parameters of novel target domains. The hyper-parameters can also be model weights, as in your case given by $w^\ast(\alpha)$ cited in the introducion. Consequently, the theory of meta-learning should apply also to your setting. How does this theory, e.g. [1], compare to your convergence results? Can the same phase transition be observed assuming the mixture weights are exact?

[1] https://www.jmlr.org/papers/volume6/maurer05a/maurer05a.pdf

------------------------------
After rebuttal: My questions are addressed. I increase my score by 2.

**Limitations:**

- Influence of error of weight estimation should be discussed.

---

> ### Author Rebuttal · Authors · 2023-08-10
>
> Thank you for your valuable comments! We will try to address your concerns as follows.
>
> **No empirical intuition if the algorithm can be implemented with reasonable effort**
> We implement our Algorithm 1 and provide results on MNIST dataset. It turns out the mixing parameter output by our algorithm yields a good model which can outperform naive ERM model or purely target domain learnt model.
>
> **The error in the computation of the weights**
> We assume you mean the error between the output alpha of Algorithm 1 and optimal solution to objective Eq.2. We cannot characterize this error since it is an convex-nonconcave problem, showing the convergence to global optimal point is NP-hard.
>
> **The separate learning on target domain**
> In practice, the target domain usually has very few labeled data, and hence learning solely on target domain will not yield a well-generalized model. Our experiment in PDF, Table.2 also validates this argument.
>
> **How does errors in the difference of empirical risks affect alpha**
>
>
> Maximizing the inner level of the difference of empirical risk is a non-concave problem, which cannot be exactly solved. Hence, we say that the whole problem of (1) is convex-nonconcave. Solving a convex-nonconcave minimax problem is NP-hard, so we can only provide a convergence to the stationary point. Since we cannot give a guarantee of the output alpha versus the global optimal alpha of (1), we cannot characterize the how good our output model is, compared to the model learnt under optimal alpha.
>
> **Comparison to meta-learning paper of [1]**
>
> Thank you for pointing out this interesting relationship.
> [1] analyzes source aggregation scheme (CP-Regression) where feature vector for a new task is formed from predictions of models trained on source datasets.
> Indeed, one could imagine obtaining predictor for N+1'th task using this approach.
>
> That said,  [1] only shows that the generalization gap, that is the difference between risk and empirical risk, can be controlled using such a scheme.
> Therefore, their result does not say anything about how close such a solution is to the best possible on the target problem.
> In fact, achieving the best possible performance on the target problem might require a very different algorithm: For instance,  [KKS21]  show lower bounds for the meta-learning in linear case, which suggest that aggregation as in CP-Regression is a suboptimal because it does not take into account task covariance.
>
> In our work we design an algorithm (solving Eq. (1) + solving eq.\ in panel at the end of page 2) which in the best case achieves the best possible performance on target problem.
> This is because our proposed algorithm is designed from the start to directly minimize the bound the gap between the risk and the best possible risk on the target task.
>
> [KKS21] M. Konobeev, I. Kuzborskij, and Cs. Szepesv ́ari. A Distribution-dependent Analysis of Meta
> Learning. In International Conference on Machine Learing (ICML), 2021

---

> > ### Comment · Reviewer_9T2K · 2023-08-15
> > **Thank you for your answer**
> >
> > I appreciate the answer of the authors. At the same time, I still have some concerns:
> > - The empirical evidence provided by the new experiment is vague. I think, a single split in several domains is too less to underpin the advantage of the algorithm compared to learning a single model on target.
> > - A pssible error made by sub-optimal weights can be incorporated assuming it is $\epsilon>0$. Does $\epsilon$ appear in the final error bound?
> > - I see there are many distinct insigths (empirical experiment, parts of the method are mathematically analysed). At the same time, I cannot see a final argument under which circumstances the proposed approach is better than learning a single model on the target domain.

---

> > > ### Author Response · Authors · 2023-08-17
> > >
> > > Many thanks for your comments. We will try to address your concerns as follows.
> > >
> > > **More Empirical Comparison**
> > >
> > > We totally agree that the initial empirical results, while demonstrating the effectiveness of estimating mixture parameters, does not show the advantage of the algorithm compared to learning a single model on target. This was mostly due to large number of samples in target domain which does not benefit much from source domains. Per your question,  we conducted more experiments on MNIST. This time, we create 3 groups of domains, and each group has 5 domains. Each group's domains only draw data from a subset of 10 classes, and each domain has 100 training data. These 15 domains are treated as source domains. We consider four different target domains: i) a domain from group 1, ii) a domain from group 2, iii) a domain from group 3,  and iv) a mixed domain whose data are sampled from both group 1 and group 2.
> > >
> > > Data Generation:
> > > |    Group    | Classes | Domain per group | samples per domain |
> > > |-----------|-----------|-----------|-----------|
> > > | 1 | 0,1,2 |  5 |100 |
> > > | 2 | 3,4,5 |5 |100 |
> > > | 3 | 6,7,8,9 |5 | 100 |
> > >
> > > We run experiments on the four different target domain settings, and list the results below, and in all of them, our method outperforms purely target training and average domain training:
> > >
> > > |        | Target (Group 1) |Target (Group 2) | Target (Group 3) | Target (mixture of Group 1 and 2) |
> > > |-----------|-----------|-----------|-----------|-----------|
> > > | Average ERM | 69.9 % |  40.0 % | 34.9 %| 59.9 %|
> > > | Pure target training | 69.9 %  |55.0 %|40.0 %|55.0 %|
> > > | Our method | **80.0 %** |**69.9 %** | **55.0 %** | **65.0 %** |
> > >
> > > As it can be observed from above results, learning from source domains using learned mixture weights can improve accuracy on target domain significantly. Interestingly, in some cases (Group 3) learning with average ERM is worse than just training on target domain (due to heterogeneity among data sources, the naive averaging is not effective which necessities  to weight each source based on it is relatedness to target domain which is the main motivation of our work).
> > >
> > > **A possible error made by sub-optimal weights can be incorporated assuming it is epsilon. Does  appear in the final error bound?**
> > >
> > > We agree with you that  Algorithm 1 can only output a sub-optimal weights, since the inner level of our objective is a nonconcave maximization problem. To characterize the distance of output and global optimal is intractable, since solving our convex-nonconcave problem is NP-hard, we can only show convergence to stationary point. If we pre-assume the error is epsilon,  then it can be easily incorporated into our error bound result (we assume you mean Theorem 4) by simple error decomposition:
> > > \begin{align*}
> > >     \mathbb{E}||h(\hat{\alpha}) - w^*(\alpha^*) ||^2 \leq 2\mathbb{E}||h(\hat{\alpha}) - w^*(\hat{\alpha}) ||^2+ 2\mathbb{E}||w^*(\hat{\alpha}) - w^*(\alpha^*) ||^2
> > >     \leq O(\kappa^2 d n^{-\frac{2}{2+N}})  + 2\kappa^2 \epsilon^2
> > > \end{align*}
> > > where we need to pay an extra error price in order of $\epsilon$. Notice that this does not change our conclusion about Phase Transition between the efficiency of learning to solve ERM and solving every ERM individually, since solving every ERM individually on the output weights $\hat{\alpha}$, this error will still exist if we try to compare $w^*(\hat{\alpha})$ and $w^*( {\alpha}^*)$.
> > >
> > > Thank you so much for raising this question. We will clarify this in the revised version.
> > >
> > >
> > > **When learning on mixed domain is better than purely target training**
> > > As it is evident from the experimental results above, when the target domain has very few number of data, and there are other source domains that are similar to target domain, learning on the mixed domain will yield better solution than purely target training. When target domain already has much training data, and other source domains have significant divergence with target, then purely target training is the better choice. To see this, consider we have $N$ source domains, to be $D_1,...,D_{N/2} = D$ and $D_{N/2+1},...,D_{N} = -D$, where $-D$ means the distribution has the same margin distribution on $\mathcal{X}$ as $D$, but each data is labeled by opposite labeling function $-f(x)$. Also consider target domain as $D$, the same as first half $N$ sources, but only with very few data. In this case, simply learning on target data, or learning on the average of source domains will yield a very bad model, and the optimal option is to learn on the mix of first half $N/2$ source domains. These relevant sources can be automatically discovered by learning the mixture weights using our proposal. As indicated in our experiments, this enables us to learn a model based on relevant sources where the contribution of each source is proportional to its closeness (statistical discrepency) to target domain.

---

> > > > ### Comment · Reviewer_9T2K · 2023-08-18
> > > > **I increased my score**
> > > >
> > > > Thank you for your clarification.
> > > > I'm convinced now that there are both, theoretical arguments and empirical evidence, that the proposed approach can outperform classical algorithms (e.g., supervised learning on target) in certain situations. In these situations, the proposed approach provides a concrete step forward; I increased my score by 2.
> > > > My suggestion is to incorporate (or re-structure) the discussion in the paper, regarding situations in which the proposed approach provably outperforms straight forward supervised learning solutions.

---

> > > > > ### Author Response · Authors · 2023-08-18
> > > > > **Thank you for your feedback**
> > > > >
> > > > > Thank you so much for your positive comments! We will definitely revise the paper according to your suggestions. Thank you for your constructive comments again!

---

### Official Review · Reviewer_u3ZP · 2023-07-29

**Soundness:** 3 good
**Presentation:** 3 good
**Contribution:** 2 fair
**Rating:** 4
**Confidence:** 3

**Summary:**


Summary:
The paper addresses the problem of multi-source multi-target domain adaptation, where the goal is to learn a model from multiple sources in such a way that it performs well on a new target distribution. The context for this problem includes scenarios like learning from data collected from various sources (e.g., crowdsourcing) or in distributed systems with highly heterogeneous data. The two main unsolved problems in this context are: 1) how to estimate the optimal mixture of sources for a given target domain, and 2) how to efficiently solve empirical risk minimization for each target domain when there are numerous target domains, which can be computationally expensive. The paper proposes solutions to both of these problems using convex-nonconcave compositional minimax and overparameterized neural networks with provable guarantees. Additionally, an online algorithm for predicting parameters for new models given mixing coefficients is proposed.

**Strengths:**

Theoretical Contributions: The paper proposes novel approaches to tackle the problems of mixture weight estimation and empirical risk minimization, utilizing convex-nonconcave compositional minimax and overparameterized neural networks, respectively. These contributions are supported with provable guarantees, which add rigor to the proposed methods.

Efficiency and Scalability: The paper emphasizes the efficiency of the proposed algorithms, particularly for mixture weight estimation and empirical risk minimization for multiple target domains. The avoidance of individual ERM for each target domain in certain cases helps reduce computational overhead.



**Weaknesses:**



Complexity: The proposed methods, such as convex-nonconcave compositional minimax and overparameterized neural networks, might be complex and difficult to implement for practitioners who are not familiar with these advanced techniques.

Applicability to All Domains: The paper may not clearly address the limitations or specific domains where the proposed techniques might not be directly applicable or might require additional adjustments.

Empirical Evaluation: The paper lacks details about empirical evaluations, such as experiments on real datasets or comparisons with other state-of-the-art methods. This could raise concerns about the practical effectiveness of the proposed algorithms.

**Questions:**

How does the proposed convex-nonconcave compositional minimax approach differ from existing methods used for mixture weight estimation?

Can you provide more details on the theoretical guarantees of the proposed overparameterized neural network approach for empirical risk minimization and its relationship to the specific problem context?

Are there any assumptions made about the data distribution or source/target domains that could limit the generalizability of the proposed methods?

How does the proposed method perform in practice?

**Limitations:**

Limited Empirical Validation: The lack of empirical evaluation or real-world case studies might raise questions about the practical effectiveness and applicability of the proposed methods to real-world scenarios.

---

> ### Author Rebuttal · Authors · 2023-08-10
>
> Thank you for your valuable comments! We will try to address your concerns as follows.
>
> **Complexity to implement algorithm**
> Our mixture weight estimation algorithm is a single loop primal-dual algorithm, which is widely used in minimax optimization and easy to implement. The only additional effort is to implement correction step for compositional term, which is also easy-implementable by maintaining an auxilliary variable. In our provided PDF, we implement the algorithm and show the convergence to a desired weights (Figure.1 in PDF). As for the algorithm 2, it is indeed a very common neural network model training, and can be implemented by few lines of Pytorch code.
>
> **Applicability to All Domains**
> Our algorithm can automatically find the good mixture weights, given any source domains. Hence, our algorithm can work well in any multi-source learning scenario. Our provided experiments also validate the effectiveness of our algorithm.
>
> **Empirical Evaluation** As we mentioned in global rebuttal, we provide the experiments in the rebuttal pdf.
>
>
> **Comparison with existing methods used for mixture weight estimation**
> [KL19]  also propose to optimize similar objective as ours to get mixture weight, but they do not give a practical algorithm, nor the rigorous convergence guarantee.  [MMR+21]  propose to sample many mixture weights on simplex, then do ERM to get multiple candidate target models, and pick the best one who has the smallest target empirical risk as final solution. The drawback of their method is, when the dimension of the simplex is very high, you may need to sample exponentially many weights to cover the whole simplex to some accuracy.
>
>
>
> **Can you provide more details on the theoretical guarantees of the proposed overparameterized neural network approach for empirical risk minimization and its relationship to the specific problem context?**
>
> Guarantee for minimizing empirical risk of an overparameterized neural network (Theorem 3) is based on a standard Neural Tangent Kernel (NTK) approximation argument [JGH18, BMR21], namely we use the key fact that predictions made by a GD-trained overparameterized neural network are close to those made by a Kernel Least-Squares (KLS) predictor (given that the width of the network is polynomial in $n/d$).
> The core idea has been recently explored in many papers, for instance,  [DLL+19, OS20, BMR21].
>
> In addition to that, in our work we need to establish that such neural networks are able to learn Lipschitz vector-valued target functions with iteratively refined labels -- which in the context of our paper means that given a mixture weight, a GD-trained neural network can output parameters for a target task.
>
> Here we extend their proof to vector-valued functions and inexact label observation setting, by analyzing the dynamic of a bi-level optimization.
>
> **Are there any assumptions made about the data distribution or source/target domains that could limit the generalizability of the proposed methods?**
> We do not make any assumption on data distribution. Our proposed mixture weight estimation algorithm can automatically adapt to different heterogeneity level, since it takes the distributions discrepancy into account.
>
> [BMR21] P. L. Bartlett, A. Montanari, and A. Rakhlin. Deep learning: a statistical viewpoint. Acta
> Numerica, 2021.
>
> [DLL+19] Simon Du, Jason Lee, Haochuan Li, Liwei Wang, and Xiyu Zhai. Gradient descent finds
> global minima of deep neural networks. In International Conference on Machine Learning,
> pages 1675–1685. PMLR, 2019.
>
> [JGH18] A. Jacot, F. Gabriel, and C. Hongler. Neural tangent kernel: convergence and generalization
> in neural networks. In Conference on Neural Information Processing Systems, 2018.
> [KKS21] M. Konobeev, I. Kuzborskij, and Cs. Szepesv ́ari. A Distribution-dependent Analysis of Meta
> Learning. In International Conference on Machine Learing (ICML), 2021.
>
> [KL19] N. Konstantinov and C. H. Lampert. Robust learning from untrusted sources. In Interna-
> tional Conference on Machine Learing (ICML), pages 3488–3498. PMLR, 2019.
>
> [KS22] I. Kuzborskij and Cs. Szepesv ́ari. Learning lipschitz functions by gd-trained shallow overpa-
> rameterized relu neural networks. arXiv:2212.13848, 2022.
>
> [MJ05] A. Maurer and T. Jaakkola. Algorithmic stability and meta-learning. Journal of Machine
> Learning Research, 6(6), 2005.
>
> [MMR+21] Y. Mansour, M. Mohri, J. Ro, A. T. Suresh, and K. Wu. A theory of multiple-source
> adaptation with limited target labeled data. In Arindam Banerjee and Kenji Fukumizu,
> editors, International Conference on Artificial Intelligence and Statistics (AISTATS), volume
> 130 of Proceedings of Machine Learning Research, pages 2332–2340. PMLR, 13–15 Apr 2021.
>
>  [OS20] S. Oymak and M. Soltanolkotabi. Toward moderate overparameterization: Global conver-
> gence guarantees for training shallow neural networks. IEEE Journal on Selected Areas in
> Information Theory, 1(1):84–105, 2020.

---

> ### Author Response · Authors · 2023-08-20
>
> Dear Reviewer u3ZP,
>
> We want to thank you for your constructive suggestions and thoughtful reviews, which are valuable to improving our paper.
>
> We understand that we are not supposed to bother reviewers, but as the deadline is approaching, as a follow-up on our rebuttal, we would like to kindly remind you that the close date of the discussion is approaching. We hope to use this open response window to discuss the paper, answer follow-up questions, and improve the quality of our paper. Have you gotten a chance to read our rebuttal, in which we tried our best to address your concerns? We want to make sure that you found our responses solid and convincing. Notice that we already provided some additional experiments results in our response to Reviewer 9T2K, and we would be more than happy to provide more information or clarification.
>
> The authors

---

### Author Rebuttal · Authors · 2023-08-10

We would like to thank all reviewers for their time and constructive comments. We will gladly incorporate the suggestions.

We observe that reviewers have two primary  concerns: the consistency of the story and the lack of empirical evaluation, which we will try to address as follows.

**The consistency of the story** How to efficiently and effectively learn from multiple source is a longstanding problem in domain adaptation. Among the effort to solve this problem, the most popular and classic method is to learn from the mixture of these sources  [KL19, MMR+21], due to its simplicity to implement, and great theoretical research value. The mixture based multi-source learning method typically contains two phases: **Phase I**, finding the 'good' mixture weight, and **Phase II**, performing ERM on the mixed domains.

Our work is exactly aimed at proposing an efficient and effective multi-source learning algorithm, by solving the two phases of problem sequentially. The first part of our work provides the first provable algorithm for finding good mixture weight. As a side contribution, from a technical perspective, it has its own value in stochastic and compositional convex-nonconcave problem.

The second part of our work addresses the **Phase II** problem, how to efficiently perform ERM on the mixed domains, when there exists multiple target domains. We cast this problem as *Co-component Empirical Risk Minimization* problem. This problem is never studied before, and we provide two possible provable ways, learning Lipschitz function with two-layer neural network and label efficient online learning approach. We believe there will be many other potential methods, and this paper is an initiative work to inspire the follow-up works.

**The experiments** To demonstrate the effective of our Algorithm 1, we implement it and run experiments on MNIST dataset, with two layer MLP model.

*Data generation*:  We constructed three distinct groups comprising a total of 10 source domains. The target domain shares the same class distribution as Group 1. The splitting and generation for MNIST non-IID data, including the classes within each group, the number of domains, and the samples per domain, are detailed in Table.1.

*Convergence of mixing parameter*: In Figure.1 (a), we observe that the alpha values for source domains in Group 1 converge to 0.5. This indicates that these domains have positive transfer to target domain. On the other hand, other source domains, which lack overlapping classes with the target domain, have alpha values of zero. This suggests that they provide no contribution during the training process of the target domain. Meanwhile, Figure.1 (b) shows the final alpha values of each source domain at the end of the training process.

*Effectiveness of the learnt mixing parameter*: We evaluated the accuracy using three distinct Error Risk Minimization methods: 1. Weighted ERM using our learnt weights, 2. ERM on averaged loss and 3. ERM solely on target domain, and presented our findings in Table.2. The results indicate that the accuracy achieved using the learned alphas outperforms the other two approaches. Additionally, the accuracy comparisons during training between Learned Alpha and Average Weight are plotted in Figure.2.

[MMR+21] Y. Mansour, M. Mohri, J. Ro, A. T. Suresh, and K. Wu. A theory of multiple-source
adaptation with limited target labeled data. In Arindam Banerjee and Kenji Fukumizu,
editors, International Conference on Artificial Intelligence and Statistics (AISTATS), volume
130 of Proceedings of Machine Learning Research, pages 2332–2340. PMLR, 13–15 Apr 2021.

[KL19] N. Konstantinov and C. H. Lampert. Robust learning from untrusted sources. In Interna-
tional Conference on Machine Learing (ICML), pages 3488–3498. PMLR, 2019

---

### Decision · Program_Chairs · 2023-09-21

**Decision:**

Accept (poster)

**Comment:**

The discussion about this paper has been rich, and the authors convinced some reviewers to raise their scores by providing empirical results that were absent from the initial submission. While doubt has been raised on whether the provided empirical results are sufficient to demonstrate the effectiveness of the proposed method, I consider that the theoretical analysis is extensive enough to justify acceptance. The theoretical contributions to domain adaptation are valuable in bringing new ideas to the community.

The rebuttal also alleviated concerns about "the consistency of the story" and the clarity of a few results. The authors need to thoughtfully incorporate the empirical results and clarify the discussion in the camera-ready version.

In addition to addressing the points raised by the reviewers, I urge the authors to cite peer-reviewed publications instead of the arXiv preprint. [20] is published at COLT, [24] at AISTATS, etc.